# The Power of Resets in Online Reinforcement Learning

**Zakaria Mhammedi**
Google Research
mhammedi@google.com

**Dylan J. Foster**
Microsoft Research
dylanfoster@microsoft.com

**Alexander Rakhlin**
MIT
rakhlin@mit.edu

## Abstract

Simulators are a pervasive tool in reinforcement learning, but most existing algorithms cannot efficiently exploit simulator access—particularly in high-dimensional domains that require general function approximation. We explore the power of simulators through online reinforcement learning with *local simulator access* (or, local planning), an RL protocol where the agent is allowed to *reset* to previously observed states and follow their dynamics during training. We use local simulator access to unlock new statistical guarantees that were previously out of reach:

1. We show that MDPs with low *coverability* [63]—a general structural condition that subsumes Block MDPs and Low-Rank MDPs—can be learned in a sample-efficient fashion with *only $Q^\star$-realizability* (realizability of the optimal state-value function); existing online RL algorithms require significantly stronger representation conditions.

2. As a consequence, we show that the notorious *Exogenous Block MDP* problem [22] is tractable under local simulator access.

The results above are achieved through a computationally-inefficient algorithm. We complement them with a more computationally efficient algorithm, RVFS (*Recursive Value Function Search*), which achieves provable sample complexity guarantees under strengthened statistical assumption known as *pushforward coverability*. RVFS can be viewed as a principled, provable counterpart to a successful empirical paradigm that combines recursive search (e.g., MCTS) with value function approximation.

## 1 Introduction

Simulators are a widely used tool in reinforcement learning. Many of the most well-known benchmarks for reinforcement learning research make use of simulators (Atari [9], MuJoCo [55], OpenAI Gym [11], DeepMind Control Suite [53]), and high-quality simulators are available for a wide range of real-world control tasks, including robotic control [45, 2], autonomous vehicles [10, 6], and game playing [51, 52]. Simulators also provide a useful abstraction for *planning* with a known or learned model, an important building block for many RL techniques [48]. Yet, in spite of the ubiquity of simulators, almost all existing research into algorithm design—empirical and theoretical—has focused on the *online reinforcement learning* (where only trajectory-based feedback is available), and does not take advantage of the extra information available through the simulator. Relatively little is known about the full power of RL with simulator access, either in terms of algorithmic principles or fundamental limits.

We explore the power of simulators through online reinforcement learning with *local simulator access* (RLLS for short), also known as *local planning* [57, 40, 3, 59, 65, 66]. Here, the agent learns by repeatedly executing policies and observing the resulting trajectories (as in online RL), but is allowed to *reset* to previously observed states and follow their dynamics during training.

Empirically, algorithms based on local simulators have received limited investigation, but with promising results. Notably, the Go-Explore algorithm [19, 20] uses local simulator access to achieve state-of-the-art performance for Montezuma's Revenge (a difficult Atari game that requires systematic

exploration), beating the performance of the best agents trained with online RL [7, 27] by a significant margin that has yet to be closed. The successful line of research on AlphaGo and successors [51, 52, 48] also uses local simulator access, albeit at test time in addition to training time.

These results suggest that developing improved algorithm design principles for RL with local simulator access could have significant practical implications, but current theoretical understanding of local simulators is limited. Recent work has shown that local simulator access has provable benefits for reinforcement learning with various types of linear function approximation [57, 40, 3, 65, 59], but essentially nothing is known for RL problems in large state spaces that demand general, potential neural function approximation. This leads us to ask: *Can we develop algorithms for reinforcement learning with general function approximation that provably benefit from local simulator access?*

From an algorithm design perspective, perhaps the greatest challenge in using local simulators to speed up learning is to understand which states are "informative" in the sense that we should prioritize revisiting them. Here, we are faced with a chicken-and-egg problem: to understand which states to prioritize, we must explore and gather information, but it is unclear how to do so efficiently unless we already have a way to understand which states are informative. It is natural to let function approximation guide us; to this end, recent research [40, 65, 59] on linearly-parameterized RL with local simulators makes use of *core-sets*: small, adaptively chosen sets of informative state-action pairs designed to cover the feature space and enable efficient value function learning. Core-sets facilitate sample complexity guarantees for linear models that are not possible without local simulator access (e.g., [40]). Yet, for general function classes—particularly rich models like neural networks that do not readily support extrapolation—defining a suitable notion of core-set is challenging. Consequently, existing techniques have yet to meaningfully leverage local simulator access beyond the linear regime.

## 1.1 Contributions

We show that local simulator access unlocks new guarantees for online RL with general value function approximation—statistical and computational—that were previously out of reach.

**Sample-efficient learning.** We show that MDPs with low *coverability* [63]—a general structural condition that subsumes Block MDPs and Low-Rank MDPs—can be learned in a sample-efficient fashion with *only $Q^\star$-realizability* (that is, realizability for the optimal state-action value function). This is achieved through a new algorithm, SimGolf, that augments the principle of global optimism with local simulator access, and improves upon the best existing guarantees for the fully online RL setting, which require significantly stronger representation conditions. As a consequence, we show for the first time that the notoriously challenging *Exogenous Block MDP* (ExBMDP) problem [22, 21] is tractable in its most general form under local simulator access.

**Practical, computationally efficient learning.** Our results above are achieved through a computationally inefficient algorithm. We complement them with a practical and computationally efficient algorithm, RVFS ("Recursive Value Function Search"), which achieves sample-efficient learning guarantees with general value function approximation under a strengthened, yet novel, statistical assumption known as *pushforward coverability* [62]. Assuming either i) realizability of the optimal state-value function $V^\star$ and a state-action gap or ii) realizability of $V^\pi$ for all $\pi$, RVFS achieves polynomial sample complexity in a computationally efficient fashion, and leads to guarantees for a new class of Exogenous Block MDPs with *weakly correlated* exogenous noise. RVFS explores by building core-sets with a novel value function-guided scheme, and can be viewed as a principled counterpart to algorithms including MCTS and AlphaZero [51, 52, 19, 20, 66], that combine recursive search with value function approximation. Compared to these approaches, RVFS is designed to provably address stochastic environments and distribution shift.

**Paper organization.** Section 2 introduces the local simulator framework. Section 3 presents our main sample complexity guarantees, and Section 4 gives computationally efficient algorithms. All proofs are deferred to the appendix.

## 2 Setup: Reinforcement Learning with Local Simulator Access

We consider an episodic reinforcement learning setting. A Markov Decision Process (MDP) is a tuple $\mathcal{M} = (\mathcal{X}, \mathcal{A}, T, R, H)$, where $\mathcal{X}$ is a (large/potentially infinite) state space, $\mathcal{A}$ is the action space (we abbreviate $A = |\mathcal{A}|$), $H \in \mathbb{N}$ is the horizon, $R = \{R_h\}_{h=1}^H$ is the reward function (where $R_h : \mathcal{X} \times \mathcal{A} \to [0, 1]$) and $T = \{T_h\}_{h=0}^H$ is the transition distribution (where $T_h : \mathcal{X} \times \mathcal{A} \to \Delta(\mathcal{X})$), with the convention that $T_0(\cdot \mid \varnothing)$ is the initial state distribution. A policy is a sequence of functions $\pi = \{\pi_h : \mathcal{X} \to$

$\Delta(\mathcal{A})\}_{h=1}^H$; we use $\Pi_S$ to denote the set of all such functions. When a policy is executed, it generates a trajectory $(\boldsymbol{x}_1, \boldsymbol{a}_1, \boldsymbol{r}_1), \ldots, (\boldsymbol{x}_H, \boldsymbol{a}_H, \boldsymbol{r}_h)$ via the process $\boldsymbol{a}_h \sim \pi_h(\boldsymbol{x}_h), \boldsymbol{r}_h \sim R_h(\boldsymbol{x}_h, \boldsymbol{a}_h), \boldsymbol{x}_{h+1} \sim T_h(\cdot \mid \boldsymbol{x}_h, \boldsymbol{a}_h)$, initialized from $\boldsymbol{x}_1 \sim T_0(\cdot \mid \varnothing)$ (we use $\boldsymbol{x}_{H+1}$ to denote a terminal state with zero reward). We write $\mathbb{P}^\pi[\cdot]$ and $\mathbb{E}^\pi[\cdot]$ to denote the law and expectation under this process.

For a policy $\pi$, $J(\pi) \coloneqq \mathbb{E}^\pi\big[\sum_{h=1}^H \boldsymbol{r}_h\big]$ denotes expected reward, and the value functions are given by $V_h^\pi(x) \coloneqq \mathbb{E}^\pi\big[\sum_{h'=h}^H \boldsymbol{r}_{h'} \mid \boldsymbol{x}_h = x\big]$, and $Q_h^\pi(x, a) \coloneqq \mathbb{E}^\pi\big[\sum_{h'=h}^H \boldsymbol{r}_{h'} \mid \boldsymbol{x}_h = x, \boldsymbol{a}_h = a\big]$. We denote by $\pi^\star$ the optimal deterministic policy that maximizes $Q^{\pi^\star}$, and write $Q^\star \coloneqq Q^{\pi^\star}$ and $V^\star \coloneqq V^{\pi^\star}$.

**Online reinforcement learning with a local simulator.** In the standard online reinforcement learning framework, the learner repeatedly interacts with an (unknown) MDP by executing a policy and observing the resulting trajectory, with the goal of maximizing the total reward. Formally, for each episode $\tau \in [N_{\mathsf{episodes}}]$, the learner selects a policy $\pi^{(\tau)} = \{\pi_h^{(\tau)}\}_{h=1}^H$, executes it in the underlying MDP $\mathcal{M}^\star$ and observes the trajectory $\{(\boldsymbol{x}_h^{(\tau)}, \boldsymbol{a}_h^{(\tau)}, \boldsymbol{r}_h^{(\tau)})\}_{h=1}^H$. After all $N_{\mathsf{episodes}}$ episodes conclude, the learner produces a policy $\widehat{\pi} \in \Pi_S$ with the goal of minimizing the risk given by $\mathbb{E}[J(\pi^\star) - J(\widehat{\pi})]$.

In online RL with local simulator access, or RLLS, [57, 40, 65, 59, 66], we augment the online RL protocol as follows: At each episode $\tau \in [N]$, instead of starting from a random initial state $\boldsymbol{x}_1 \sim T_0(\cdot \mid \varnothing)$, the agent can *reset* the MDP to any layer $h \in [H]$ and any state $\boldsymbol{x}_h$ previously encountered, and proceed with a new episode starting from $\boldsymbol{x}_h$. As in the online RL protocol, the goal is to produce a policy $\widehat{\pi} \in \Pi_S$ such that $\mathbb{E}[J(\pi^\star) - J(\widehat{\pi})] \le \varepsilon$ with as few episodes of interaction as possible; our main results take $N_{\mathsf{episodes}} = \mathrm{poly}(C, \varepsilon^{-1})$ for a suitable problem parameter $C$.

**Executable versus non-executable policies.** We focus on learning policies that can be executed without access to a local simulator (in other words, the local simulator used at train time, but not test time). Some recent work using local simulators for RL with linear function approximation [57] considers a more permissive setting where the final policy $\pi$ produced by the learner can be *non-executable*; our function approximation requirements can be slightly relaxed in this case.

**Definition 2.1** (Non-executable policy). *We refer to a policy $\pi$ for which computing $\pi(x) \in \Delta(\mathcal{A})$ for any $x \in \mathcal{X}$ requires $n$ local simulator queries as a* non-executable policy with sample complexity $n$.

**Additional notation.** For any $m, n \in \mathbb{N}$, we denote by $[m..n]$ the integer interval $\{m, \ldots, n\}$. We also let $[n] \coloneqq [1..n]$. We refer to a scalar $c > 0$ as an *absolute constant* to indicate that it is independent of all problem parameters and use $\widetilde{O}(\cdot)$ to denote a bound up to factors poly-logarithmic in parameters appearing in the expression. We define $\pi_{\mathsf{unif}} \in \Pi_S$ as the random policy that selects actions in $\mathcal{A}$ uniformly. We define the occupancy measure for policy $\pi$ via $d_h^\pi(x, a) \coloneqq \mathbb{P}^\pi[\boldsymbol{x}_h = x, \boldsymbol{a}_h = a]$. For functions $g : \mathcal{X} \times \mathcal{A} \to \mathbb{R}$ and $f : \mathcal{X} \to \mathbb{R}$, we define Bellman backup operators by $\mathcal{T}_h[g](x, a) = \mathbb{E}[\boldsymbol{r}_h + \max_{a' \in \mathcal{A}} g(\boldsymbol{x}_{h+1}, a') \mid \boldsymbol{x}_h = x, \boldsymbol{a}_h = a]$ and $\mathcal{P}_h[f](x, a) = \mathbb{E}[\boldsymbol{r}_h + f(\boldsymbol{x}_{h+1}) \mid \boldsymbol{x}_h = x, \boldsymbol{a}_h = a]$. For a stochastic policy $\pi \in \Pi_S$, we will occasionally use the bold notation $\boldsymbol{\pi}_h(x)$ as shorthand for the random variable $\boldsymbol{a}_h \sim \pi_h(x) \in \Delta(\mathcal{A})$. For a function $f : \mathcal{A} \to \mathbb{R}$, we write $a' \in \arg\max_{a \in \mathcal{A}} f(a)$ to denote the action that maximizes $f$. If there are ties, we break them by picking the action with the smallest index; we assume without loss of generality that actions in $\mathcal{A}$ are index from $1, \ldots, |\mathcal{A}|$.

## 3 New Sample-Efficient Learning Guarantees via Local Simulators

This section presents our most powerful results for RLLS. We present a new algorithm for learning with local simulator access, SimGolf (Section 3.1), and show that it enables sample-efficient RL for MDPs with low *coverability* [63] using only $Q^\star$-realizability (Section 3.2). We then give implications for the Exogenous Block MDP problem (Section 3.3).

**Function approximation setup and coverability.** To achieve sample complexity guarantees for online reinforcement learning that are suitable for large, high-dimensional state spaces, we appeal to *value function approximation*. We assume access to a function class $\mathcal{Q} \subset (\mathcal{X} \times \mathcal{A} \times [H] \to [0, H])$ that contains the optimal state-action value function $Q^\star$; we define $\mathcal{Q}_h = \{Q_h \mid Q \in \mathcal{Q}\}$.

**Assumption 3.1** ($Q^\star$-realizability). *For all $h \in [H]$, we have $Q_h^\star \in \mathcal{Q}_h$.*

$Q^\star$-realizability is widely viewed as a minimal representation condition for online RL [61, 17, 16, 39, 58, 56]. The class $\mathcal{Q}$ encodes the learner's prior knowledge about the MDP, and can be parameterized by rich function approximators like neural networks. We assume for simplicity of exposition that $\mathcal{Q}$

and $\Pi$ are finite, and aim for sample complexity guarantees scaling with $\log|\mathcal{Q}|$ and $\log|\Pi|$; extending our results to infinite classes via standard uniform convergence arguments is straightforward.

**Coverability.** Beyond representation conditions like realizability, online RL algorithms require *structural conditions* that limit the extent to which deliberately designed algorithms can be surprised by substantially new state distributions. We focus on a structural condition known as *coverability* [63], which is inspired by connections between online and offline RL.

**Assumption 3.2.** *The* coverability *coefficient is* $C_{\mathsf{cov}} \coloneqq \max_{h\in[H]} \inf_{\mu_h \in \Delta(\mathcal{X}\times\mathcal{A})} \sup_{\pi\in\Pi_{\mathsf{S}}} \left\| \frac{d_h^\pi}{\mu_h} \right\|_\infty$.

Coverability is an intrinsic strutural property of the underlying MDP. Examples of MDP families with low coverability include (Exogenous) Block MDPs, which have $C_{\mathsf{cov}} \le |\mathcal{S}||\mathcal{A}|$, where $\mathcal{S}$ is the *latent state space* [63], and Low-Rank MDPs, which have $C_{\mathsf{cov}} \le d|\mathcal{A}|$, where $d$ is the feature dimension [29]; importantly, these settings exhibit high-dimensional state spaces and require nonlinear function approximation. As in prior work [63, 4], our algorithms require *no prior knowledge* of the distribution $\mu_h$ that achieves the minimum in Assumption 3.2.

### 3.1 Algorithm

Our main algorithm, SimGolf, is displayed in Algorithm 1. The algorithm is a variant of the GOLF method of Jin et al. [32], Xie et al. [63] with novel adaptations to exploit the availability of a local simulator. Like GOLF, SimGolf explores using the principle of *global optimisim*: At each iteration $t \in [N]$, it maintains a confidence set (or, version space) $\mathcal{Q}^{(t)} \subset \mathcal{Q}$ of candidate value functions with low squared Bellman error under the data collected so far, and chooses a new exploration policy $\pi^{(t)}$ by picking the most "optimistic" value function in this set. As the algorithm gathers more data, the confidence set shrinks, leaving only near-optimal policies.

The main novelty in SimGolf arises in the data collection strategy and design of confidence sets. Like GOLF, SimGolf algorithm constructs the confidence set $\mathcal{Q}^{(t)} \subset \mathcal{Q}$ such that all value functions $g \in \mathcal{Q}^{(t)}$ have small squared Bellman error:

$$\sum_{i<t} \mathbb{E}^{\pi^{(i)}}\left[ \left( g_h(\boldsymbol{x}_h, \boldsymbol{a}_h) - \mathcal{T}_h[g_{h+1}](\boldsymbol{x}_h, \boldsymbol{a}_h) \right)^2 \right] \lesssim \log|\mathcal{Q}|, \quad \forall h \in [H]. \tag{1}$$

Due to the presence of the Bellman backup $\mathcal{T}_h[g_{h+1}]$ in Eq. (1), naively estimating squared Bellman error leads to the notorious *double sampling* problem. To avoid this, the approach taken with GOLF and related work [67, 32] is to adapt a certain de-biasing technique to remove double sampling bias, but this requires access to a value function class that satisfies *Bellman completeness*, a representation significantly more restrictive than realizability (e.g., Foster et al. [26]).

The idea behind SimGolf is to use local simulator access to directly produce high-quality estimates for the Bellman backup function $\mathcal{T}_h[g_{h+1}]$ in Eq. (1). In particular, for a given state-action pair $(x, a) \in \mathcal{X} \times \mathcal{A}$, we can estimate the Bellman backup $\mathcal{T}_h[g_{h+1}](x, a)$ for all functions $g \in \mathcal{Q}$ simultaneously by collecting $K$ next-state transitions $\widetilde{\boldsymbol{x}}_{h+1}^{(1)}, \ldots, \widetilde{\boldsymbol{x}}_{h+1}^{(K)} \overset{\text{i.i.d.}}{\sim} T_h(\cdot \mid x, a)$ and $K$ rewards $\widetilde{\boldsymbol{r}}_h^{(1)}, \ldots, \widetilde{\boldsymbol{r}}_h^{(K)} \overset{\text{i.i.d.}}{\sim} R_h(x, a)$, then taking the empirical mean: $\mathcal{T}_h[g_{h+1}](x, a) \approx \frac{1}{K}\sum_{k=1}^K (\widetilde{\boldsymbol{r}}_h^{(k)} + \max_{a'\in\mathcal{A}} g_{h+1}(\widetilde{\boldsymbol{x}}_{h+1}^{(k)}, a'))$. Line 8 of SimGolf uses this technique to directly estimate the Bellman residual backup under a trajectory gathered with $\pi^{(t)}$, sidestepping the double sampling problem and removing the need for Bellman completeness. We suspect this technique (estimation with respect to squared Bellman error using local simulator access) may find broader use.

### 3.2 Main Result

We now state the main guarantee for SimGolf and discuss some of its implications.

**Theorem 3.1** (Main guarantee for SimGolf). *Let $\varepsilon, \delta \in (0, 1)$ be given and suppose Assumption 3.1 ($Q^\star$-realizability) and Assumption 3.2 (coverability) hold with $C_{\mathsf{cov}} > 0$. Then the policy $\widehat{\pi}$ produced by* SimGolf$(\mathcal{Q}, C_{\mathsf{cov}}, \varepsilon, \delta)$ *(Algorithm 1) has $J(\pi^\star) - \mathbb{E}[J(\widehat{\pi})] \le \varepsilon$ with probability at least $1 - \delta$. The total sample complexity in the RLLS framework is bounded by $\widetilde{O}\left( H^5 C_{\mathsf{cov}}^2 \log(|\mathcal{Q}|/\delta) \cdot \varepsilon^{-4} \right)$.*

This result (whose proof is in Appendix E) shows that under only $Q^\star$-realizability and coverability, SimGolf learns an $\varepsilon$-optimal policy with polynomial sample complexity, significantly relaxing the representation assumptions (Bellman completeness, weight function realizability) required by prior algorithms for coverability [63, 4]. This is the first instance we are aware of where local simulator

**Algorithm 1** `SimGolf`: Global Optimism via Local Simulator Access

---

1: **input:** Value function class $\mathcal{Q}$, coverability $C_{\mathsf{cov}} > 0$, suboptimality $\varepsilon > 0$, and confidence $\delta > 0$.
2: Set $N \leftarrow \widetilde{\Theta}(H^2 C_{\mathsf{cov}} \beta / \varepsilon^2)$, $\beta_{\mathsf{stat}} \leftarrow 16 \log(2HN|\mathcal{Q}|\delta^{-1})$, $\beta \leftarrow 2\beta_{\mathsf{stat}}$, and $K \leftarrow \frac{8N}{\beta_{\mathsf{stat}}}$.
3: **initialize:** $\mathcal{Q}^{(1)} \leftarrow \mathcal{Q}$.
4: **for** iteration $t = 1, 2, \ldots, N$ **do**
5:     Select $g^{(t)} = \arg\max_{g \in \mathcal{Q}^{(t)}} \sum_{s=1}^{t-1} \max_{a \in \mathcal{A}} g_1(\boldsymbol{x}_1^{(s)}, a)$.
6:     For each $h \in [H]$ and $x \in \mathcal{X}$, define $\pi_h^{(t)}(x) \in \arg\max_{a \in \mathcal{A}} g_h^{(t)}(x, a)$.
7:     Execute $\pi^{(t)}$ for an episode and observe $\boldsymbol{\tau}^{(t)} := (\boldsymbol{x}_1^{(t)}, \boldsymbol{a}_1^{(t)}), \ldots, (\boldsymbol{x}_H^{(t)}, \boldsymbol{a}_H^{(t)})$.
8:     For $h \in [H]$, draw $K$ independent samples $\boldsymbol{x}_{h+1}^{(t,k)} \sim T_h(\cdot \mid \boldsymbol{x}_h^{(t)}, \boldsymbol{a}_h^{(t)})$, $\boldsymbol{r}_h^{(t,k)} \sim R_h(\boldsymbol{x}_h^{(t)}, \boldsymbol{a}_h^{(t)})$.
9:     Compute confidence set:

$$\mathcal{Q}^{(t+1)} \leftarrow \left\{ g \in \mathcal{Q} : \sum_{s \leq t} \left( g_h(\boldsymbol{x}_h^{(s)}, \boldsymbol{a}_h^{(s)}) - \frac{1}{K} \sum_{k=1}^{K} \left( \boldsymbol{r}_h^{(s,k)} + \max_{a \in \mathcal{A}} g_{h+1}(\boldsymbol{x}_{h+1}^{(s,k)}, a) \right) \right)^2 \leq \beta, \ \forall h \in [H] \right\}.$$

10: **return:** $\widehat{\pi} = \mathtt{unif}(\pi^{(1)}, \ldots, \pi^{(N)})$.

---

access unlocks sample complexity guarantees for reinforcement learning with *nonlinear* function approximation that were previously out of reach; perhaps the most important technical idea here is our approach to combining global optimism with local simulator access, in contrast to greedy layer-by-layer schemes used in prior work on local simulators (with the exception of Weisz et al. [57]). In particular, we suspect that the idea of performing estimation with respect to squared Bellman error directly using local simulator access may find broader use beyond coverability. Improving the polynomial dependence on problem parameters is an interesting question for future work.

**A conjecture.** By analogy to results in offline reinforcement learning, where $Q^\star$-realizability and concentrability (the offline counterpart to coverability) alone are known to be insufficient for sample-efficient learning [12, 26], we conjecture that $Q^\star$-realizability and coverability alone are not sufficient for polynomial sample complexity in vanilla online RL. If true, this would imply a new separation between online RL with and without local simulators.

### 3.3 Implications for Exogenous Block MDPs

We now apply `SimGolf` and Theorem 3.1 to the *Exogenous Block MDP* (ExBMDP) problem [22, 21, 38, 30], a challenging rich-observation reinforcement learning setting in which the observed states $\boldsymbol{x}_h$ are high-dimensional, while the underlying dynamics of the system are low-dimensional, yet confounded by temporally correlated exogenous noise.

Formally, an Exogenous Block MDP $\mathcal{M} = (\mathcal{X}, \mathcal{S}, \Xi, \mathcal{A}, H, T, R, g)$ is defined by a *latent state space* and an *observation space*. We begin with the latent state space. Starting from an initial *endogenous state* $\boldsymbol{s}_1 \in \mathcal{S}$ and *exogenous state* $\boldsymbol{\xi}_1 \in \Xi$, the latent state $\boldsymbol{z}_h = (\boldsymbol{s}_h, \boldsymbol{\xi}_h)$ evolves for $h \in [H]$ via $\boldsymbol{s}_{h+1} \sim T_h^{\mathsf{endo}}(\cdot \mid \boldsymbol{s}_h, \boldsymbol{a}_h)$ and $\boldsymbol{\xi}_{h+1} \sim T_h^{\mathsf{exo}}(\cdot \mid \xi_h)$, where $\boldsymbol{a}_h \in \mathcal{A}$ is the agent's action at layer $h$; we adopt the convention that $\boldsymbol{s}_1 \sim T_0^{\mathsf{endo}}(\cdot \mid \varnothing)$ and $\boldsymbol{\xi}_1 \sim T_0^{\mathsf{exo}}(\cdot \mid \varnothing)$. Note that only the endogenous state is causally influenced by the action. The latent state is not observed; instead, at each step $h$, the agent receives an *observation* $\boldsymbol{x}_h \in \mathcal{X}$ generated via[1] $\boldsymbol{x}_h = g_h^{\mathsf{obs}}(\boldsymbol{s}_h, \boldsymbol{\xi}_h)$, where $g_h^{\mathsf{obs}} : \mathcal{S} \times \Xi \to \mathcal{X}$ is the *emission function*. We assume the endogenous latent space $\mathcal{S}$ and action space $\mathcal{A}$ are finite, and define $S := |\mathcal{S}|$ and $A := |\mathcal{A}|$. However, the exogenous state space $\Xi$ and observation space $\mathcal{X}$ may be arbitrarily large or infinite, with $|\Xi|, |\mathcal{X}| \gg |\mathcal{S}|$.[2]

The final property of the ExBMDP model is *decodability*, which asserts the existence of a *decoder* such that $\phi_\star : \mathcal{X} \to \mathcal{S}$ such that $\phi_\star(\boldsymbol{x}_h) = \boldsymbol{s}_h$ a.s. for all $h \in [H]$ with $\boldsymbol{x}_h = g_h^{\mathsf{obs}}(\boldsymbol{s}_h, \boldsymbol{\xi}_h)$,. Informally, decodability ensures the existence of an (unknown to the learner) mapping that allows one to perfectly recover the endogenous latent state from observations. In addition to decodability, we assume the rewards in the ExBMDP are *endogenous*; that is, the reward distribution $R_h(\boldsymbol{x}_h, \boldsymbol{a}_h)$ only

---

[1]A more standard formulation [22, 21, 38, 30] assumes that observations are generated via $\boldsymbol{x}_h \sim q_h(\boldsymbol{s}_h, \boldsymbol{\xi}_h)$, where $q_h(\cdot, \cdot)$ is a conditional distribution with the decodability property. This is equivalent to $\boldsymbol{x}_h = g_h^{\mathsf{obs}}(\boldsymbol{s}_h, \boldsymbol{\xi}_h)$, as randomness in the emission process can be included in the exogenous state w.l.o.g.

[2]To simplify presentation, we assume that $\Xi$ and $\mathcal{X}$ are countable; our results trivially extend to the case where the corresponding variables are continuous with an appropriate measure-theoretic treatment.

depends on the observations $(\boldsymbol{x}_h)$ through the corresponding latent states $(\phi^\star(\boldsymbol{x}_h) = \boldsymbol{s}_h)$. To enable sample-efficient learning, we assume access to a *decoder class* $\Phi$ that contains $\phi^\star$, as in prior work.

**Assumption 3.3** (Decoder realizability). *We have access to a decoder class $\Phi$ such that $\phi^\star \in \Phi$.*

**Applying `SimGolf` and Theorem 3.1.** To apply Theorem 3.1 to the ExBMDP problem, we need to verify that $Q^\star$-realizability and coverability hold. Realizability is a straightforward consequence of decodability (Lemma D.1 in Part II of the appendix). For coverability, Xie et al. [63] show that ExBMDPs have $C_{\text{cov}} \leq SA$ under decodability, in spite of the time-correlated exogenous noise process $(\boldsymbol{\xi}_h)$ and potentially infinite observation space $\mathcal{X}$ (interestingly, coverability is essentially the only useful structural property that ExBMDPs are known to satisfy, which is our primary motivation for studying it). This leads to the following corollary of Theorem 3.1.

**Corollary 3.1** (`SimGolf` for ExBMDPs). *Consider the ExBMDP setting. Suppose that Assumption 3.3 holds, and let $\mathcal{Q}$ be constructed as in Lemma D.1 of Part II. Then for any $\varepsilon, \delta \in (0, 1)$, the policy $\widehat{\pi} = \texttt{SimGolf}(\mathcal{Q}, SA, \varepsilon, \delta)$ has $J(\pi^\star) - J(\widehat{\pi}) \leq \varepsilon$ with probability at least $1 - \delta$. The total sample complexity in the RLLS framework is $N = \widetilde{O}(H^5 S^3 A^3 \log|\Phi| \cdot \varepsilon^{-4})$.*

This shows for the first time that general ExBMDPs are learnable with local simulator access. Prior to this work, online RL algorithms for ExBMDPs required either (i) deterministic latent dynamics [22], or (ii) factored emission structure [21]. Xie et al. [63] observed that ExBMDPs admit low coverability, but their algorithm requires Bellman completeness, which is not satisfied by ExBMDPs (see Islam et al. [30]). See Appendix A for more discussion.

# 4 Computationally Efficient Learning with Local Simulators

Our result in Section 3 show that local simulator access facilitates sample-efficient learning in MDPs with low coverability, a challenging setting that was previously out of reach. However, our algorithm `SimGolf` is computationally-inefficient because it relies on global optimism, a drawback found in most prior work on RL with general function approximation [31, 32, 18]. It remains an open question whether any form of global optimism can be implemented efficiently, and some variants have provable barriers to efficient implementation [14].

To address this drawback, in this section we present a new algorithm, `RVFS` (Recursive Value Function Search; Algorithm 5), which requires stronger versions of the coverability and realizability assumptions in Section 3, but is computationally efficient in the sense that it reduces to convex optimization over the state-value function class $\mathcal{V}$. `RVFS` makes use of a sophisticated recursive exploration scheme based on core-sets, sidestepping the need for global optimism.

## 4.1 Function Approximation and Statistical Assumptions

To begin, we require the following strengthening of the coverability assumption in Assumption 3.2.

**Assumption 4.1** (Pushforward coverability). *The pushforward coverability coefficient $C_{\text{push}} > 0$ is given by* $C_{\text{push}} = \max_{h \in [H]} \inf_{\mu_h \in \Delta(\mathcal{X})} \sup_{(x_{h-1}, a_{h-1}, x_h) \in \mathcal{X}_{h-1} \times \mathcal{A} \times \mathcal{X}} \frac{T_{h-1}(x_h | x_{h-1}, a_{h-1})}{\mu_h(x_h)}.$

Pushforward coverability is inspired by the *pushforward concentrability* condition used in offline RL by [62, 26]. Concrete examples include, (i) Block MDPs with latent space $\mathcal{S}$, which admit $C_{\text{push}} \leq |\mathcal{S}|$, (ii) Low-Rank MDPs in dimension $d$, which admit $C_{\text{push}} \leq d$ [62], and (iii) Exogenous Block MDPs for which the exogenous noise process satisfies a *weak correlation condition* that we introduce in Appendix B. Note that $C_{\text{cov}} \leq C_{\text{push}}|\mathcal{A}|$, but the converse is not true in general.

Instead of state-action value function approximation as in `SimGolf`, in this section we make use of a state value function class $\mathcal{V} \subset (\mathcal{X} \times [H] \to [0, H])$, but require somewhat stronger representation conditions than in Section 3. We consider two complementary setups:

- **Setup I:** Assumptions 4.2 and 4.3 ($V^\star/\pi^\star$-realizability) and Assumption 4.4 ($\Delta$-gap) hold.

- **Setup II:** Assumption 4.5 ($V^\pi$-realizability) and Assumption 4.6 ($\pi$-realizability) hold.

We describe these assumptions in more detail below.

**Function approximation setup I.** First, instead of $Q^\star$-realizability, we consider the weaker $V^\star$-realizability [31, 57, 3].

**Assumption 4.2** ($V^\star$-realizability). *For all $h \in [H]$, we have $V_h^\star \in \mathcal{V}_h$.*

Under $V^\star$-realizability, our algorithm learns a near-optimal policy, but the policy is *non-executable* (cf. Definition 2.1); this property is shared by prior work on local simulator access with value function realizability [57] . To produce executable policies, we additionally require access to a policy class $\Pi \subset \Pi_S$ containing $\pi^\star$; we define $\Pi_h = \{\pi_h \mid \pi \in \Pi\}$.

**Assumption 4.3** ($\pi^\star$-realizability). *The policy class $\Pi$ contains the optimal policy $\pi^\star$.*

$V^\star$-realizability (Assumption 4.2) and $\pi^\star$-realizability (Assumption 4.3) are both implied by $Q^\star$-realizability, and hence are weaker. However, we also assume the optimal $Q$-function admits constant gap (this makes the representation conditions for **Setup I** incomparable to Assumption 3.1).

**Assumption 4.4** ($\Delta$-Gap). *The optimal action $\pi_h^\star(x)$ is unique, and there exists $\Delta > 0$ such that for all $h \in [H]$, $x \in \mathcal{X}$, and $a \in \mathcal{A} \smallsetminus \{\pi_h^\star(x)\}$, $Q_h^\star(x, \pi_h^\star(x)) > Q_h^\star(x, a) + \Delta$.*

This condition has been used in a many prior works on computationally efficient RL with function approximation [16, 17, 24, 56].

**Function approximation setup II.** We also provide guarantees under the assumption that the class $\mathcal{V}$ satisfies *all-policy realizability* [59, 65, 60] in the sense that $V^\pi \in \mathcal{V}$ for all $\pi \in \Pi_S$.

**Assumption 4.5** ($V^\pi$-realizability). *The class $\mathcal{V} = \mathcal{V}_{1:H}$ has $V_h^\pi \in \mathcal{V}_h$ for all $\pi \in \Pi_S$ and $h \in [H]$.*

This assumption will be sufficient to learn a non-executable policy, but to learn executable policies we require an analogous strengthening of Assumption 4.5.

**Assumption 4.6** ($\pi$-realizability). *For all $\pi \in \Pi_S$, we have that $x \mapsto \arg\max_{a \in \mathcal{A}} \mathcal{P}_h[V_{h+1}^\pi](\cdot, a) \in \Pi$.*

This assumption has been used by a number of prior works on computationally efficient RL [8, 44]. Assumptions 4.5 and 4.6 are both implied by the slightly simpler-to-state assumption of $Q^\pi$-*realizability* [59, 65, 60], which asserts access to a class $\mathcal{Q}$ that contains $Q^\pi$ for all $\pi \in \Pi_S$.

### 4.2 Algorithm

For ease of exposition, we defer the full version of our algorithm, RVFS (Algorithm 5), to Appendix F and present a simplified version here (Algorithm 2). The algorithms are nearly identical, except that the simplified version assumes that certain quantities of interest (e.g., Bellman backups) can be computed exactly, while the full version (provably) approximates them from samples.

RVFS maintains a value function estimator $\widehat{V} = \widehat{V}_{1:H}$ that aims to approximate the optimal value function $V_{1:H}^\star$, as well as *core sets* $\mathcal{C}_1, \ldots, \mathcal{C}_H$ of state-action pairs that are used to perform estimation and guide exploration. At a high level, RVFS alternates between (i) fitting the value function $\widehat{V}_h$ for a given layer $h \in [H]$ based on Monte-Carlo rollouts, and (ii) using the core-sets to test whether the current value function estimates $\widehat{V}_{h+1:H}$ remain accurate as the roll-in policy induced by $\widehat{V}_h$ changes.

In more detail, RVFS is based on recursion across the layers $h \in [H]$. When invoked for layer $h$ with value function estimates $\widehat{V}_{h+1:H}$ and core-sets $\mathcal{C}_h, \ldots, \mathcal{C}_H$, RVFS$_h$ performs two steps:

1. For each state-action pair $(x_{h-1}, a_{h-1}) \in \mathcal{C}_h$,[4] the algorithm gathers $N_{\text{test}}$ trajectories by rolling out from $(x_{h-1}, a_{h-1})$ with the greedy policy $\widehat{\pi}_\ell(x) \in \arg\max_{a \in \mathcal{A}} \mathcal{P}_\ell[\widehat{V}_{\ell+1}](x, a)$ that optimizes the estimated value function; in the full version of RVFS (see Algorithm 5), we estimate the bellman backup $\mathcal{P}_\ell[\widehat{V}_{\ell+1}](x, a)$ using the local simulator. For all states $x_{\ell-1} \in \{\boldsymbol{x}_h, \ldots, \boldsymbol{x}_{H-1}\}$ encountered during this process, the algorithm checks whether $\left|\mathbb{E}[\widehat{V}_\ell(\boldsymbol{x}_\ell) - V_\ell^\star(\boldsymbol{x}_\ell) \mid \boldsymbol{x}_{\ell-1} = x_{\ell-1}, \boldsymbol{a}_{\ell-1} = a_{\ell-1}]\right| \lesssim \varepsilon$ for all $a_{\ell-1} \in \mathcal{A}$ using a test based on (implicitly maintained) confidence sets. If the test fails, this indicates that distribution shift has occurred, and the algorithm adds the pair $(x_{\ell-1}, a_{\ell-1})$ to $\mathcal{C}_\ell$ and recurses on layer $\ell$ via RVFS$_\ell$.

2. If all tests above pass, this means that $\widehat{V}_{h+1}, \ldots, \widehat{V}_H$ are accurate, and no distribution shift has occurred. In this case, the algorithm fits $\widehat{V}_h$ by collecting Monte-Carlo rollouts from all state-action pairs in the core-set $\mathcal{C}_h$ with $\widehat{\pi}_\ell(x) \in \arg\max_{a \in \mathcal{A}} \mathcal{P}_\ell[\widehat{V}_{\ell+1}](x, a)$ (cf. Line 16), and returns.

When the tests in Item 1 succeed for all $h \in [H]$, the algorithm returns the estimated value functions $\widehat{V}_{1:H}$; in this case, the greedy policy $\widehat{\pi}_\ell(x) \in \arg\max_{a \in \mathcal{A}} \mathcal{P}_\ell[\widehat{V}_{\ell+1}](x, a)$ is guaranteed to be near

---

[4]Informally, $\mathcal{C}_h$ represents a collection of state-action pairs $(x_{h-1}, a_{h-1})$ at layer $h-1$ for which we want $\mathbb{E}[|\widehat{V}_h(\boldsymbol{x}_h) - V_h^\star(\boldsymbol{x}_h)| \mid \boldsymbol{x}_{h-1} = x_{h-1}, \boldsymbol{a}_{h-1} = a_{h-1}] \leq \varepsilon$ for some small $\varepsilon > 0$.

---

**Algorithm 2** $\mathrm{RVFS}_h$: Recursive Value Function Search (Informal version of Algorithm 5)

1: **parameters:** Value function class $\mathcal{V}$, suboptimality $\varepsilon \in (0,1)$, confidence $\delta \in (0,1)$.
2:   **input:** Level $h \in [0\mathbin{..}H]$, value functions $\widehat{V}_{h+1:H}$, confidence sets $\widehat{\mathcal{V}}_{h+1:H}$, core-sets $\mathcal{C}_{h:H}$.
3: Initialize parameters $M$, $N_{\text{test}}$, $N_{\text{reg}}$, $\varepsilon_{\text{reg}}^2$, and $\beta$ (see Algorithm 5 for parameter settings).
    `/* Test the fit for the estimated value functions` $\widehat{V}_{h+1:H}$ `at future layers. */`
4: **for** $(x_{h-1}, a_{h-1}) \in \mathcal{C}_h$ and $\ell = H, \dots, h+1$ **do**
5:     **for** $n = 1, \dots, N_{\text{test}}$ **do**
6:         Draw $\boldsymbol{x}_h \sim T_{h-1}(\cdot \mid x_{h-1}, a_{h-1})$, then draw $\boldsymbol{x}_{\ell-1}$ by rolling out with $\widehat{\pi}_{h:H}$, where[3]

$$\forall \tau \in [H], \quad \widehat{\pi}_\tau(\cdot) \in \arg\max_{a \in \mathcal{A}} \mathcal{P}_\tau[\widehat{V}_{\tau+1}](\cdot, a). \tag{2}$$

7:         **for** $a_{\ell-1} \in \mathcal{A}$ **do**
            `/* Test fit; if test fails, re-fit value functions` $\widehat{V}_{h+1:\ell}$ `up to layer` $\ell$`. */`
8:             **if** $\sup_{f \in \widehat{\mathcal{V}}_\ell} |(\mathcal{P}_{\ell-1}[\widehat{V}_\ell] - \mathcal{P}_{\ell-1}[f_\ell])(\boldsymbol{x}_{\ell-1}, a_{\ell-1})| > \varepsilon + \varepsilon \cdot \beta$ **then**
9:                 $\mathcal{C}_\ell \leftarrow \mathcal{C}_\ell \cup \{(\boldsymbol{x}_{\ell-1}, a_{\ell-1})\}$.
10:                **for** $\tau = \ell, \dots, h+1$ **do**
11:                    $(\widehat{V}_{\tau:H}, \widehat{\mathcal{V}}_{h:H}, \mathcal{C}_{\tau:H}) \leftarrow \mathrm{RVFS}_\tau(\widehat{V}_{\tau+1:H}, \widehat{\mathcal{V}}_{h+1:H}, \mathcal{C}_{\tau:H}; \mathcal{V}, \varepsilon, \delta)$.
12:                **go to line 4.**
13: **if** $h = 0$ **then return:** $(\widehat{V}_{1:H}, \cdot, \cdot, \cdot)$.
    `/* Re-fit` $\widehat{V}_h$ `and build a new confidence set. */`
14: **for** $(x_{h-1}, a_{h-1}) \in \mathcal{C}_h$ **do**     `//` $\mathbb{E}^{\widehat{\pi}_{h:H}}[\sum_{\ell=h}^{H} r_\ell \mid x_h]$ `can be estimated using local simulator.`
15:     Set $\mathcal{D}_h(x_{h-1}, a_{h-1}) \leftarrow \varnothing$. For $i = 1, \dots, N_{\text{reg}}$, sample $\boldsymbol{x}_h \sim T_{h-1}(\cdot \mid x_{h-1}, a_{h-1})$ and update
    $\mathcal{D}_h(x_{h-1}, a_{h-1}) \leftarrow \mathcal{D}_h(x_{h-1}, a_{h-1}) \cup \{(\boldsymbol{x}_h, \mathbb{E}^{\widehat{\pi}_{h:H}}[\sum_{\ell=h}^{H} r_\ell \mid \boldsymbol{x}_h])\}$.
16: Let $\widehat{V}_h := \arg\min_{f \in \widehat{\mathcal{V}}} \sum_{(x_{h-1}, a_{h-1}) \in \mathcal{C}_h} \sum_{(x_h, v_h) \in \mathcal{D}_h(x_{h-1}, a_{h-1})} (f(x_h) - v_h)^2$.
17: Compute value function confidence set:

$$\widehat{\mathcal{V}}_h := \left\{ f \in \mathcal{V} \;\middle|\; \sum_{(x_{h-1}, a_{h-1}) \in \mathcal{C}_h} \frac{1}{N_{\text{reg}}} \sum_{(x_h, \cdot) \in \mathcal{D}_h(x_{h-1}, a_{h-1})} \left(\widehat{V}_h(x_h) - f(x_h)\right)^2 \le \varepsilon_{\text{reg}}^2 \right\}.$$

18: **return** $(\widehat{V}_{h:H}, \widehat{\mathcal{V}}_{h:H}, \mathcal{C}_{h:H})$.

---

optimal. The full version of RVFS in Algorithm 5 uses local simulator access to estimate the Bellman backups $\mathcal{P}_h[\widehat{V}_{h+1}](x, a)$ for different state-action pairs $(x, a)$. These backups are used to (i) compute actions of the greedy policy that maximizes $\widehat{V}_{1:H}$ via (e.g., Eq. (2)); (ii) generate trajectories by rolling out from state-action pairs in the core-sets (Line 6); and (iii) perform the test in Item 1 (Line 8).

RVFS is inspired by the DMQ algorithm [16, 56] originally introduced in the context of online reinforcement learning with linearly realizable $Q^\star$. RVFS incorporates local simulator access (most critically, via core-set construction) to allow for more general *nonlinear* function approximation without restrictive statistical assumptions. Prior algorithms for RLLS have used core-sets of state-action pairs in a similar fashion [40, 65, 59], but in a way that is tailored to linear function approximation.

In what follows, we discuss various features of the algorithm in greater detail.

**Bellman backup policies.** Since RVFS works with state value functions instead of state-action value functions, we need a way to extract policies from the former. The most natural way to extract a policy from estimated value functions $\widehat{V}_{1:H} \in \mathcal{V}$ is as follows: for all $h \in [H]$, define $\widehat{\pi}_h(x) \in \arg\max_{a \in \mathcal{A}} \mathcal{P}_h[\widehat{V}_{h+1}](x, a)$. In reality, we do not have access to $\mathcal{P}_h[\widehat{V}_{h+1}](x, a)$ directly, so the full version of RVFS (Algorithm 5) estimates this quantity on the fly using the local simulator using the following scheme (Algorithm 7 in Appendix F): Given a state $x$, for each $a$, we sample $K$ rewards $\boldsymbol{r}_h \sim R_h(x, a)$ and next-state transitions $\boldsymbol{x}_{h+1} \sim T_h(\cdot \mid x, a)$, then approximate $\mathcal{P}_h[\widehat{V}_{h+1}](x, a)$ by the empirical mean. We remark that the use of these Bellman backup policies is actually crucial in the analysis for RVFS; even if we were to work with estimated state-action value functions $\widehat{Q}_{1:H}$ instead, our analysis would require executing the Bellman backup policies $\widehat{\pi}_h(x) \in \arg\max_{a \in \mathcal{A}} \mathcal{T}_h[\widehat{Q}_{h+1}](x, a)$ (instead of naively using $\widehat{\pi}_h(x) \in \arg\max_{a \in \mathcal{A}} \widehat{Q}_h(x, a)$).

**Invoking the algorithm.** The base invocation of RVFS takes the form

$$\widehat{V}_{1:H} \leftarrow \mathsf{RVFS}_0\big(\widehat{V}_{1:H} = \mathsf{arbitrary}, \widehat{\mathcal{V}}_{1:H} = \{\mathcal{V}_h\}_{h=1}^H, \mathcal{C}_{0:H} = \{\varnothing\}_{h=0}^H, ; \mathcal{V}, \varepsilon, \delta\).$$

Whenever this call returns, the greedy policy induced by $\widehat{V}_{1:H}$ is guaranteed to be near-optimal. Naively, the approximate Bellman backup policy induced by $\widehat{V}_{1:H}$ (described above) is non-executable, and must be computed by invoking the local simulator. To provide an end-to-end guarantee to learn an executable policy, we give an outer-level algorithm, RVFS.bc (Algorithm 6, deferred to Appendix F for space), which invokes $\mathsf{RVFS}_0$, then extracts an executable policy from $\widehat{V}_{1:H}$ using behavior cloning. Subsequent recursive calls to RVFS take the form $(\widehat{V}_{h:H}, \widehat{\mathcal{V}}_{h:H}, \mathcal{C}_{h:H}) \leftarrow \mathsf{RVFS}_h(\widehat{V}_{h+1:H}, \widehat{\mathcal{V}}_{h+1:H}, \mathcal{C}_{h:H}; \mathcal{V}, \varepsilon, \delta)$. The arguments here are: Importantly, the confidence sets $\widehat{\mathcal{V}}_{h+1:H}$ do not need to be explicitly maintained, and can be used implicitly whenever a *regression oracle* for the value function class is available (discussed below).

**Remark 4.1** (Oracle-efficiency). *RVFS is computationally efficient in the sense that it reduces to convex optimization over the value function class $\mathcal{V}$. In particular, the only computationally intensive steps in the algorithm are (i) the regression step in Line 16, and (ii) the test in Line 8 involving the confidence set $\widehat{\mathcal{V}}_\ell$. For the latter, we do not explicitly need to maintain $\widehat{\mathcal{V}}_\ell$, as the optimization problem over this set in Line 8 (for the full version of RVFS in Algorithm 5) reduces to solving $\arg\max_{V \in \mathcal{V}}\big\{\pm \sum_{i=1}^n V(\widetilde{x}^{(i)}) \mid \sum_{i=1}^n (V(x^{(i)}) - y^{(i)})^2 \le \beta^2\big\}$ for a dataset $\{(x^{(i)}, \widetilde{x}^{(i)}, y^{(i)})\}_{i=1}^n$. This is convex optimization problem in function space, and in particular can be implemented in a provably efficient fashion whenever $\mathcal{V}$ is linearly parameterized. We expect that the problem can also be reduced to a square loss regression by adapting the techniques in Krishnamurthy et al. [37], Foster et al. [23], but we do not pursue this here.*

## 4.3 Main Result

We present the main guarantee for RVFS under the function approximation assumptions in Section 4.1.

**Theorem 4.1** (Main guarantee for RVFS). *Let $\varepsilon, \delta \in (0, 1)$ be given, and suppose that Assumption 4.1 (pushforward coverability) holds with $C_{\mathsf{push}} > 0$. Further, suppose that one the following holds:*

- **Setup I:** *Assumptions 4.2 and 4.3 ($V^\star/\pi^\star$-realizability) and Assumption 4.4 ($\Delta$-gap) hold, and $\varepsilon \le 6H \cdot \Delta$.*

- **Setup II:** *Assumption 4.5 ($V^\pi$-realizability) and Assumption 4.6 ($\pi$-realizability) hold.*

*Then, $\mathsf{RVFS.bc}(\Pi, \mathcal{V}, \varepsilon, \delta)$ (Algorithm 6) returns a policy $\widehat{\pi}_{1:H}$ such that $J(\pi^\star) - J(\widehat{\pi}_{1:H}) \le 2\varepsilon$ with probability at least $1 - \delta$, and has total sample complexity bounded by*

$$\widetilde{O}\big(C_{\mathsf{push}}^8 H^{23} A \cdot \varepsilon^{-13}\big).$$

*Furthermore, the algorithm makes at most $\mathrm{poly}(C_{\mathsf{push}}, H, A, \varepsilon^{-1})$ calls to the convex optimization oracle over value function space described in Remark 4.1.*

Theorem 4.1 shows for the first time that sample- and computationally-efficient RL with local simulator access is possible under pushforward coverability. In particular, RVFS is the first computationally efficient algorithm for RL with local simulator access that supports nonlinear function approximation. The assumptions in Theorem 4.1, while stronger than those in Section 3, are not known to enable sample-efficient RL without simulator access. Nonetheless, understanding whether RVFS can be strengthened to support general coverability or weaker function approximation is an important open problem. See Appendix H.1 for an overview of the analysis; we remark (Appendix I.1) that the result is actually proven under slightly weaker assumptions than those in **Setup I/Setup II**.

**Connection to empirical algorithms.** RVFS bears some similarity to Monte-Carlo Tree Search (MCTS) [13, 35] and AlphaZero [52], which perform planning with local simulator. Informally, MCTS can be viewed as a form of breadth-first search over the state space (where each node represents a state at a given layer), and AlphaZero is a particular instantiation of a MCTS that leverages $V$–value function approximation to allow for generalization across states. Compared to RVFS, MCTS and AlphaZero perform exploration via simple bandit-style heuristics, and are not explicitly designed to handle *distribution shifts* that arise in settings where actions have long-term downstream effects. What is more, MCTS requires finite states to iterate over all possible child nodes of each state, making it inapplicable in environments with continuous states. RVFS may be viewed as a provable counterpart

that can handle continuous states and uses function approximation to address distribution shift in a principled fashion (in particular, through the use of confidence sets and the test in Line 14).[5]

**Applying RVFS to Exogenous Block MDPs.** ExBMDPs satisfy coverability (Assumption 3.2), but do not satisfy the pushforward coverability assumption (Assumption 4.1) in general. However, it turns out that ExBMDPs *do* satisfy pushforward coverability when the exogenous noise process is weakly correlated across time, a new statistical assumption we refer to the *weak correlation condition*. In Appendix B (Theorem B.1), we give a variant of RVFS for ExBMDPs that succeeds under (i) weak correlation, and (ii) decoder realizability, sidestepping the need for the $\Delta$-gap or $V^\pi$-realizability.

## 5 Discussion and Future Work

In this paper, we demonstrated that resets can substantially expand the range of reinforcement learning (RL) settings that are tractable, both statistically and computationally. Our practical algorithm, RVFS, provides a principled counterpart to MCTS by supporting continuous state spaces and offering provable guarantees, setting it apart from traditional MCTS. Statistically, our results extend to MDPs with a finite Sequential Estimation Coefficient (SEC) [63], capturing a broader class of MDPs beyond those with finite coverability—encompassing low Bellman Eluder MDPs [32] and MDPs with finite bilinear rank [18]. Although not formally developed here, it is possible to generalize push-forward coverability and our analysis to encompass a range of linear function approximation settings [57, 40, 3, 59, 65, 66], thereby recovering known positive results under local access in these settings.

While our focus has been on theoretical contributions—analyzing the sample and computational complexity of RL with access to a local simulator—this work also raises promising empirical questions. We are particularly interested in exploring these questions in future work, aiming to bridge these theoretical advances with empirical validation in practical RL settings.

---

[5]We note in passing that in the context of tree search, the pushforward coverability assumption (Assumption 4.1) may be viewed as the stochastic analogue of branching factor.

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

# Contents of Appendix

# A    Additional Related Work

**Local simulators: Theoretical research.**    RL with local simulators has received extensive interest in the context of linear function approximation. Most notably, Weisz et al. [57] show that reinforcement learning with linear $V^\star$ is tractable with local simulator access, and Li et al. [40] show that RL with linear $Q^\star$ and a state-action gap is tractable; online RL is known to be intractable under the same assumptions [57, 56]. Amortila et al. [3] show that the gap assumption can be removed if a small number of expert queries are available. Also of note are the works of Yin et al. [65], Weisz et al. [59], which give computationally efficient algorithms under linear $Q^\pi$-realizability for all $\pi$; this setting is known to be tractable in the online RL model [60], but computationally efficient algorithms are currently only known for RLLS.

*Global simulators*—in which the agent can query arbitrary state-action pairs and observe next state transitions—have also received theoretical investigation, but like local simulators, results are largely restricted to tabular reinforcement learning and linear models [34, 33, 50, 17, 64, 39].

**Local simulators: Empirical research.**    The Go-Explore algorithm [19, 20] uses local simulator access to achieve state-of-the-art performance for the Atari games Montezuma's Revenge and Pitfall—both notoriously difficult games that require systematic exploration. To the best of our knowledge, the performance of Go-Explore on these tasks has yet to be matched by online reinforcement learning; the performing agents [7, 27] are roughly a factor of four worse in terms of cumulative reward. Interestingly, like RVFS, Go-Explore makes use of core sets of informative state-action pairs to guide exploration. However, Go-Explore uses an ad-hoc, domain specific approach to designing the core set, and does not use function approximation to drive exploration.

Recent work of Yin et al. [66] provides an empirical framework for online RL with local planning that can take advantage of deep neural function approximation, and is inspired by the theoretical works in Weisz et al. [57], Li et al. [40], Yin et al. [65], Weisz et al. [59]. This approach does not have provable guarantees, but achieves super-human performance at Montezuma's Revenge.

Other notable empirical works that incorporate local simulator access, as highlighted by Yin et al. [66], include Schulman et al. [49], Salimans and Chen [47], Tavakoli et al. [54].

**Planning.**    RL with local simulator access is a convenient abstraction for the problem of *planning*: Given a known (e.g., learned) model, compute an optimal policy. Planning with a learned model is an important task in theory [25, 41] and practice (e.g., MuZero [48]). Since the model is known, computing an optimal policy is a purely computational problem, not a statistical problem. Nonetheless, for planning problems in large state spaces, where enumerating over all states is undesirable, algorithms for online RL with local simulator access can be directly applied, treating the model as if it were the environment the agent is interacting with. Here, any computationally efficient RLLS algorithm immediately yields an efficient algorithm for planning.

Empirically, Monte-Carlo Tree Search [13, 35] is a successful paradigm for planning, acting as a key component in AlphaGo [51] and AlphaZero [52].[6] Viewed as a planning algorithm, a potential advantage of RVFS is that it is well suited to stochastic environments, and provides a principled way to use estimated (neural) value function estimates to guide exploration.

**Coverability.**    Xie et al. [63] introduced coverability as a structural parameter for online reinforcement leanring, inspired by connections between online and offline RL. Existing guarantees for the online RL framework based on coverability require either Bellman completeness [63], model-based realizability [5], or weight function realizability [4, 5]), and it is not currently known whether value function realizability is sufficient in this framework.

**Exogenous Block MDPs.**    Our results in Section 3.3 (Corollary 3.1) show that general Exogenous Block MDPs are learnable with local simulator access. Prior work, on learning EXBMDPs in the online RL model requires additional assumptions:

- *Deterministic ExBMDP* [22]. In this setting, the latent transition distribution $T^{\mathrm{endo}}$ is assumed to be deterministic. In this case, it suffices to learn *open-loop* policies (i.e., policies that play a deterministic sequence of actions). This avoids compounding errors due to

---

[6]Compare to our work, a small difference is that these works are not concerned with producing executable policies, c.f. Definition 2.1.

learning imperfect decoders that depend on the exogenous noise, making this setting much less challenging than the general ExBMDP setting.

- *Factored ExMDP* [21]. This is an ExBMDP setting with a restrictive structure in which the observation is a $d$-dimensional vector and the latent state is a $k$-dimensional subset of the observed coordinates. This structure prevents the setting from subsuming the basic (non-exogenous) Block MDP framework, and makes it possible to learn decoders that act only on the endogenous state, preventing compounding errors.

- *Bellman completeness*. Xie et al. [63] observed that ExBMDPs admit low coverability, but their algorithm requires Bellman completeness, which is not satisfied by ExBMDPs (see Efroni et al. [22], Islam et al. [30]).

# Part I

# Additional Results

This section of the appendix contains additional results omitted from the main body due to space constraints.

## B    Applying RVFS to Exogenous Block MDPs

We now apply RVFS to the Exogenous Block MDP (ExBMDP) model introduced in Section 3.3. ExBMDPs satisfy coverability (Assumption 3.2), but do not satisfy the pushforward coverability assumption (Assumption 4.1) required by RVFS in general. However, it turns out that ExBMDPs *do* satisfy pushforward coverability when the exogenous noise process is weakly correlated across time; we refer to this new statistical assumption as the *weak correlation condition*.

**Assumption B.1** (Weak correlation condition). *For the underlying ExBMDP $\mathcal{M}$, there is a constant $C_{\mathsf{exo}} \geq 1$ such that for all $h \in [H-1]$ and $(\xi, \xi') \in \Xi_{h-1} \times \Xi_h$, we have[7]*

$$\mathbb{P}[\boldsymbol{\xi}_h = \xi, \boldsymbol{\xi}_{h+1} = \xi'] \leq C_{\mathsf{exo}} \cdot \mathbb{P}[\boldsymbol{\xi}_h = \xi] \cdot \mathbb{P}[\boldsymbol{\xi}_{h+1} = \xi'].$$

The weak correlation property asserts that the joint law for the exogenous noise variables $\boldsymbol{\xi}_h$ and $\boldsymbol{\xi}_{h+1}$ is at most a multiplicative factor $C_{\mathsf{exo}} \geq 1$ larger than the corresponding product distribution obtained by sampling $\boldsymbol{\xi}_h$ and $\boldsymbol{\xi}_{h+1}$ independently from their marginals. This setting strictly generalizes the (non-exogenous) Block MDP model [36, 15, 43, 68, 42], by allowing for arbitrary stochastic dynamics for the endogenous state and an arbitrary emission process, but requires that temporal correlations in the exogenous noise decay over time.

We show that under Assumption B.1, pushforward coverability is satisfied with $C_{\mathsf{push}} \leq C_{\mathsf{exo}} \cdot SA$ (Lemma K.3 in Appendix K.1). In addition, $V^\star$-realizability is implied by decoder realizability (Lemma D.1). Thus, by applying Theorem 4.1 (**Setup I**), we conclude that RVFS efficiently learns a near-optimal policy for any weakly correlated ExBMDP for which the optimal value function has $\Delta$-gap.

**An improved algorithm for ExBMDPs: RVFS^{exo}.**    At first glance, removing the gap assumption for RVFS in ExBMDPs seems difficult: The $V^\pi$-realizability assumption required to invoke Theorem 4.1 (**Setup II**) is not satisfied by ExBMDPs, as decoder realizability only implies $V^\pi$ realizability for *endogenous* policies $\pi$.[8] In spite of this, we now show that with a slight modification, RVFS can efficiently learn any weakly correlated ExBMDP under decoder realizability alone (without gap or $V^\pi$-realizability).

Our new variant of RVFS, RVFS^{exo}, is presented in Algorithm 8 (deferred to Appendix F for space). The algorithm is almost identical to RVFS (Algorithm 5), with the main difference being that we use an additional *randomized rounding* step to compute the policies $\widehat{\pi}_{1:H}$ from the learned value functions $\widehat{V}_{1:H}$. In particular, instead of directly defining the policies $\widehat{\pi}_{1:H}$ based on the bellman backups $\mathcal{P}_h[\widehat{V}_{h+1}]$ as in Eq. (14), RVFS^{exo} targets a "rounded" version of the backup given by

$$\varepsilon \cdot \lceil \mathcal{P}_h[\widehat{V}_{h+1}](x, a)/\varepsilon + \boldsymbol{\zeta}_h \rceil, \tag{3}$$

where $\varepsilon \in (0, 1)$ is a rounding parameter and $\boldsymbol{\zeta}_1, \ldots, \boldsymbol{\zeta}_H$ are i.i.d. random variables sampled uniformly at random from the interval $[0, 1/2]$ (at the beginning of the algorithm's execution). Concretely, RVFS^{exo} estimates the bellman backup $\mathcal{P}_h[\widehat{V}_{h+1}](x, a)$ in Eq. (3) using the local simulator (as in Eq. (14) of Algorithm 5), and defines its policies via

$$\widehat{\pi}_h(\cdot) \in \arg\max_{a \in \mathcal{A}} \lceil \mathcal{P}_h[\widehat{V}_{h+1}](\cdot, a)/\varepsilon + \boldsymbol{\zeta}_h \rceil. \tag{4}$$

---

[7]Throughout this paper, when considering the law for the exogenous variables $\boldsymbol{\xi}_1, \ldots, \boldsymbol{\xi}_H$, we write $\mathbb{P}[\cdot]$ instead of $\mathbb{P}^\pi[\cdot]$ to emphasize that the law is independent of the agent's policy.

[8]We say that a policy $\pi$ is *endogenous* if it does not depend on exogenous noise, in the sense that $\pi(\boldsymbol{x}_h)$ is a measurable function of $\phi^\star(\boldsymbol{x}_h)$.

This rounding scheme, which quantizes the Bellman backup into $\varepsilon^{-1}$ bins with a random offset, is designed to emulate certain properties implied by the $\Delta$-gap assumption (Assumption 4.4). Specifically, we show that with constant probability over the draw of $\zeta_{1:H}$, the policy $\widehat{\pi}$ in (4) "snaps" on to an *endogenous* policy $\pi$. This means that for $\mathsf{RVFS}^{\mathsf{exo}}$ to succeed (with constant probability), it suffices to pass it a class $\mathcal{V}$ that realizes the value functions $(V_h^\pi)$ for endogenous policies $\pi \in \Pi_{\mathsf{S}}$. Fortunately, such a function class can be constructed explicitly under decoder realizability (Assumption 3.3).

**Lemma B.1** ([21]). *For the ExBMDP setting, under Assumption 3.3, the function class $\mathcal{V}_h \coloneqq \{x \mapsto f(\phi(x)) : f \in [0, H]^S, \phi \in \Phi\}$ is such that $V_h^\pi \in \mathcal{V}_h$ for all endogenous policies $\pi$. Furthermore, the policy class $\Pi_h \coloneqq \{\pi(\cdot) \in \arg\max_{a \in \mathcal{A}} f(\phi(\cdot), a) : f \in [0, H]^{S \times A}, \phi \in \Phi\}$ contains all endogenous policies.*

A small technical challenge with the scheme above is that it is only guaranteed to succeed with constant probability over the draw of the rounding parameters $\zeta_1, \ldots, \zeta_H$. To address this, we provide an outer-level algorithm, $\mathsf{RVFS}^{\mathsf{exo}}.\mathsf{bc}$ (Algorithm 9, deferred to Appendix F for space), which performs confidence boosting by invoking $\mathsf{RVFS}^{\mathsf{exo}}$ multiple times independently, and extracts a high-quality executable policy using behavior cloning.

**Main result.** We now state the main guarantee for $\mathsf{RVFS}^{\mathsf{exo}}$ (the proof is in Appendix K).

**Theorem B.1** (Main guarantee of $\mathsf{RVFS}^{\mathsf{exo}}$ for EXBMDPs). *Consider the ExBMDP setting. Suppose the decoder class $\Phi$ satisfies Assumption 3.3, and that Assumption B.1 holds with $C_{\mathsf{exo}} > 0$. Let $\varepsilon, \delta \in (0, 1)$ be given, and let $\mathcal{V}_h$ and $\Pi_h$ be as in Lemma B.1. Then $\mathsf{RVFS}^{\mathsf{exo}}.\mathsf{bc}(\Pi, \mathcal{V}_{1:H}, \varepsilon, \zeta_{1:H}, \delta)$ (Algorithm 9) produces a policy $\widehat{\pi}_{1:H}$ such that $J(\pi^\star) - J(\widehat{\pi}_{1:H}) \leq \varepsilon$, and has total sample complexity*

$$\widetilde{O}\left(C_{\mathsf{exo}}^8 S^8 H^{36} A^9 \cdot \varepsilon^{-26}\right).$$

This result shows for the first time that sample- and computationally-efficient learning is possible for ExBMDPs beyond deterministic or factored settings [22, 21].

We mention in passing that our use of randomized rounding to emulate certain consequences of the $\Delta$-gap assumption leverages the fact that ExBMDPs have a finite number of (endogenous) latent states. It is unclear if this technique can be used when the (latent) state space is large or infinite.

**Algorithm 3** $\text{RVFS}_h^{\text{exo}}$: Recursive Value Function Search for Exogenous Block MDPs

---

1: **parameters:** Value function class $\mathcal{V}$, suboptimality $\varepsilon \in (0,1)$, seeds $\zeta_{1:H} \in (0,1)$, confidence $\delta \in (0,1)$.

2: **input:** Level $h \in [0 \ldotp\ldotp H]$, value function estimates $\widehat{V}_{h+1:H}$, confidence sets $\widehat{\mathcal{V}}_{h+1:H}$, state-action collections $\mathcal{C}_{h:H}$, and buffers $\mathcal{B}_{h:H}$, and counters $t_{h:H}$.

    `/* Initialize parameters. */`

3: Set $M \leftarrow \lceil 8\varepsilon^{-2}C_{\text{exo}}SAH \rceil$.

4: Set $N_{\text{test}} \leftarrow 2^8 M^2 H \varepsilon^{-2} \log(8M^6 H^8 \varepsilon^{-2}\delta^{-1})$, $N_{\text{reg}} \leftarrow 2^8 M^2 \varepsilon^{-2} \log(8|\mathcal{V}|HM^2\delta^{-1})$.

5: Set $N_{\text{est}}(k) \leftarrow 2N_{\text{reg}}^2 \log(8AN_{\text{reg}}Hk^3/\delta)$ and $\delta' \leftarrow \delta/(4M^7 N_{\text{test}}^2 H^8 |\mathcal{V}|)$.

6: Set $\varepsilon_{\text{reg}}^2 \leftarrow \dfrac{9MH^2 \log(8M^2 H |\mathcal{V}|/\delta)}{N_{\text{reg}}} + \dfrac{34MH^3 \log(8M^6 N_{\text{test}}^2 H^8/\delta)}{N_{\text{test}}}$.

7: Set $\beta(t) \leftarrow \sqrt{\log_{1/\delta'}(8MA|\mathcal{V}|t^2/\delta)}$.

    `/* Test the fit for the estimated value functions `$\widehat{V}_{h+1:H}$` at future layers. */`

8: **for** $(x_{h-1}, a_{h-1}) \in \mathcal{C}_h$ **do**

9:     **for** layer $\ell = H, \ldots, h+1$ **do**

10:         **for** $n = 1, \ldots, N_{\text{test}}$ **do**

11:             Draw $\boldsymbol{x}_h \sim T_{h-1}(\cdot \mid x_{h-1}, a_{h-1})$, then draw $\boldsymbol{x}_{\ell-1}$ by rolling out with $\widehat{\pi}_{h+1:H}$, where

$$\forall \tau \in [H], \ \ \widehat{\pi}_\tau(\cdot) \in \arg\max_{a \in \mathcal{A}} \left[ \widehat{\boldsymbol{\mathcal{P}}}_{\tau,\varepsilon^2,\delta'}[\widehat{V}_{\tau+1}](\cdot, a) \cdot \varepsilon^{-1} + \zeta_\tau \right], \quad \text{with} \quad \widehat{V}_{H+1} \equiv 0.$$

12:             **for** $a_{\ell-1} \in \mathcal{A}$ **do**

13:                 Update $t_\ell \leftarrow t_\ell + 1$.

                `/* Test fit; if test fails, re-fit value functions `$\widehat{V}_{h+1:\ell}$` up to layer `$\ell$`. */`

14:                 **if** $\sup_{f \in \widehat{\mathcal{V}}_\ell} |(\widehat{\boldsymbol{\mathcal{P}}}_{\ell-1,\varepsilon^2,\delta'}[\widehat{V}_\ell] - \widehat{\boldsymbol{\mathcal{P}}}_{\ell-1,\varepsilon^2,\delta'}[f_\ell])(\boldsymbol{x}_{\ell-1}, a_{\ell-1})| > \varepsilon^2 + \varepsilon^2 \cdot \beta(t_\ell)$ **then**

15:                     $\mathcal{C}_\ell \leftarrow \mathcal{C}_\ell \cup \{(\boldsymbol{x}_{\ell-1}, a_{\ell-1})\}$ and $\mathcal{B}_\ell \leftarrow \mathcal{B}_\ell \cup \{(\boldsymbol{x}_{\ell-1}, a_{\ell-1}, \widehat{V}_\ell, \widehat{\mathcal{V}}_\ell, t_\ell)\}$.

16:                     **for** $\tau = \ell, \ldots, h+1$ **do**

17:                         $(\widehat{V}_{\tau:H}, \widehat{\mathcal{V}}_{\tau:H}, \mathcal{C}_{\tau:H}, \mathcal{B}_{\tau:H}, t_{\tau:H}) \leftarrow \text{RVFS}_\tau^{\text{exo}}(\widehat{V}_{\tau+1:H}, \widehat{\mathcal{V}}_{\tau+1:H}, \mathcal{C}_{\tau:H}, \mathcal{B}_{\tau:H}, t_{\tau:H}; \mathcal{V}, \varepsilon, \zeta_{1:H}, \delta)$.

18:                     **go to line 8**.

19: **if** $h = 0$ **then return** $(\widehat{V}_{1:H}, \cdot, \cdot, \cdot, \cdot)$.

    `/* Re-fit `$\widehat{V}_h$` and build a new confidence set. */`

20: **for** $(x_{h-1}, a_{h-1}) \in \mathcal{C}_h$ **do**

21:     Set $\mathcal{D}_h(x_{h-1}, a_{h-1}) \leftarrow \varnothing$.

22:     **for** $i = 1, \ldots, N_{\text{reg}}$ **do**

23:         Sample $\boldsymbol{x}_h \sim T_{h-1}(\cdot \mid x_{h-1}, a_{h-1})$.

24:         For each $a \in \mathcal{A}$, let $\widehat{V}_h(\boldsymbol{x}_h)$ be a Monte-Carlo estimate for $\mathbb{E}^{\widehat{\pi}_{h:H}}[\sum_{\ell=h}^{H} \boldsymbol{r}_\ell \mid \boldsymbol{x}_h]$ computed by collecting $N_{\text{est}}(|\mathcal{C}_h|)$ trajectories starting from $\boldsymbol{x}_h$ and rolling out with $\widehat{\pi}_{h:H}$.

25:         Update $\mathcal{D}(x_{h-1}, a_{h-1}) \leftarrow \mathcal{D}(x_{h-1}, a_{h-1}) \cup \{(\boldsymbol{x}_h, \widehat{V}_h(\boldsymbol{x}_h))\}$.

26: Let $\widehat{V}_h := \arg\min_{f \in \mathcal{V}_h} \sum_{(x_{h-1}, a_{h-1}) \in \mathcal{C}_h} \sum_{(x_h, v_h) \in \mathcal{D}_h(x_{h-1}, a_{h-1})} (f(x_h) - v_h)^2$.

27: Compute value function confidence set

$$\widehat{\mathcal{V}}_h := \left\{ f \in \mathcal{V}_h \ \middle| \ \sum_{(x_{h-1}, a_{h-1}) \in \mathcal{C}_h} \frac{1}{N_{\text{reg}}} \sum_{(x_h, \text{-}) \in \mathcal{D}_h(x_{h-1}, a_{h-1})} \left(\widehat{V}_h(x_h) - f(x_h)\right)^2 \leq \varepsilon_{\text{reg}}^2 \right\}. \quad (5)$$

28: **return** $(\widehat{V}_{h:H}, \widehat{\mathcal{V}}_{h:H}, \mathcal{C}_{h:H}, \mathcal{B}_{h:H}, t_{h:H})$.

---

**Algorithm 4** $\mathsf{RVFS^{exo}.bc}$: Learn an executable policy with $\mathsf{RVFS^{exo}}$ via imitation learning.

1: **input:** Decoder class $\Phi$, suboptimality $\varepsilon \in (0, 1)$, confidence $\delta \in (0, 1)$.
    /* Set parameters for RVFS and define the value function and policy classes. */
2: Set $\varepsilon_{\mathsf{RVFS}} \leftarrow \varepsilon H^{-1}/48$.
3: Set $\mathcal{V} = \mathcal{V}_{1:H}$, where $\mathcal{V}_h = \{x \mapsto f(\phi(x)) : f \in [0, H]^S, \phi \in \Phi\}$, $\forall h \in [H]$.
4: Set $\Pi = \Pi_{1:H}$, where $\Pi_h = \{\pi(\cdot) \in \arg\max_{a \in \mathcal{A}} f(\phi(\cdot), a) : f \in [0, H]^{S \times A}, \phi \in \Phi\}$, $\forall h \in [H]$.
    /* Set parameters for BehaviorCloning. */
5: Set $N_{\mathsf{bc}} \leftarrow 8H^2 \log(4H|\Pi|/\delta)/\varepsilon$, $N_{\mathsf{boost}} \leftarrow \log(1/\delta)/\log(24SAH\varepsilon)$, $N_{\mathsf{eval}} \leftarrow 16^2\varepsilon^{-2}\log(2N_{\mathsf{boost}}/\delta)$.
6: Set $M \leftarrow \lceil 8\varepsilon_{\mathsf{RVFS}}^{-1}SAC_{\mathsf{cov}}H \rceil$, $N_{\mathsf{test}} \leftarrow 2^8 M^2 H\varepsilon_{\mathsf{RVFS}}^{-1}\log(80M^6H^8N_{\mathsf{boost}}\varepsilon_{\mathsf{RVFS}}^{-2}\delta^{-1})$, and $\delta' = \frac{\delta}{40M^7N^2H^8|\mathcal{V}|N_{\mathsf{boost}}}$.
7: Set $N_{\mathsf{reg}} \leftarrow 2^8 M^2\varepsilon_{\mathsf{RVFS}}^{-1}\log(80|\Phi|^2HM^2N_{\mathsf{boost}}\delta^{-1})$.
8: Set $\widehat{V}_{1:H} \leftarrow$ arbitrary, $\widehat{\mathcal{V}}_{1:H} \leftarrow \mathcal{V}$, $\mathcal{C}_{0:H} \leftarrow \varnothing$, $\mathcal{B}_{0:H} \leftarrow \varnothing$, $i_{\mathsf{opt}} = 1$, and $J_{\max} = 0$.
    /* Repeatedly invoke RVFS$^{\mathsf{exo}}$ and extract policy with BehaviorCloning to boost confidence. */
9: **for** $i = 1, \ldots, N_{\mathsf{boost}}$ **do**
       /* Invoke RVFS$^{\mathsf{exo}}$. */
10:     $(\widehat{V}_{1:H}^{(i)}, \cdot, \cdot, \cdot) \leftarrow \mathsf{RVFS_0^{exo}}(\widehat{V}_{1:H}, \widehat{\mathcal{V}}_{1:H}, \mathcal{C}_{0:H}, \mathcal{B}_{0:H}; \mathcal{V}, N_{\mathsf{reg}}, N_{\mathsf{test}}, \varepsilon_{\mathsf{RVFS}}, \delta/(10N_{\mathsf{boost}}))$.
       /* Imitation learning with BehaviorCloning. */
11:     Define $\widehat{\boldsymbol{\pi}}_h^{\mathsf{RVFS}}(\cdot) \in \arg\max_{a \in \mathcal{A}} \widehat{\mathcal{P}}_{h, \varepsilon_{\mathsf{RVFS}}, \delta'}[\widehat{V}_{h+1}^{(i)}](\cdot, a)$.
12:     Compute $\widehat{\pi}_{1:H}^{(i)} \leftarrow \mathsf{BehaviorCloning}(\Pi, \varepsilon, \widehat{\pi}_{1:H}^{\mathsf{RVFS}}, \delta/(2N_{\mathsf{boost}}))$.
       /* Evaluate current policy. */
13:     $v = 0$.
14:     **for** $= 1, \ldots, N_{\mathsf{eval}}$ **do**
15:         Sample trajectory $(\boldsymbol{x}_1, \boldsymbol{a}_1, \boldsymbol{r}_1, \ldots, \boldsymbol{x}_H, \boldsymbol{a}_H, \boldsymbol{r}_H)$ by executing $\widehat{\pi}_{1:H}^{(i)}$.
16:         Set $v \leftarrow v + \sum_{h=1}^H \boldsymbol{r}_h$.
17:     Set $\widehat{J}(\widehat{\pi}_{1:H}^{(i)}) \leftarrow v/N_{\mathsf{eval}}$.
18:     **if** $\widehat{J}(\widehat{\pi}_{1:H}^{(i)}) > J_{\max}$ **then**
19:         Set $i_{\mathsf{opt}} = i$.
20:         Set $J_{\max} = \widehat{J}(\widehat{\pi}_{1:H}^{(i)})$.
21: **return:** $\widehat{\pi}_{1:H} = \widehat{\pi}_{1:H}^{(i_{\mathsf{opt}})}$.

## C   Helper Lemmas

This section of the appendix contains supporting lemmas used within the proofs of our main results.

### C.1   Concentration and Probability

**Lemma C.1.** *Let $\delta \in (0, 1)$ and $H \geq 1$ be given. If a sequence of events $\mathcal{E}_1, \ldots, \mathcal{E}_H$ satisfies $\mathbb{P}[\mathcal{E}_h \mid \mathcal{E}_1, \ldots, \mathcal{E}_{h-1}] \geq 1 - \delta/H$ for all $h \in [H]$, then*

$$\mathbb{P}[\mathcal{E}_{1:H}] \geq 1 - \delta.$$

**Proof of Lemma C.1.** By the chain rule, we have

$$\mathbb{P}[\mathcal{E}_{1:H}] = \prod_{h \in [H]} \mathbb{P}[\mathcal{E}_h \mid \mathcal{E}_1, \ldots, \mathcal{E}_{h-1}] \geq \prod_{h \in [H]} (1 - \delta/H) = (1 - \delta/H)^H \geq 1 - \delta.$$

$\square$

We make use of the following version of Freedman's inequality, due to Agarwal et al. [1, Lemma 9]:

**Lemma C.2.** *Let $R > 0$ be given and let $\boldsymbol{w}_1, \ldots \boldsymbol{w}_n$ be a sequence of real-valued random variables adapted to filtration $\mathcal{H}_1, \cdots, \mathcal{H}_n$. Assume that for all $t \in [n]$, $\boldsymbol{w}_i \leq R$ and $\mathbb{E}[\boldsymbol{w}_i \mid \mathcal{H}_{i-1}] = 0$. Define $\boldsymbol{S}_n := \sum_{t=1}^n \boldsymbol{w}_i$ and $V_n := \sum_{t=1}^n \mathbb{E}[\boldsymbol{w}_i^2 \mid \mathcal{H}_{i-1}]$. Then, for any $\delta \in (0, 1)$ and $\lambda \in [0, 1/R]$, with probability at least $1 - \delta$,*

$$\boldsymbol{S}_n \leq \lambda V_n + \log(1/\delta)/\lambda.$$

We will also use the following lemma, which is a standard consequence of Freedman's inequality.

**Lemma C.3** (e.g., Foster et al. [25])**.** *Let $(\boldsymbol{w}_t)_{t \le T}$ be a sequence of random variables adapted to a filtration $(\mathcal{H}_t)_{t \le T}$. If $0 \le \boldsymbol{w}_t \le R$ almost surely, then with probability at least $1 - \delta$,*

$$\sum_{t=1}^{T} \boldsymbol{w}_t \le \frac{3}{2} \sum_{t=1}^{T} \mathbb{E}_{t-1}[\boldsymbol{w}_t] + 4R \log(2\delta^{-1}),$$

*and*

$$\sum_{t=1}^{T} \mathbb{E}_{t-1}[\boldsymbol{w}_t] \le 2 \sum_{t=1}^{T} \boldsymbol{w}_t + 8R \log(2\delta^{-1}).$$

## C.2  Regression

Using Lemmas C.2 and C.3, we obtain the following concentration lemma, which will be used to prove guarantees for square loss regression within our algorithms.

**Lemma C.4.** *Let $B > 0$ and $n \in \mathbb{N}$ be given, and let $\mathcal{Y}$ be an abstract set. Further, let $\mathcal{Q} \subseteq \{g : \mathcal{Y} \to [0, B]\}$ be a finite function class and $\boldsymbol{y}_1, \ldots, \boldsymbol{y}_n$ be a sequence of random variables in $\mathcal{Y}$ adapted to filtration a $\mathcal{H}_1, \cdots, \mathcal{H}_n$. Then, for any $\delta \in (0, 1)$, with probability at least $1 - \delta$, we have*

$$\forall g \in \mathcal{Q}, \quad \frac{1}{2}\|g\|^2 - 2B^2 \log(2|\mathcal{Q}|/\delta) \le \|g\|_n^2 \le 2\|g\|^2 + 2B^2 \log(2|\mathcal{Q}|/\delta),$$

*where $\|g\|^2 := \sum_{i \in [n]} \mathbb{E}[g(\boldsymbol{y}_i)^2 \mid \mathcal{H}_{i-1}]$ and $\|g\|_n^2 := \sum_{i=1}^{n} g(\boldsymbol{y}_i)^2$.*

**Proof of Lemma C.4.** Fix $g \in \mathcal{Q}$. Applying Lemma C.2 with $\boldsymbol{w}_i = g(\boldsymbol{y}_i)^2 - \mathbb{E}[g(\boldsymbol{y}_i)^2 \mid \mathcal{H}_{i-1}]$, for all $i \in [n]$, and $(R, \lambda) = (B^2, 1/B^2)$, we get that with probability at least $1 - \delta/(2|\mathcal{Q}|)$:

$$\|g\|_n^2 - \|g\|^2 \le \lambda B^2 \|g\|^2 + \log(2|\mathcal{Q}|/\delta)/\lambda.$$

By substituting $\lambda = B^{-2}$ and rearranging, we get

$$\|g\|_n^2 \le 2\|g\|^2 + B^2 \log(2|\mathcal{Q}|/\delta). \tag{6}$$

Similarly, applying Lemma C.2 with $\boldsymbol{w}_i = \mathbb{E}[g(\boldsymbol{y}_i)^2 \mid \mathcal{H}_{i-1}] - g(\boldsymbol{y}_i)^2$, for all $i \in [n]$, and $(R, \lambda) = (B^2, 1/(2B^2))$, we get that with probability at least $1 - \delta/(2|\mathcal{Q}|)$:

$$\|g\|^2 - \|g\|_n^2 \le \lambda B^2 \|g\|^2 + \log(2|\mathcal{Q}|/\delta)/\lambda.$$

By substituting $\lambda = 2^{-1} B^{-2}$ and rearranging, we get

$$\|g\|_n^2 \ge \frac{1}{2}\|g\|^2 - 2B^2 \log(2|\mathcal{Q}|/\delta).$$

Combining this with (6) and the union bound, we get the desired result.  □

With this lemma, we now prove the following key result for square loss regression.

**Lemma C.5** (Generic regression guarantee)**.** *Let $B > 0$ and $n \in \mathbb{N}$ be given and $\mathcal{Y}$ be an abstract set. Further, let $\mathcal{F} \subseteq \{f : \mathcal{Y} \to [0, B]\}$ be a finite function class, and suppose that there is a function $f_\star \in \mathcal{F}$ and a sequence of random variables $(\boldsymbol{y}_1, \boldsymbol{x}_1), \ldots, (\boldsymbol{y}_n, \boldsymbol{x}_n) \in \mathcal{Y} \times \mathbb{R}$ such that for all $i \in [n]$:*

- *$\boldsymbol{x}_i = f_\star(\boldsymbol{y}_i) + \varepsilon_i + \boldsymbol{b}_i$;*
- *$|\boldsymbol{b}_i| \le \xi$;*
- *$\varepsilon_i \in [-B, B]$; and*
- *$\mathbb{E}[\varepsilon_i \mid \mathfrak{F}_i] = 0$, where $\mathfrak{F}_i := \sigma(\boldsymbol{y}_{1:i}, \varepsilon_{1:i-1}, \boldsymbol{x}_{1:i-1}, \boldsymbol{b}_{1:i-1})$.*

*Then, for $\widehat{f} \in \arg\min_{f \in \mathcal{F}} \sum_{i=1}^{n} (f(\boldsymbol{y}_i) - \boldsymbol{x}_i)^2$ and any $\delta \in (0, 1)$, with probability at least $1 - \delta/2$,*

$$\|\widehat{f} - f_\star\|_n^2 \le 4B^2 \log(2|\mathcal{F}|/\delta) + 4B \sum_{i=1}^{n} |\boldsymbol{b}_i|,$$

*where $\|\widehat{f} - f_\star\|_n^2 := \sum_{i=1}^{n} (\widehat{f}(\boldsymbol{y}_i) - f^\star(\boldsymbol{y}_i))^2$.*

**Proof of Lemma C.5.** Fix $\delta \in (0,1)$ and let $\widehat{L}_n(f) \coloneqq \sum_{i=1}^n (f(\boldsymbol{y}_i) - \boldsymbol{x}_i)^2$, for $f \in \mathcal{F}$, and note that since $\widehat{f} \in \arg\min_{f \in \mathcal{F}} \widehat{L}_n(f)$, we have

$$0 \geq \widehat{L}_n(\widehat{f}) - \widehat{L}_n(f_\star) = \nabla\widehat{L}_n(f_\star)[\widehat{f} - f_\star] + \|\widehat{f} - f_\star\|_n^2,$$

where $\nabla$ denotes directional derivative. Rearranging, we get that

$$\begin{aligned}
\|\widehat{f} - f_\star\|_n^2 &\leq -2\nabla\widehat{L}_n(f_\star)[\widehat{f} - f_\star] - \|\widehat{f} - f_\star\|_n^2, \\
&= 4\sum_{i=1}^n (\boldsymbol{x}_i - f_\star(\boldsymbol{y}_i))(\widehat{f}(\boldsymbol{y}_i) - f_\star(\boldsymbol{y}_i)) - \|\widehat{f} - f_\star\|_n^2, \\
&\leq 4\sum_{i=1}^n (\boldsymbol{\varepsilon}_i + \boldsymbol{b}_i)(\widehat{f}(\boldsymbol{y}_i) - f_\star(\boldsymbol{y}_i)) - \|\widehat{f} - f_\star\|_n^2, \\
&\leq \underbrace{4\sum_{i=1}^n \boldsymbol{\varepsilon}_i \cdot (\widehat{f}(\boldsymbol{y}_i) - f_\star(\boldsymbol{y}_i)) - \|\widehat{f} - f_\star\|_n^2}_{\text{I}} + \underbrace{4\sum_{i=1}^n \boldsymbol{b}_i \cdot (\widehat{f}(\boldsymbol{y}_i) - f_\star(\boldsymbol{y}_i))}_{\text{II}}. \quad (7)
\end{aligned}$$

*Bounding Term I.* To bound Term I, we apply Lemma C.2 with $\boldsymbol{w}_i = \boldsymbol{\varepsilon}_i \cdot (\widehat{f}(\boldsymbol{y}_i) - f_\star(\boldsymbol{y}_i))$, $R = B^2$, $\lambda = 1/(8B^2)$, and $\mathcal{H}_i = \mathfrak{F}_{i+1}^-$, and use

1. the union bound over $f \in \mathcal{F}$; and

2. the facts that $\mathbb{E}[\boldsymbol{y}_i \mid \mathfrak{F}_i^-] = \boldsymbol{y}_i$ and $\mathbb{E}[\boldsymbol{\varepsilon}_i \mid \mathfrak{F}_i^-] = 0$,

to get that with probability at least $1 - \delta/2$,

$$4\sum_{i=1}^n \boldsymbol{\varepsilon}_i \cdot (\widehat{f}(\boldsymbol{y}_i) - f_\star(\boldsymbol{y}_i)) \leq \|\widehat{f} - f_\star\|_n^2 + 4B^2\log(2|\mathcal{F}|/\delta).$$

By rearranging, we get that with probability at least $1 - \delta/2$,

$$\text{Term I} \leq 4B^2\log(2|\mathcal{F}|/\delta).$$

*Bounding Term II.* We now bound the second term in (7). For this, note that since $\|\widehat{f} - f_\star\|_\infty \leq B$, we have

$$\text{Term II} \leq 4B\sum_{i=1}^n |\boldsymbol{b}_i|.$$

This completes the proof. $\qquad\square$

## C.3  Reinforcement Learning

**Lemma C.6** (Performance Difference Lemma (e.g., Kakade [33]))**.** *For any two policies $\widehat{\pi}, \pi \in \Pi_{\mathsf{S}}$ and $t \in [H]$, we have*

$$\mathbb{E}^\pi\left[V_t^\pi(\boldsymbol{x}_t) - V_t^{\widehat{\pi}}(\boldsymbol{x}_t)\right] = \mathbb{E}^\pi\left[\sum_{h=t}^H Q_h^{\widehat{\pi}}(\boldsymbol{x}_h, \boldsymbol{\pi}_h(\boldsymbol{x}_h)) - Q_h^{\widehat{\pi}}(\boldsymbol{x}_h, \widehat{\boldsymbol{\pi}}_h(\boldsymbol{x}_h))\right].$$

*In particular, applying this for $t = 1$ gives*

$$J(\pi) - J(\widehat{\pi}) = \mathbb{E}^\pi\left[\sum_{h=1}^H Q_t^{\widehat{\pi}}(\boldsymbol{x}_h, \boldsymbol{\pi}_h(\boldsymbol{x}_h)) - Q_h^{\widehat{\pi}}(\boldsymbol{x}_h, \widehat{\boldsymbol{\pi}}_h(\boldsymbol{x}_h))\right].$$

**Lemma C.7** (Potential lemma [63])**.** *Fix $h \in [H]$. Suppose we have a sequence of functions $g^{(1)}, \ldots, g^{(T)} \in [0, B]$ and policies $\pi^{(1)}, \ldots, \pi^{(T)}$ such that*

$$\forall t \in [T], \quad \sum_{i<t} \mathbb{E}^{\pi^{(i)}}\left[(g^{(t)}(\boldsymbol{x}_h))^2\right] \leq \beta^2$$

*for some $\beta \geq 0$. Then under [Assumption 4.1](), we have*

$$\sum_{t=1}^{T} \mathbb{E}^{\pi^{(t)}}[g^{(t)}(\boldsymbol{x}_h)] \leq 2\sqrt{\beta^2 C_{\mathsf{push}} T \log(2T)} + 2BC_{\mathsf{push}},$$

*and consequently*

$$\min_{t \in [T]} \mathbb{E}^{\pi^{(t)}}[g^{(t)}(\boldsymbol{x}_h)] \leq 2\sqrt{\frac{\beta^2 C_{\mathsf{push}} \log(2T)}{T}} + \frac{2BC_{\mathsf{push}}}{T}.$$

**Proof of [Lemma C.8]().** See proof of [63, Theorem 1]. $\qquad\square$

The following result is a variant of the coverability-based potential argument given in Xie et al. [63].

**Lemma C.8** (Pushforward coverability potential lemma). *Fix $h \in [H]$. Suppose we have a sequence of functions $g^{(1)}, \ldots, g^{(T)} \in [0, B]$ and state-action pairs $(x^{(1)}, a^{(1)}), \ldots, (x^{(T)}, a^{(T)})$ such that*

$$\forall t \in [T], \quad \sum_{i < t} \mathbb{E}\big[(g^{(t)}(\boldsymbol{x}_h))^2 \mid \boldsymbol{x}_{h-1} = x^{(i)}, \boldsymbol{a}_{h-1} = a^{(i)}\big] \leq \beta^2$$

*for some $\beta \geq 0$. Then under [Assumption 4.1](), we have*

$$\sum_{t=1}^{T} \mathbb{E}\big[g^{(t)}(\boldsymbol{x}_h) \mid \boldsymbol{x}_{h-1} = x^{(t)}, \boldsymbol{a}_{h-1} = a^{(t)}\big] \leq 2\sqrt{\beta^2 C_{\mathsf{push}} T \log(2T)} + 2BC_{\mathsf{push}},$$

*and consequently*

$$\min_{t \in [T]} \mathbb{E}\big[g^{(t)}(\boldsymbol{x}_h) \mid \boldsymbol{x}_{h-1} = x^{(t)}, \boldsymbol{a}_{h-1} = a^{(t)}\big] \leq 2\sqrt{\frac{\beta^2 C_{\mathsf{push}} \log(2T)}{T}} + \frac{2BC_{\mathsf{push}}}{T}.$$

**Proof of [Lemma C.8]().** Define $d_h^{(t)}(x) \coloneqq \mathbb{P}[\boldsymbol{x}_h = x \mid \boldsymbol{x}_{h-1} = x^{(t)}, \boldsymbol{a}_{h-1} = a^{(t)}]$, and let $\widetilde{d}_h^{(t)} \coloneqq \sum_{i < t} d^{(i)}$. Let

$$\tau_h(x) \coloneqq \min\big\{t \mid \widetilde{d}_h^{(t)}(x) \geq C_{\mathsf{push}} \cdot \mu_h(x)\big\}.$$

We have

$$\sum_{t=1}^{T} \mathbb{E}\big[g^{(t)}(\boldsymbol{x}_h) \mid \boldsymbol{x}_{h-1} = x^{(t)}, \boldsymbol{a}_{h-1} = a^{(t)}\big]$$

$$= \sum_{t=1}^{T} \sum_{x \in \mathcal{X}} d_h^{(t)}(x) g^{(t)}(x)$$

$$\leq \sum_{t=1}^{T} \sum_{x \in \mathcal{X}} d_h^{(t)}(x) g^{(t)}(x) \mathbb{I}\{t \geq \tau_h(x)\} + B \sum_{t=1}^{T} \sum_{x \in \mathcal{X}} d_h^{(t)}(x) \mathbb{I}\{t < \tau_h(x)\}.$$

From the definition of pushforward coverability, we can bound

$$\sum_{t=1}^{T} \sum_{x \in \mathcal{X}} d_h^{(t)}(x) \mathbb{I}\{t < \tau_h(x)\} = \sum_{x \in \mathcal{X}} \widetilde{d}_h^{(\tau_h(x))}(x) \leq 2C_{\mathsf{push}} \sum_{x \in \mathcal{X}} \mu_h(x) = 2C_{\mathsf{push}}.$$

For the other term, we bound

$$\sum_{t=1}^{T} \sum_{x \in \mathcal{X}} d_h^{(t)}(x) g^{(t)}(x) \mathbb{I}\{t \geq \tau_h(x)\}$$

$$\leq \left( \sum_{t=1}^{T} \sum_{x \in \mathcal{X}} \frac{(d_h^{(t)}(x))^2}{\widetilde{d}_h^{(t)}(x)} \mathbb{I}\{t \geq \tau_h(x)\} \right)^{1/2} \cdot \left( \sum_{t=1}^{T} \sum_{x \in \mathcal{X}} \widetilde{d}_h^{(t)}(x)(g_h^{(t)}(x))^2 \right)^{1/2}$$

$$= \left( \sum_{t=1}^{T} \sum_{x \in \mathcal{X}} \frac{(d_h^{(t)}(x))^2}{\widetilde{d}_h^{(t)}(x)} \mathbb{I}\{t \geq \tau_h(x)\} \right)^{1/2} \cdot \left( \sum_{t=1}^{T} \sum_{i < t} \mathbb{E}\big[(g^{(t)}(\boldsymbol{x}_h))^2 \mid \boldsymbol{x}_{h-1} = x^{(i)}, \boldsymbol{a}_{h-1} = a^{(i)}\big] \right)^{1/2}$$

$$\leq \left( \sum_{t=1}^{T} \sum_{x \in \mathcal{X}} \frac{(d_h^{(t)}(x))^2}{\widetilde{d}_h^{(t)}(x)} \mathbb{I}\{t \geq \tau_h(x)\} \right)^{1/2} \cdot \sqrt{\beta^2 T}.$$

Finally, we have

$$\sum_{t=1}^{T} \sum_{x \in \mathcal{X}} \frac{(d_h^{(t)}(x))^2}{\widetilde{d}_h^{(t)}(x)} \mathbb{I}\{t \geq \tau_h(x)\} \leq 2 \sum_{t=1}^{T} \sum_{x \in \mathcal{X}} \frac{(d_h^{(t)}(x))^2}{\widetilde{d}_h^{(t)}(x) + C_{\mathsf{push}} \mu_h(x)}$$

$$\leq 2 C_{\mathsf{push}} \sum_{t=1}^{T} \sum_{x \in \mathcal{X}} \mu_h(x) \frac{d_h^{(t)}(x)}{\widetilde{d}_h^{(t)}(x) + C_{\mathsf{push}} \mu_h(x)}$$

$$= 2 C_{\mathsf{push}} \sum_{x \in \mathcal{X}} \mu_h(x) \sum_{t=1}^{T} \frac{d_h^{(t)}(x)}{\widetilde{d}_h^{(t)}(x) + C_{\mathsf{push}} \mu_h(x)}$$

$$= 4 C_{\mathsf{push}} \log(T + 1),$$

where the last line uses Lemma 4 of Xie et al. [63]. $\qquad \square$

# Part II

# Proofs for `SimGolf` (Section 3)

As described in Section 3.1, the main difference between `SimGolf` and `GOLF` lies in the construction of the confidence sets. The most important new step in the proof of Theorem 3.1 is to show that the local simulator-based confidence set construction in Line 9 is valid in the sense that the property Eq. (1) holds with high probability. From here, the sample complexity bound follows by adapting the change-of-measure argument based on coverability from Xie et al. [63].

To this end, this part of the appendix is organized as follows. We first state and prove technical lemmas concerning realizability (Lemma D.1) and the confidence set construction (Lemma D.2 and Lemma D.3) in Appendix D. Then, in Appendix E, we prove Theorem 3.1 as a consequence.

## D Preliminary Lemmas for Proof of Theorem 3.1

For this section, we define

$$\ell_h^{(t)}(g) \coloneqq \left(g_h(\boldsymbol{x}_h^{(t)}, \boldsymbol{a}_h^{(t)}) - \frac{1}{K}\sum_{k=1}^{K}\left(\boldsymbol{r}_h^{(t,k)} + \max_{a \in \mathcal{A}} g_{h+1}(\boldsymbol{x}_{h+1}^{(t,k)}, a)\right)\right)^2$$

and

$$\bar{\ell}_h^{(t)}(g) \coloneqq \mathbb{E}^{\pi^{(t)}}\Big[\big(g_h(\boldsymbol{x}_h, \boldsymbol{a}_h) - \mathcal{T}_h[g_{h+1}](\boldsymbol{x}_h, \boldsymbol{a}_h)\big)^2\Big],$$

where $(\boldsymbol{x}_h^{(t)}, \boldsymbol{a}_h^{(t)}, \boldsymbol{r}_h^{(t,k)}, \boldsymbol{x}_{h+1}^{(t,k)})$ are as in Algorithm 1.

**Lemma D.1** ([21]). *For the ExBMDP setting, under Assumption 3.3, the function class* $\mathcal{Q}_h \coloneqq \{(x,a) \mapsto g(\phi(x), a) : g \in [0, H]^{SA}, \phi \in \Phi\}$ *satisfies Assumption 3.1 and has* $\log|\mathcal{Q}_h| = \log|\Pi_h| = \widetilde{O}(SA + \log|\Phi|).$[9]

**Lemma D.2.** *With probability at least* $1 - \delta$*, for all* $h \in [H]$*,* $t \in [N]$*, and* $g \in \mathcal{Q}$*,*

$$\sum_{i<t}\ell_h^{(i)}(g) \leq 3\sum_{i<t}\mathbb{E}^{\pi^{(i)}}\Big[\big(g_h(\boldsymbol{x}_h, \boldsymbol{a}_h) - \mathcal{T}_h[g_{h+1}](\boldsymbol{x}_h, \boldsymbol{a}_h)\big)^2\Big] + \frac{8N}{K} + 16\log(2HN|\mathcal{Q}|\delta^{-1}),$$

*and*

$$\sum_{i<t}\mathbb{E}^{\pi^{(i)}}\Big[\big(g_h(\boldsymbol{x}_h, \boldsymbol{a}_h) - \mathcal{T}_h[g_{h+1}](\boldsymbol{x}_h, \boldsymbol{a}_h)\big)^2\Big] \leq 4\sum_{i<t}\ell_h^{(i)}(g) + \frac{8N}{K} + 64\log(2HN|\mathcal{Q}|\delta^{-1}).$$

**Proof of Lemma D.2.** Let $t \in [N]$ and $h \in [H]$ be fixed. Let us denote $\boldsymbol{z}_h^{(t)} = \big\{(\boldsymbol{r}_h^{(t,k)}, \boldsymbol{x}_{h+1}^{(t,k)})\big\}_{k \in [K]}$. Define a filtration

$$\mathcal{H}^{(t)} = \sigma\big(\boldsymbol{\tau}^{(1)}, \boldsymbol{z}_1^{(1)}, \ldots, \boldsymbol{z}_H^{(1)}, \ldots, \boldsymbol{\tau}^{(t)}, \boldsymbol{z}_1^{(t)}, \ldots, \boldsymbol{z}_H^{(t)}\big),$$

where $\boldsymbol{\tau}^{(i)}$ is the trajectory generated in the $i$th iteration of Algorithm 1 (see Line 7). Fix $g \in \mathcal{Q}$. Observe that $\ell_h^{(i)}(g) \in [0, 4]$, so Lemma C.3 ensures that with probability at least $1 - \delta$,

$$\sum_{i<t}\ell_h^{(i)}(g) \leq \frac{3}{2}\sum_{i<t}\mathbb{E}\big[\ell_h^{(i)}(g) \mid \mathcal{H}^{(i-1)}\big] + 16\log(2\delta^{-1}),$$

and

$$\sum_{i<t}\mathbb{E}\big[\ell_h^{(i)}(g) \mid \mathcal{H}^{(i-1)}\big] \leq 2\sum_{i<t}\ell_h^{(i)}(g) + 32\log(2\delta^{-1}). \tag{8}$$

---

[9] Formally, this requires a standard covering number argument; we omit the details.

By the AM-GM inequality, for all $i < t$, we can bound

$$\mathbb{E}\big[\ell_h^{(i)}(g) \mid \mathcal{H}^{(i-1)}\big]$$

$$= \mathbb{E}\left[\left(g_h(\boldsymbol{x}_h^{(i)}, \boldsymbol{a}_h^{(i)}) - \frac{1}{K}\sum_{k=1}^{K}\left(\boldsymbol{r}_h^{(i,k)} + \max_{a\in\mathcal{A}} g_{h+1}(\boldsymbol{x}_{h+1}^{(i,k)}, a)\right)\right)^2 \mid \mathcal{H}^{(i-1)}\right]$$

$$\leq 2\,\mathbb{E}\left[\left(g_h(\boldsymbol{x}_h^{(i)}, \boldsymbol{a}_h^{(i)}) - \mathcal{T}_h[g_{h+1}](\boldsymbol{x}_h^{(i)}, \boldsymbol{a}_h^{(i)})\right)^2 \mid \mathcal{H}^{(i-1)}\right]$$

$$\quad + 2\,\mathbb{E}\left[\left(\mathcal{T}_h[g_{h+1}](\boldsymbol{x}_h^{(i)}, \boldsymbol{a}_h^{(i)}) - \frac{1}{K}\sum_{k=1}^{K}\left(\boldsymbol{r}_h^{(i,k)} + \max_{a\in\mathcal{A}} g_{h+1}(\boldsymbol{x}_{h+1}^{(i,k)}, a)\right)\right)^2 \mid \mathcal{H}^{(i-1)}\right].$$

and

$$\mathbb{E}\big[\ell_h^{(i)}(g) \mid \mathcal{H}^{(i-1)}\big]$$

$$\geq \frac{1}{2}\,\mathbb{E}\left[\left(g_h(\boldsymbol{x}_h^{(i)}, \boldsymbol{a}_h^{(i)}) - \mathcal{T}_h[g_{h+1}](\boldsymbol{x}_h^{(i)}, \boldsymbol{a}_h^{(i)})\right)^2 \mid \mathcal{H}^{(i-1)}\right]$$

$$\quad - \mathbb{E}\left[\left(\mathcal{T}_h[g_{h+1}](\boldsymbol{x}_h^{(i)}, \boldsymbol{a}_h^{(i)}) - \frac{1}{K}\sum_{k=1}^{K}\left(\boldsymbol{r}_h^{(i,k)} + \max_{a\in\mathcal{A}} g_{h+1}(\boldsymbol{x}_{h+1}^{(i,k)}, a)\right)\right)^2 \mid \mathcal{H}^{(i-1)}\right].$$

We have

$$\mathbb{E}\left[\left(g_h(\boldsymbol{x}_h^{(i)}, \boldsymbol{a}_h^{(i)}) - \mathcal{T}_h[g_{h+1}](\boldsymbol{x}_h^{(i)}, \boldsymbol{a}_h^{(i)})\right)^2 \mid \mathcal{H}^{(i-1)}\right] = \mathbb{E}^{\pi^{(i)}}\left[\left(g_h(\boldsymbol{x}_h, \boldsymbol{a}_h) - \mathcal{T}_h[g_{h+1}](\boldsymbol{x}_h, \boldsymbol{a}_h)\right)^2\right]$$

and

$$\mathbb{E}\left[\left(\mathcal{T}_h[g_{h+1}](\boldsymbol{x}_h^{(i)}, \boldsymbol{a}_h^{(i)}) - \frac{1}{K}\sum_{k=1}^{K}\left(\boldsymbol{r}_h^{(i,k)} + \max_{a\in\mathcal{A}} g_{h+1}(\boldsymbol{x}_{h+1}^{(i,k)}, a)\right)\right)^2 \mid \mathcal{H}^{(i-1)}\right]$$

$$= \mathbb{E}\left[\mathbb{E}\left[\left(\mathcal{T}_h[g_{h+1}](\boldsymbol{x}_h, \boldsymbol{a}_h) - \frac{1}{K}\sum_{k=1}^{K}\left(\boldsymbol{r}_h^{(i,k)} + \max_{a\in\mathcal{A}} g_{h+1}(\boldsymbol{x}_{h+1}^{(i,k)}, a)\right)\right)^2 \mid \boldsymbol{x}_h = \boldsymbol{x}_h^{(i)}, \boldsymbol{a}_h = \boldsymbol{a}_h^{(i)}\right] \mid \mathcal{H}^{(i-1)}\right].$$

Since

$$\mathbb{E}\left[\boldsymbol{r}_h^{(i,k)} + \max_{a\in\mathcal{A}} g_{h+1}(\boldsymbol{x}_{h+1}^{(i,k)}, a) \mid \boldsymbol{x}_h = \boldsymbol{x}_h^{(i)}, \boldsymbol{a}_h = \boldsymbol{a}_h^{(i)}\right] = \mathcal{T}_h[g_{h+1}](\boldsymbol{x}_h^{(i)}, \boldsymbol{a}_h^{(i)})$$

and $\big\{(\boldsymbol{r}_h^{(i,k)}, \boldsymbol{x}_{h+1}^{(i,k)})\big\}_{k\in[K]}$ are i.i.d. conditioned on $(\boldsymbol{x}_h, \boldsymbol{a}_h) = (\boldsymbol{x}_h^{(i)}, \boldsymbol{a}_h^{(i)})$, we have,

$$\mathbb{E}\left[\left(\mathcal{T}_h[g_{h+1}](\boldsymbol{x}_h, \boldsymbol{a}_h) - \frac{1}{K}\sum_{k=1}^{K}\left(\boldsymbol{r}_h^{(i,k)} + \max_{a\in\mathcal{A}} g_{h+1}(\boldsymbol{x}_{h+1}^{(i,k)}, a)\right)\right)^2 \mid \boldsymbol{x}_h = \boldsymbol{x}_h^{(i)}, \boldsymbol{a}_h = \boldsymbol{a}_h^{(i)}\right]$$

$$= \frac{1}{K}\,\mathbb{E}\left[\left(\mathcal{T}_h[g_{h+1}](\boldsymbol{x}_h, \boldsymbol{a}_h) - \left(\boldsymbol{r}_h^{(i,k)} + \max_{a\in\mathcal{A}} g_{h+1}(\boldsymbol{x}_h^{(i,k)}, a)\right)\right)^2 \mid \boldsymbol{x}_h = \boldsymbol{x}_h^{(i)}, \boldsymbol{a}_h = \boldsymbol{a}_h^{(i)}\right]$$

$$\leq \frac{4}{K},$$

so that

$$\mathbb{E}\left[\left(\mathcal{T}_h[g_{h+1}](\boldsymbol{x}_h^{(i)}, \boldsymbol{a}_h^{(i)}) - \frac{1}{K}\sum_{k=1}^{K}\left(\boldsymbol{r}_h^{(i,k)} + \max_{a\in\mathcal{A}} g_{h+1}(\boldsymbol{x}_{h+1}^{(i,k)}, a)\right)\right)^2 \mid \mathcal{H}^{(i-1)}\right] \leq \frac{4}{K}.$$

Combining these bounds with (8) and rearranging thus gives

$$\sum_{i<t}\ell_h^{(i)}(g) \leq 3\sum_{i<t}\mathbb{E}^{\pi^{(i)}}\left[\left(g_h(\boldsymbol{x}_h, \boldsymbol{a}_h) - \mathcal{T}_h[g_{h+1}](\boldsymbol{x}_h, \boldsymbol{a}_h)\right)^2\right] + \frac{8N}{K} + 16\log(2\delta^{-1}),$$

and

$$\sum_{i<t}\mathbb{E}^{\pi^{(i)}}\left[\left(g_h(\boldsymbol{x}_h, \boldsymbol{a}_h) - \mathcal{T}_h[g_{h+1}](\boldsymbol{x}_h, \boldsymbol{a}_h)\right)^2\right] \leq 4\sum_{i<t}\ell_h^{(i)}(g) + \frac{8N}{K} + 64\log(2\delta^{-1}).$$

Taking a union bound yields the result. $\qquad\square$

**Lemma D.3.** *Define $\beta_{\text{stat}} = 16\log(2HN|\mathcal{Q}|\delta^{-1})$. Suppose we set $K \geq \frac{8N}{\beta_{\text{stat}}}$ and $\beta \geq 2\beta_{\text{stat}}$. Then with probability at least $1 - \delta$, for all $t \in [N]$ and $h \in \mathcal{H}$:*

- $Q^\star \in \mathcal{Q}^{(t)}$.

- *All $g \in \mathcal{Q}^{(t)}$ satisfy*

$$\sum_{i<t} \mathbb{E}^{\pi^{(i)}} \Big[ (g_h(\boldsymbol{x}_h, \boldsymbol{a}_h) - \mathcal{T}_h[g_{h+1}](\boldsymbol{x}_h, \boldsymbol{a}_h))^2 \Big] \leq 9\beta.$$

**Proof of Lemma D.3.** Condition on the event in Lemma D.2. For any fixed $t \in [N]$ and $h \in [H]$, we have that

$$\sum_{i<t} \ell_h^{(i)}(Q^\star) \leq 3\sum_{i<t} \mathbb{E}^{\pi^{(i)}} \Big[ (Q_h^\star(\boldsymbol{x}_h, \boldsymbol{a}_h) - \mathcal{T}_h[Q_{h+1}^\star](\boldsymbol{x}_h, \boldsymbol{a}_h))^2 \Big] + \frac{8N}{K} + 16\log(2HN|\mathcal{Q}|\delta^{-1})$$

$$\leq \frac{8N}{K} + 16\log(2HN|\mathcal{Q}|\delta^{-1}) \leq 2\beta_{\text{stat}},$$

where the first inequality uses that $Q_h^\star = \mathcal{T}_h[Q_{h+1}^\star]$ and the second inequality uses our choice for $K$. It follows that $Q^\star \in \mathcal{Q}^{(t)}$ as long as $\beta \geq 2\beta_{\text{stat}}$.

To prove the second claim, we note that for all $g \in \mathcal{Q}^{(t)}$, by construction,

$$\sum_{i<t} \mathbb{E}^{\pi^{(i)}} \Big[ (g_h(\boldsymbol{x}_h) - \mathcal{T}_h[g_{h+1}](\boldsymbol{x}_h, \boldsymbol{a}_h))^2 \Big] \leq 4\sum_{i<t} \ell_h^{(i)}(g) + \frac{8N}{K} + 64\log(2HN|\mathcal{Q}|\delta^{-1})$$

$$\leq 4\sum_{i<t} \ell_h^{(i)}(g) + 5\beta_{\text{stat}} \leq 9\beta.$$

$\square$

# E    Proof of Theorem 3.1

**Proof of Theorem 3.1.** From Lemma D.3, the parameter setting in the theorem statement ensures that with probability at least $1 - \delta$, for all $t \in [2 \mathinner{..} N]$, $Q^\star \in \mathcal{Q}^{(t)}$, and all $g \in \mathcal{Q}^{(t)}$ satisfy

$$\sum_{i<t} \mathbb{E}^{\pi^{(i)}} \Big[ (g_h(\boldsymbol{x}_h) - \mathcal{T}_h[g_{h+1}](\boldsymbol{x}_h, \boldsymbol{a}_h))^2 \Big] \leq 9\beta. \tag{9}$$

for all $h$. Let us condition on this event going forward. First, note that since $Q^\star \in \mathcal{Q}^{(t)}$ for all $t \in [2 \mathinner{..} N]$, we have that

$$J(\pi^\star) \leq \mathbb{E}\Big[ \max_{a \in \mathcal{A}} Q_1^\star(\boldsymbol{x}_1, a) \Big] \leq \sup_{g \in \mathcal{Q}^{(t)}} \mathbb{E}\Big[ \max_{a \in \mathcal{A}} g_1(\boldsymbol{x}_1, a) \Big]. \tag{10}$$

On the other hand, we have $g^{(t)} \in \arg\max_{g \in \mathcal{Q}^{(t)}} \sum_{s<t} \max_{a \in \mathcal{A}} g_1(\boldsymbol{x}_1^{(s)}, a)$, and so since $\boldsymbol{x}_1^{(1)}, \boldsymbol{x}_1^{(2)}, \ldots$ are i.i.d. and any $g \in \mathcal{Q}^{(t)}$ take values in $[0, H]$, we have that by Hoeffding's inequality, there is an event $\mathcal{E}$ of probability at least $1 - \delta$ under which

$$\forall t \in [2 \mathinner{..} N], \forall g \in \mathcal{Q}, \quad \left| \mathbb{E}\Big[ \max_{a \in \mathcal{A}} g_1(\boldsymbol{x}_1, a) \Big] - \frac{1}{t-1}\sum_{s<t} \max_{a \in \mathcal{A}} g_1(\boldsymbol{x}_1^{(s)}, a) \right| \leq \sqrt{(t-1)^{-1}\log(2N|\mathcal{Q}|/\delta)}. \tag{11}$$

This implies that under $\mathcal{E}$, we have

$$\forall t \in [2 \mathinner{..} N], \quad \sup_{g \in \mathcal{Q}^{(t)}} \mathbb{E}\Big[ \max_{a \in \mathcal{A}} g_1(\boldsymbol{x}_1, a) \Big] \leq \sup_{g \in \mathcal{Q}^{(t)}} \frac{1}{t-1}\sum_{s<t} \max_{a \in \mathcal{A}} g_1(\boldsymbol{x}_1^{(s)}, a) + \sqrt{(t-1)^{-1}\log(2N|\mathcal{Q}|/\delta)},$$

$$= \frac{1}{t-1}\sum_{s<t} \max_{a \in \mathcal{A}} g_1^{(t)}(\boldsymbol{x}_1^{(s)}, a) + \sqrt{(t-1)^{-1}\log(2N|\mathcal{Q}|/\delta)},$$

$$\leq \mathbb{E}\Big[ \max_{a \in \mathcal{A}} g_1^{(t)}(\boldsymbol{x}_1, a) \Big] + 2\sqrt{(t-1)^{-1}\log(2N|\mathcal{Q}|/\delta)}, \tag{12}$$

where in the last inequality we have used (11) with $f = g^{(t)}$. Thus, summing (12) for $t = 2, \ldots N$ and using (10) gives that under $\mathcal{E}$:

$$\sum_{t=2}^{N} J(\pi^{\star}) \leq \sum_{t=2}^{N} \mathbb{E}[g_1^{(t)}(\boldsymbol{x}_1, \boldsymbol{a}_1)] + 4\sqrt{N \log(2N|\mathcal{Q}|/\delta)},$$

and so since $J(\pi^{\star}) \leq H$,

$$\sum_{t=1}^{N} J(\pi^{\star}) \leq \sum_{t=1}^{N} \mathbb{E}[g_1^{(t)}(\boldsymbol{x}_1, \boldsymbol{a}_1)] + 4\sqrt{N \log(2N|\mathcal{Q}|/\delta)} + H. \tag{13}$$

On the other hand, using that $g_{H+1}^{(t)} \equiv 0$, we get

$$\sum_{t=1}^{N} (\mathbb{E}\left[g_1^{(t)}(\boldsymbol{x}_1, \boldsymbol{a}_1)\right] - J(\pi^{(t)}))$$

$$\leq \sum_{t=1}^{N} \sum_{h=1}^{H} \mathbb{E}^{\pi^{(t)}} \left[ g_h^{(t)}(\boldsymbol{x}_h, \boldsymbol{a}_h) - \boldsymbol{r}_h - \max_{a \in \mathcal{A}} g_{h+1}^{(t)}(\boldsymbol{x}_{h+1}, a) \right],$$

$$= \sum_{t=1}^{N} \sum_{h=1}^{H} \mathbb{E}^{\pi^{(t)}} \left[ g_h^{(t)}(\boldsymbol{x}_h, \boldsymbol{a}_h) - \mathbb{E}\left[ \boldsymbol{r}_h + \max_{a \in \mathcal{A}} g_{h+1}^{(t)}(\boldsymbol{x}_{h+1}, a) \mid \boldsymbol{x}_h, \boldsymbol{a}_h \right] \right], \quad \text{(law of total expectation)}$$

$$= \sum_{t=1}^{N} \sum_{h=1}^{H} \mathbb{E}^{\pi^{(t)}} \left[ g_h^{(t)}(\boldsymbol{x}_h, \boldsymbol{a}_h) - \mathcal{T}_h[g_{h+1}^{(t)}](\boldsymbol{x}_h, \boldsymbol{a}_h) \right].$$

and so, by the potential lemma (Lemma C.7) and (9), we have

$$\leq 6H\sqrt{C_{\mathsf{cov}}\beta N \log(2N)} + 2H^2 C_{\mathsf{cov}}.$$

Combining this with (13), we obtain that with probability at least $1 - 2\delta$,

$$\sum_{t=1}^{N} (J(\pi^{\star}) - J(\pi^{(t)})) \leq 6H\sqrt{C_{\mathsf{cov}}\beta N \log(2N)} + 4\sqrt{N \log(2N|\mathcal{Q}|/\delta)} + 3H^2 C_{\mathsf{cov}}.$$

It follows that if $N = \widetilde{O}(H^2 C_{\mathsf{cov}}\beta/\varepsilon^2)$, then the policy

$$\widehat{\pi} \in \mathsf{unif}(\pi^{(1)}, \ldots, \pi^{(N)})$$

returned by SimGolf satisfies, with probability at least $1 - \delta$:

$$J(\pi^{\star}) - \mathbb{E}[J(\widehat{\pi})] \leq \varepsilon.$$

**Sample complexity.** We now bound the number of episodes. Note that that within an iteration $t$ of SimGolf, the local simulator is called $KH$ times to update the confidence set, where $K \leq N/\log(2HN|\mathcal{Q}|/\delta))$. Consequently, the total sample complexity is bounded by

$$HNK \leq \widetilde{O}(H^5 C_{\mathsf{cov}}^2 \log(|\mathcal{Q}|/\delta)/\varepsilon^4).$$

$\square$

# Part III

# Proofs for RVFS (Section 4)

## F  Full Version of RVFS

Algorithm 5 displays the full version of RVFS. Algorithm 6 contains an "outer-level" wrapper for RVFS, RVFS.bc, which invokes RVFS and extracts an executable policy with imitation learning, and Algorithm 7 contains the subroutine used within Algorithm 5 to approximate Bellman backups for value functions using local simulator access. Additionally, we display the variant of RVFS for Exogenous Block MDPs, described in Appendix B, in Algorithms 8 and 9. Before diving into the proof, we first describe how the full version of the algorithm differs from the informal version presented in the main body in greater detail.

**Differences between full version (Algorithm 5) and informal version (Algorithm 2) of RVFS.** The main difference between Algorithm 2 and its full version in Algorithm 5 is that in the former we simply assume access to quantities involving conditional expectations such as:

- The bellman backups $\mathcal{P}_h[\widehat{V}_{h+1}]$, which are required to evaluate the actions of RVFS's policies (see (2)), and to perform the tests in Line 8; and

- The value functions $\mathbb{E}^{\widehat{\pi}_{h+1:H}}\left[\sum_{\ell=h}^{H} r_\tau \mid \boldsymbol{x}_h = x, \boldsymbol{a}_h = a\right]$ in Line 15, which are needed in the regression problem in Line 16.

These quantities are not available to the algorithm directly, but they can be estimated using the local simulator. This is reflected in the full version of RVFS in Algorithm 5.

**Extracting policies from value functions.** Let us briefly comment in more detail on how Algorithm 5 extracts the policy $\pi^{(t)}$ from the optimistic value function $f^{(t)} \in \mathcal{V}$ at iteration $t$. From the Bellman equation, the ideal choice would be to set $\pi_h^{(t)}(x) = \arg\max_{a \in \mathcal{A}} \mathcal{P}_h[f_{h+1}^{(t)}](x, a)$, but this requires knowledge of the transition distribution. Instead, given parameters $\varepsilon, \delta \in (0, 1)$, SimGolf invokes Algorithm 7 via $\pi_h^{(t)}(x) \in \arg\max_{a \in \mathcal{A}} \widehat{\boldsymbol{\mathcal{P}}}_{h,\varepsilon,\delta}[f_{h+1}^{(t)}](x, a)$. The operator $\widehat{\boldsymbol{\mathcal{P}}}_{h,\varepsilon,\delta}[f]$ (Algorithm 7), when given input $(x, a) \in \mathcal{X} \times \mathcal{A}$ and $f_{h+1} : \mathcal{X} \to \mathbb{R}$, uses the local simulator to generate $N_{\mathtt{sim}} \geq 1$ next states $\boldsymbol{x}_{h+1}^{(1)}, \ldots, \boldsymbol{x}_{h+1}^{(N_{\mathtt{sim}})} \overset{\text{i.i.d.}}{\sim} T_h(\cdot \mid x, a)$ to estimate the bellman back-up $\mathcal{P}_h[f_{h+1}]$ via $\frac{1}{N_{\mathtt{sim}}} \sum_{i=1}^{N_{\mathtt{sim}}} (\boldsymbol{r}_h^{(i)} + f_{h+1}(\boldsymbol{x}_{h+1}^{(i)}))$, where $\boldsymbol{r}_h^{(1)}, \ldots, \boldsymbol{r}_h^{(N_{\mathtt{sim}})} \overset{\text{i.i.d.}}{\sim} R_h(x, a)$. The number of samples $N_{\mathtt{sim}}$ in Algorithm 7 is set as a function of $(\varepsilon, \delta)$ such that with probability at least $1 - \delta$, $|\widehat{\boldsymbol{\mathcal{P}}}_{h,\varepsilon,\delta}[f_{h+1}](x, a) - \mathcal{P}_h[f_{h+1}](x, a)| \leq \varepsilon$.

**Invoking the algorithm.** The base invocation of RVFS takes the form

$$\widehat{V}_{1:H} \leftarrow \mathsf{RVFS}_0\big(\widehat{V}_{1:H} = \text{arbitrary}, \widehat{\mathcal{V}}_{1:H} = \{\mathcal{V}\}_{h=1}^{H}, \mathcal{C}_{0:H} = \{\varnothing\}_{h=0}^{H}, \mathcal{B}_{0:H} = \{\varnothing\}_{h=0}^{H}, t_{1:H} = \{1\}_{h=1}^{H}; \cdots\big).$$

Whenever this call returns, the greedy policy induced by $\widehat{V}_{1:H}$ is guaranteed to be near-optimal. Naively, the policy induced by $\widehat{V}_{1:H}$ is non-executable, and must be computed by invoking the local simulator through Line 14. To provide an end-to-end guarantee to learn an executable policy, the outer-level algorithm, RVFS.bc (Algorithm 6, invokes $\mathsf{RVFS}_0$, then extracts an executable policy from $\widehat{V}_{1:H}$ using imitation learning.

Subsequent recursive calls take the form

$$(\widehat{V}_{h:H}, \widehat{\mathcal{V}}_{h:H}, \mathcal{C}_{h:H}, \mathcal{B}_{h:H}, t_{h:H}) \leftarrow \mathsf{RVFS}_h(\widehat{V}_{h+1:H}, \widehat{\mathcal{V}}_{h+1:H}, \mathcal{C}_{h:H}, \mathcal{B}_{h:H}, t_{h:H}; \mathcal{V}, \varepsilon, \delta).$$

For such a call, the arguments above are:

- $\widehat{V}_{h+1:H}$: Value function estimates for subsequent layers.

- $\widehat{\mathcal{V}}_{h+1:H}$: Value function confidence sets $\widehat{\mathcal{V}}_{h+1:H} \subset \mathcal{V}_{h+1:H}$, which are used in the test on Line 14 to quantify uncertainty on new state-action pairs and decide whether to expand the core-sets.

- $\mathcal{C}_{h:H}$: Core-sets for current and subsequent layers.

- $\mathcal{B}_{h:H}$: Buffers of tuples $(x_{h-1}, a_{h-1}, \widehat{V}_h, \widehat{\mathcal{V}}_h, t_h)$, which record relevant features of the algorithm's state whenever the test on Line 14 fails and a recursive call is performed.

- $t_{h:H}$: Counters that track the number of times Algorithm 7 is called in the test on Line 14, which facilitate tuning of confidence parameters.

Importantly, the confidence sets $\widehat{\mathcal{V}}_{h+1:H}$ do not need to be explicitly maintained, and can be invoked implicitly whenever a *regression oracle* for the value function class is available (cf. discussion in Section 4). Likewise, the buffers $\mathcal{B}_{h:H}$ are only used in our analysis, and do not need to be explicitly maintained.

### F.1 RVFS Pseudocode

---

**Algorithm 5** $\mathrm{RVFS}_h$: Recursive Value Function Search

---

1: **parameters:** Value function class $\mathcal{V}$, suboptimality $\varepsilon \in (0,1)$, confidence $\delta \in (0,1)$.
2: **input:**
 • Level $h \in \{0, \ldots, H\}$.
 • Value function estimates $\widehat{V}_{h+1:H}$, confidence sets $\widehat{\mathcal{V}}_{h+1:H}$, state-action collections $\mathcal{C}_{h:H}$, buffers $\mathcal{B}_{h:H}$, and counters $t_{h:H}$.
 /* Initialize parameters. */
3: Set $M \leftarrow \lceil 8\varepsilon^{-1} C_{\mathsf{push}} H \rceil$.
4: Set $N_{\mathsf{test}} \leftarrow 2^8 M^2 H \varepsilon^{-1} \log(8M^6 H^8 \varepsilon^{-2} \delta^{-1})$, $N_{\mathsf{reg}} \leftarrow 2^8 M^2 \varepsilon^{-1} \log(8|\mathcal{V}|^2 H M^2 \delta^{-1})$,
5: Set $N_{\mathsf{est}}(k) \leftarrow 2N_{\mathsf{reg}}^2 \log(8A N_{\mathsf{reg}} H k^3/\delta)$ and $\delta' \leftarrow \delta/(8M^7 N_{\mathsf{test}}^2 H^8 |\mathcal{V}|)$.
6: Set $\varepsilon_{\mathsf{reg}}^2 \leftarrow \frac{9MH^2 \log(8M^2 H |\mathcal{V}|^2/\delta)}{N_{\mathsf{reg}}} + \frac{34MH^3 \log(8M^6 N_{\mathsf{test}}^2 H^8/\delta)}{N_{\mathsf{test}}}$.
7: Set $\beta(t) \leftarrow \sqrt{2\log_{1/\delta'}(8AM|\mathcal{V}|t^2/\delta)}$.

 /* Test the fit for the estimated value functions $\widehat{V}_{h+1:H}$ at future layers. */
8: **for** $(x_{h-1}, a_{h-1}) \in \mathcal{C}_h$ **do**
9:  **for** layer $\ell = H, \ldots, h+1$ **do**
10:    **for** $n = 1, \ldots, N_{\mathsf{test}}$ **do**
11:      Draw $\boldsymbol{x}_h \sim T_{h-1}(\cdot \mid x_{h-1}, a_{h-1})$, then draw $\boldsymbol{x}_{\ell-1}$ by rolling out with $\widehat{\pi}_{h:H}$, where[10]

$$\forall \tau \in [H], \quad \widehat{\boldsymbol{\pi}}_\tau(\cdot) \in \arg\max_{a \in \mathcal{A}} \widehat{\boldsymbol{\mathcal{P}}}_{\tau,\varepsilon,\delta'}[\widehat{V}_{\tau+1}](\cdot, a), \quad \text{with} \quad \widehat{V}_{H+1} \equiv 0. \tag{14}$$

12:      **for** $a_{\ell-1} \in \mathcal{A}$ **do**
        /* Number of times $\widehat{\mathcal{P}}_{\ell-1,\varepsilon,\delta'}$ (Algorithm 7) is called in the test on Line 14. */
13:        Update $t_\ell \leftarrow t_\ell + 1$.
        /* Test fit; if test fails, re-fit value functions $\widehat{V}_{h+1:\ell}$ up to layer $\ell$. */
14:        **if** $\sup_{f \in \widehat{\mathcal{V}}_\ell} |(\widehat{\boldsymbol{\mathcal{P}}}_{\ell-1,\varepsilon,\delta'}[\widehat{V}_\ell] - \widehat{\boldsymbol{\mathcal{P}}}_{\ell-1,\varepsilon,\delta'}[f_\ell])(\boldsymbol{x}_{\ell-1}, a_{\ell-1})| > \varepsilon + \varepsilon \cdot \beta(t_\ell)$ **then**
15:          $\mathcal{C}_\ell \leftarrow \mathcal{C}_\ell \cup \{(\boldsymbol{x}_{\ell-1}, a_{\ell-1})\}$ and $\mathcal{B}_\ell \leftarrow \mathcal{B}_\ell \cup \{(\boldsymbol{x}_{\ell-1}, a_{\ell-1}, \widehat{V}_\ell, \widehat{\mathcal{V}}_\ell, t_\ell)\}$.
16:          **for** $\tau = \ell, \ldots, h+1$ **do**
17:            $(\widehat{V}_{\tau:H}, \widehat{\mathcal{V}}_{\tau:H}, \mathcal{C}_{\tau:H}, \mathcal{B}_{\tau:H}, t_{\tau:H}) \leftarrow \mathrm{RVFS}_\tau(\widehat{V}_{\tau+1:H}, \widehat{\mathcal{V}}_{\tau+1:H}, \mathcal{C}_{\tau:H}, \mathcal{B}_{\tau:H}, t_{\tau:H}; \mathcal{V}, \varepsilon, \delta)$.
18:          **go to line 8.**
19: **if** $h = 0$ **then return:** $(\widehat{V}_{1:H}, \cdot, \cdot, \cdot, \cdot)$.
 /* Re-fit $\widehat{V}_h$ and build a new confidence set. */
20: **for** $(x_{h-1}, a_{h-1}) \in \mathcal{C}_h$ **do**
21:  Set $\mathcal{D}_h(x_{h-1}, a_{h-1}) \leftarrow \varnothing$.
22:  **for** $i = 1, \ldots, N_{\mathsf{reg}}$ **do**
23:    Sample $\boldsymbol{x}_h \sim T_{h-1}(\cdot \mid x_{h-1}, a_{h-1})$.
24:    Let $\widehat{V}_h(\boldsymbol{x}_h)$ be a Monte-Carlo estimate for $\mathbb{E}^{\widehat{\pi}_{h:H}}[\sum_{\ell=h}^H \boldsymbol{r}_\ell \mid \boldsymbol{x}_h]$ computed by collecting $N_{\mathsf{est}}(|\mathcal{C}_h|)$ trajectories starting from $\boldsymbol{x}_h$ and rolling out with $\widehat{\pi}_{h:H}$.
25:    Update $\mathcal{D}_h(x_{h-1}, a_{h-1}) \leftarrow \mathcal{D}_h(x_{h-1}, a_{h-1}) \cup \{(\boldsymbol{x}_h, \widehat{V}_h(\boldsymbol{x}_h))\}$.
26: Let $\widehat{V}_h := \arg\min_{f \in \widehat{\mathcal{V}}} \sum_{(x_{h-1}, a_{h-1}) \in \mathcal{C}_h} \sum_{(x_h, v_h) \in \mathcal{D}_h(x_{h-1}, a_{h-1})} (f(x_h) - v_h)^2$.
27: Compute value function confidence set

$$\widehat{\mathcal{V}}_h := \left\{ f \in \mathcal{V} \;\middle|\; \sum_{(x_{h-1}, a_{h-1}) \in \mathcal{C}_h} \frac{1}{N_{\mathsf{reg}}} \sum_{(x_h, \cdot) \in \mathcal{D}_h(x_{h-1}, a_{h-1})} \left(\widehat{V}_h(x_h) - f(x_h)\right)^2 \le \varepsilon_{\mathsf{reg}}^2 \right\}. \tag{15}$$

28: **return** $(\widehat{V}_{h:H}, \widehat{\mathcal{V}}_{h:H}, \mathcal{C}_{h:H}, \mathcal{B}_{h:H}, t_{h:H})$.

---

---

**Algorithm 6** RVFS.bc: Learn an executable policy with RVFS via behavior cloning.

---

1: **input:** Value function class $\mathcal{V}$, policy class $\Pi$, suboptimality $\varepsilon \in (0,1)$, confidence $\delta \in (0,1)$.
  /* Set parameters for RVFS. */
2: Set $\varepsilon_{\mathsf{RVFS}} \leftarrow \varepsilon H^{-1}/48$.
3: Set $\widehat{V}_{1:H} \leftarrow$ arbitrary, $\widehat{\mathcal{V}}_{1:H} \leftarrow \mathcal{V}$, $\mathcal{C}_{0:H} \leftarrow \varnothing$, $\mathcal{B}_{0:H} \leftarrow \varnothing$, and $t_i \leftarrow 0$, for all $i \in [0..H]$.
  /* Set parameters for BehaviorCloning. */
4: Set $M \leftarrow \lceil 8\varepsilon_{\mathsf{RVFS}}^{-1} C_{\mathsf{push}} H \rceil$ and $N_{\mathsf{test}} \leftarrow 2^8 M^2 H \varepsilon_{\mathsf{RVFS}}^{-1} \log(80 M^6 H^8 \varepsilon_{\mathsf{RVFS}}^{-2} \delta^{-1})$.
5: $N_{\mathsf{reg}} \leftarrow 2^8 M^2 \varepsilon_{\mathsf{RVFS}}^{-1} \log(80|\mathcal{V}|^2 H M^2 \delta^{-1})$ and $\delta' = \frac{\delta}{40 M^7 N^2 H^8 |\mathcal{V}|}$.
  /* Get value functions from RVFS */
6: $(\widehat{V}_{1:H}, \cdot, \cdot, \cdot, \cdot) \leftarrow \mathsf{RVFS}_0(\widehat{V}_{1:H}, \widehat{\mathcal{V}}_{1:H}, \mathcal{C}_{0:H}, \mathcal{B}_{0:H}, t_{0:H}; \mathcal{V}, N_{\mathsf{reg}}, N_{\mathsf{test}}, \varepsilon_{\mathsf{RVFS}}, \delta/10)$.
  /* Extract executable policy via BehaviorCloning algorithm for imitation learning. */
7: Define $\widehat{\boldsymbol{\pi}}_h^{\mathsf{RVFS}}(\cdot) \in \arg\max_{a \in \mathcal{A}} \widehat{\boldsymbol{\mathcal{P}}}_{h,\varepsilon_{\mathsf{RVFS}},\delta'}[\widehat{V}_{h+1}](\cdot, a)$.
8: Compute $\widehat{\pi}_{1:H} \leftarrow \mathsf{BehaviorCloning}(\Pi, \varepsilon, \widehat{\pi}_{1:H}^{\mathsf{RVFS}}, \delta/2)$
9: **return:** $\widehat{\pi}_{1:H}$.

---

---

**Algorithm 7** $\widehat{\boldsymbol{\mathcal{P}}}_{h,\varepsilon,\delta}[f]$: Estimate conditional expectation $\mathbb{E}[\boldsymbol{r}_h + f(\boldsymbol{x}_{h+1}) \mid \boldsymbol{x}_h = \cdot, \boldsymbol{a}_h = \cdot]$.

---

1: **parameters:** Layer $h$, suboptimality $\varepsilon \in (0,1)$, confidence $\delta \in (0,1)$, target function $f$.
2: **input:** $(x,a) \in \mathcal{X} \times \mathcal{A}$.
3: Set $N_{\mathsf{sim}} := 2\log(1/\delta)/\varepsilon^2$.
4: Set $\mathcal{D} \leftarrow \varnothing$
5: **for** $i = 1, \ldots, N_{\mathsf{sim}}$ **do**
6:   Sample $\boldsymbol{r}_h \sim R_h(x,a)$ and $\boldsymbol{x}_{h+1} \sim T_h(\cdot \mid x,a)$.     //Uses local simulator access.
7:   Update $\mathcal{D} \leftarrow \mathcal{D} \cup \{(\boldsymbol{r}_h, \boldsymbol{x}_{h+1})\}$.
8: **return:** $N_{\mathsf{sim}}^{-1} \cdot \sum_{(r,x) \in \mathcal{D}} (r + f(x))$.

---

---

**Algorithm 8** RVFS$_h^{\text{exo}}$: Recursive Value Function Search for Exogenous Block MDPs

---

1: **parameters:** Value function class $\mathcal{V}$, suboptimality $\varepsilon \in (0,1)$, seeds $\zeta_{1:H} \in (0,1)$, confidence $\delta \in (0,1)$.

2: **input:** Level $h \in [0..H]$, value function estimates $\widehat{V}_{h+1:H}$, confidence sets $\widehat{\mathcal{V}}_{h+1:H}$, state-action collections $\mathcal{C}_{h:H}$, and buffers $\mathcal{B}_{h:H}$, and counters $t_{h:H}$.

    `/* Initialize parameters. */`

3: Set $M \leftarrow \lceil 8\varepsilon^{-2} C_{\text{exo}} SAH \rceil$.

4: Set $N_{\text{test}} \leftarrow 2^8 M^2 H \varepsilon^{-2} \log(8M^6 H^8 \varepsilon^{-2} \delta^{-1})$, $N_{\text{reg}} \leftarrow 2^8 M^2 \varepsilon^{-2} \log(8|\mathcal{V}| H M^2 \delta^{-1})$.

5: Set $N_{\text{est}}(k) \leftarrow 2N_{\text{reg}}^2 \log(8A N_{\text{reg}} H k^3 / \delta)$ and $\delta' \leftarrow \delta / (4M^7 N_{\text{test}}^2 H^8 |\mathcal{V}|)$.

6: Set $\varepsilon_{\text{reg}}^2 \leftarrow \dfrac{9MH^2 \log(8M^2 H |\mathcal{V}|/\delta)}{N_{\text{reg}}} + \dfrac{34MH^3 \log(8M^6 N_{\text{test}}^2 H^8 / \delta)}{N_{\text{test}}}$.

7: Set $\beta(t) \leftarrow \sqrt{\log_{1/\delta'}(8MA|\mathcal{V}|t^2/\delta)}$.

    `/* Test the fit for the estimated value functions` $\widehat{V}_{h+1:H}$ `at future layers. */`

8: **for** $(x_{h-1}, a_{h-1}) \in \mathcal{C}_h$ **do**

9:     **for** layer $\ell = H, \ldots, h+1$ **do**

10:         **for** $n = 1, \ldots, N_{\text{test}}$ **do**

11:             Draw $\boldsymbol{x}_h \sim T_{h-1}(\cdot \mid x_{h-1}, a_{h-1})$, then draw $\boldsymbol{x}_{\ell-1}$ by rolling out with $\widehat{\pi}_{h+1:H}$, where

$$\forall \tau \in [H], \ \ \widehat{\pi}_\tau(\cdot) \in \arg\max_{a \in \mathcal{A}} \left[ \widehat{\mathcal{P}}_{\tau, \varepsilon^2, \delta'}[\widehat{V}_{\tau+1}](\cdot, a) \cdot \varepsilon^{-1} + \zeta_\tau \right], \quad \text{with} \quad \widehat{V}_{H+1} \equiv 0.$$

12:             **for** $a_{\ell-1} \in \mathcal{A}$ **do**

13:                 Update $t_\ell \leftarrow t_\ell + 1$.

                `/* Test fit; if test fails, re-fit value functions` $\widehat{V}_{h+1:\ell}$ `up to layer` $\ell$. `*/`

14:                 **if** $\sup_{f \in \widehat{\mathcal{V}}_\ell} |(\widehat{\mathcal{P}}_{\ell-1, \varepsilon^2, \delta'}[\widehat{V}_\ell] - \widehat{\mathcal{P}}_{\ell-1, \varepsilon^2, \delta'}[f_\ell])(\boldsymbol{x}_{\ell-1}, a_{\ell-1})| > \varepsilon^2 + \varepsilon^2 \cdot \beta(t_\ell)$ **then**

15:                     $\mathcal{C}_\ell \leftarrow \mathcal{C}_\ell \cup \{(\boldsymbol{x}_{\ell-1}, a_{\ell-1})\}$ and $\mathcal{B}_\ell \leftarrow \mathcal{B}_\ell \cup \{(\boldsymbol{x}_{\ell-1}, a_{\ell-1}, \widehat{V}_\ell, \widehat{\mathcal{V}}_\ell, t_\ell)\}$.

16:                     **for** $\tau = \ell, \ldots, h+1$ **do**

17:                         $(\widehat{V}_{\tau:H}, \widehat{\mathcal{V}}_{\tau:H}, \mathcal{C}_{\tau:H}, \mathcal{B}_{\tau:H}, t_{\tau:H}) \leftarrow \text{RVFS}_\tau^{\text{exo}}(\widehat{V}_{\tau+1:H}, \widehat{\mathcal{V}}_{\tau+1:H}, \mathcal{C}_{\tau:H}, \mathcal{B}_{\tau:H}, t_{\tau:H}; \mathcal{V}, \varepsilon, \zeta_{1:H}, \delta)$.

18:                     **go to line 8**.

19: **if** $h = 0$ **then return** $(\widehat{V}_{1:H}, \cdot, \cdot, \cdot, \cdot)$.

    `/* Re-fit` $\widehat{V}_h$ `and build a new confidence set. */`

20: **for** $(x_{h-1}, a_{h-1}) \in \mathcal{C}_h$ **do**

21:     Set $\mathcal{D}_h(x_{h-1}, a_{h-1}) \leftarrow \varnothing$.

22:     **for** $i = 1, \ldots, N_{\text{reg}}$ **do**

23:         Sample $\boldsymbol{x}_h \sim T_{h-1}(\cdot \mid x_{h-1}, a_{h-1})$.

24:         For each $a \in \mathcal{A}$, let $\widehat{V}_h(\boldsymbol{x}_h)$ be a Monte-Carlo estimate for $\mathbb{E}^{\widehat{\pi}_{h:H}}[\sum_{\ell=h}^H r_\ell \mid \boldsymbol{x}_h]$ computed by collecting $N_{\text{est}}(|\mathcal{C}_h|)$ trajectories starting from $\boldsymbol{x}_h$ and rolling out with $\widehat{\pi}_{h:H}$.

25:         Update $\mathcal{D}(x_{h-1}, a_{h-1}) \leftarrow \mathcal{D}(x_{h-1}, a_{h-1}) \cup \{(\boldsymbol{x}_h, \widehat{V}_h(\boldsymbol{x}_h))\}$.

26: Let $\widehat{V}_h := \arg\min_{f \in \mathcal{V}_h} \sum_{(x_{h-1}, a_{h-1}) \in \mathcal{C}_h} \sum_{(x_h, v_h) \in \mathcal{D}_h(x_{h-1}, a_{h-1})} (f(x_h) - v_h)^2$.

27: Compute value function confidence set

$$\widehat{\mathcal{V}}_h := \left\{ f \in \mathcal{V}_h \ \middle| \ \sum_{(x_{h-1}, a_{h-1}) \in \mathcal{C}_h} \frac{1}{N_{\text{reg}}} \sum_{(x_h, -) \in \mathcal{D}_h(x_{h-1}, a_{h-1})} \left( \widehat{V}_h(x_h) - f(x_h) \right)^2 \leq \varepsilon_{\text{reg}}^2 \right\}. \qquad (16)$$

28: **return** $(\widehat{V}_{h:H}, \widehat{\mathcal{V}}_{h:H}, \mathcal{C}_{h:H}, \mathcal{B}_{h:H}, t_{h:H})$.

---

**Algorithm 9** $\mathsf{RVFS}^{\mathsf{exo}}$.bc: Learn an executable policy with $\mathsf{RVFS}^{\mathsf{exo}}$ via imitation learning.

---

1: **input:** Decoder class $\Phi$, suboptimality $\varepsilon \in (0,1)$, confidence $\delta \in (0,1)$.

    /* Set parameters for RVFS and define the value function and policy classes. */

2: Set $\varepsilon_{\mathsf{RVFS}} \leftarrow \varepsilon H^{-1}/48$.

3: Set $\mathcal{V} = \mathcal{V}_{1:H}$, where $\mathcal{V}_h = \{x \mapsto f(\phi(x)) : f \in [0,H]^S, \phi \in \Phi\}$, $\forall h \in [H]$.

4: Set $\Pi = \Pi_{1:H}$, where $\Pi_h = \{\pi(\cdot) \in \arg\max_{a \in \mathcal{A}} f(\phi(\cdot), a) : f \in [0,H]^{S \times A}, \phi \in \Phi\}$, $\forall h \in [H]$.

    /* Set parameters for BehaviorCloning. */

5: Set $N_{\mathsf{bc}} \leftarrow 8H^2 \log(4H|\Pi|/\delta)/\varepsilon$, $N_{\mathsf{boost}} \leftarrow \log(1/\delta)/\log(24SAH\varepsilon)$, $N_{\mathsf{eval}} \leftarrow 16^2 \varepsilon^{-2} \log(2N_{\mathsf{boost}}/\delta)$.

6: Set $M \leftarrow \lceil 8\varepsilon_{\mathsf{RVFS}}^{-1} SAC_{\mathsf{cov}}H \rceil$, $N_{\mathsf{test}} \leftarrow 2^8 M^2 H \varepsilon_{\mathsf{RVFS}}^{-1} \log(80 M^6 H^8 N_{\mathsf{boost}} \varepsilon_{\mathsf{RVFS}}^{-2} \delta^{-1})$, and $\delta' = \frac{\delta}{40 M^7 N^2 H^8 |\mathcal{V}| N_{\mathsf{boost}}}$.

7: Set $N_{\mathsf{reg}} \leftarrow 2^8 M^2 \varepsilon_{\mathsf{RVFS}}^{-1} \log(80|\Phi|^2 H M^2 N_{\mathsf{boost}} \delta^{-1})$.

8: Set $\widehat{V}_{1:H} \leftarrow$ arbitrary, $\widehat{\mathcal{V}}_{1:H} \leftarrow \mathcal{V}$, $\mathcal{C}_{0:H} \leftarrow \varnothing$, $\mathcal{B}_{0:H} \leftarrow \varnothing$, $i_{\mathsf{opt}} = 1$, and $J_{\mathsf{max}} = 0$.

    /* Repeatedly invoke RVFS$^{\mathsf{exo}}$ and extract policy with BehaviorCloning to boost confidence. */

9: **for** $i = 1, \ldots, N_{\mathsf{boost}}$ **do**

      /* Invoke RVFS$^{\mathsf{exo}}$. */

10:    $(\widehat{V}_{1:H}^{(i)}, \cdot, \cdot, \cdot, \cdot) \leftarrow \mathsf{RVFS}_0^{\mathsf{exo}}(\widehat{V}_{1:H}, \widehat{\mathcal{V}}_{1:H}, \mathcal{C}_{0:H}, \mathcal{B}_{0:H}; \mathcal{V}, N_{\mathsf{reg}}, N_{\mathsf{test}}, \varepsilon_{\mathsf{RVFS}}, \delta/(10N_{\mathsf{boost}}))$.

      /* Imitation learning with BehaviorCloning. */

11:    Define $\widehat{\boldsymbol{\pi}}_h^{\mathsf{RVFS}}(\cdot) \in \arg\max_{a \in \mathcal{A}} \widehat{\mathcal{P}}_{h, \varepsilon_{\mathsf{RVFS}}, \delta'}[\widehat{V}_{h+1}^{(i)}](\cdot, a)$.

12:    Compute $\widehat{\pi}_{1:H}^{(i)} \leftarrow \mathsf{BehaviorCloning}(\Pi, \varepsilon, \widehat{\pi}_{1:H}^{\mathsf{RVFS}}, \delta/(2N_{\mathsf{boost}}))$.

      /* Evaluate current policy. */

13:    $v = 0$.

14:    **for** $= 1, \ldots, N_{\mathsf{eval}}$ **do**

15:      Sample trajectory $(\boldsymbol{x}_1, \boldsymbol{a}_1, \boldsymbol{r}_1, \ldots, \boldsymbol{x}_H, \boldsymbol{a}_H, \boldsymbol{r}_H)$ by executing $\widehat{\pi}_{1:H}^{(i)}$.

16:      Set $v \leftarrow v + \sum_{h=1}^H \boldsymbol{r}_h$.

17:    Set $\widehat{J}(\widehat{\pi}_{1:H}^{(i)}) \leftarrow v/N_{\mathsf{eval}}$.

18:    **if** $\widehat{J}(\widehat{\pi}_{1:H}^{(i)}) > J_{\mathsf{max}}$ **then**

19:      Set $i_{\mathsf{opt}} = i$.

20:      Set $J_{\mathsf{max}} = \widehat{J}(\widehat{\pi}_{1:H}^{(i)})$.

21: **return:** $\widehat{\pi}_{1:H} = \widehat{\pi}_{1:H}^{(i_{\mathsf{opt}})}$.

---

# G Organization

This remainder of Part III of the appendix contains the proofs for the main results concerning the RVFS algorithm (Theorem 4.1 and Theorem B.1).

- First, in Appendix H we give a brief overview of the analysis and introduce a restricted set of *benchmark policies* which will be used throughout the proofs for Theorem 4.1 and Theorem B.1. The benchmark policy class is central to the regret decomposition for RVFS, and facilitates an analysis that does not require optimism.

- In Appendix I, we prove Theorem 4.1 under **Setup II** ($V^\pi$-realizability). This constitutes the main technical development for Theorem 4.1. The central technical results proven here are Theorem I.1, Theorem I.2 and which generalize Theorem 4.1.

- In Appendix J, we prove Theorem 4.1 under **Setup I** ($V^\star$-realizability), as a straightforward consequence of the tools developed in Appendix I (Theorem I.1 and Theorem I.2).

- Finally, in Appendix K, we prove Theorem B.1 (analysis of RVFS$^{\mathsf{exo}}$ for the ExBMDP problem). This analysis has a similar structure to the proof of Theorem 4.1 under **Setup II** in Appendix I, and builds on the same analysis techniques, but requires specialized arguments due to extra technical challenges in the ExBMDP setting.

- Appendix M gives a self-contained presentation of the BehaviorCloning algorithm for imitation learning, which is used within RVFS.bc and RVFS$^{\mathsf{exo}}$.bc.

# H Overview of Analysis and Preliminaries

In this section, we present some notation, technical tools, and preliminary results we require for the analysis of RVFS in the settings we described in Section 4. We start by defining a set of restricted *benchmark policies* used in the regret decomposition for RVFS.

## H.1 Overview of Analysis

In this section we give a brief overview of the analysis techniques behind Theorem 4.1 and Theorem B.1. We focus on Theorem 4.1 to begin.

Recall that RVFS is recursive in the sense that whenever the test in Line 14 fails for a layer $h \in [H]$, an recursive call RVFS$_h$ is initiated. Throughout the recursion, via the steps in Item 1 and Item 2, RVFS maintains the following invariant: whenever a call to RVFS$_h$ (an instance of RVFS initiated at layer $h$) terminates, the confidence sets $\widehat{\mathcal{V}}_{h+1:H}$ that it outputs satisfy, with high probability:

$$\forall \ell \in [h+1..H], \quad V_\ell^\star \in \widehat{\mathcal{V}}_\ell. \tag{Inv1}$$

In addition, RVFS$_h$ can only return if the value function tests in Line 14 (which involve the confidence sets $\widehat{\mathcal{V}}_{h+1:H}$) all succeed. From the invariant in (Inv1), it can be shown that the tests only succeed if the estimated value functions $\widehat{V}_{h+1:H}$ satisfy:

$$\forall \ell \in [h+1..H], \quad \mathbb{P}^{\widehat{\pi}}\left[\sup_{a \in \mathcal{A}} |(\mathcal{P}_\ell[\widehat{V}_{\ell+1}] - \mathcal{P}_\ell[V_{\ell+1}^\star])(\boldsymbol{x}_h, a)| \geq 3\varepsilon\right] \leq \varepsilon_{\mathsf{test}}, \tag{Inv2}$$

where $\varepsilon_{\mathsf{test}} > 0$ is a parameter set by the algorithm. We use pushforward coverability to show that RVFS can only expand the core-sets $\mathcal{C}_{1:H}$ a polynomial number of times before the algorithm terminates and (Inv2) is satisfied.

The invariant in (Inv2) is useful because it ensures that the greedy policies

$$\widehat{\pi}_h(x) \approx \arg\max_{a \in \mathcal{A}} \mathcal{P}_h[\widehat{V}_{h+1}](x, a)$$

induced by the learned value functions $\widehat{V}_{1:H}$ are near-optimal. To make this precise, recall that given parameters $\varepsilon, \delta \in (0, 1)$, the action $\widehat{\pi}_h(x)$ of RVFS's policy at layer $h$ and state $x \in \mathcal{X}$, is given by

$$\widehat{\pi}_h(x) \in \arg\max_{a \in \mathcal{A}} \widehat{\boldsymbol{\mathcal{P}}}_{h,\varepsilon,\delta}[\widehat{V}_{h+1}](x, a), \tag{17}$$

where $\widehat{V}_{h+1}$ is the estimated value function at layer $h+1$. The operator $\widehat{\boldsymbol{\mathcal{P}}}_{h,\varepsilon,\delta}[\widehat{V}_{h+1}]$ (Algorithm 7), when given input $(x, a) \in \mathcal{X} \times \mathcal{A}$, ensures that probability at least $1 - \delta$, $|\widehat{\boldsymbol{\mathcal{P}}}_{h,\varepsilon,\delta}[\widehat{V}_{h+1}](x, a) -$

$\mathcal{P}_h[\widehat{V}_{h+1}](x,a)| \le \varepsilon$. Combining this with the invariant in (Inv2) and the fact that $V_h^\star \equiv \mathcal{P}_h[V_{h+1}^\star]$, one can see that with high probability (over the randomness in $\boldsymbol{x}_h \sim \mathbb{P}^{\widehat{\pi}}$ and $\widehat{\mathcal{P}}$), we have:

$$\max_{a\in\mathcal{A}} |\widehat{\mathcal{P}}_{h,\varepsilon,\delta}[\widehat{V}_{h+1}](\boldsymbol{x}_h,a) - V_h^\star(\boldsymbol{x}_h,a)| \le 4\varepsilon. \tag{18}$$

**Analysis under Setup I.** For Theorem 4.1 (**Setup I**), Eq. (18) together with the definition of $\widehat{\pi}_h$ in Eq. (17) and the gap assumption (Assumption 4.4) implies that if $\varepsilon \le \Delta/8$, we have that with high probability (over the randomness in $\boldsymbol{x}_h \sim \mathbb{P}^{\widehat{\pi}}$ and $\widehat{\mathcal{P}}$),

$$\widehat{\pi}_h(\boldsymbol{x}_h) \in \arg\max_{a\in\mathcal{A}} \widehat{\mathcal{P}}_{h,\varepsilon,\delta}[\widehat{V}_{h+1}](\boldsymbol{x}_h,a) = \arg\max_{a\in\mathcal{A}} V_h^\star(x,a) = \pi_h^\star(\boldsymbol{x}_h).$$

This suffices to show that $\widehat{\pi}$ is near-optimal, since by the performance lemma [33], the suboptimality of $\widehat{\pi}$ can be bounded as

$$J(\pi^\star) - J(\widehat{\pi}) \le \sum_{h=1}^H \mathbb{P}^{\widehat{\pi}}[\widehat{\pi}_h(\boldsymbol{x}_h) \ne \pi_h^\star(\boldsymbol{x}_h)]. \tag{19}$$

This suffices to prove the performance bound in Theorem 4.1 under **Setup I**.

**Analysis under Setup II.** For Theorem 4.1 (**Setup II**), an immediate challenge in applying a similar analysis to **Setup I** is the lack of suboptimality gap $\Delta$, which makes it impossible to directly bound the probability that $\widehat{\pi} \ne \pi^\star$ in Eq. (19). To address this, we introduce a *restricted benchmark policy class* $\Pi_\varepsilon \subset \Pi_S$ in Appendix H.2 below. The class $\Pi_\varepsilon \subset \Pi_S$ is constructed such that that there exists a policy $\pi \in \Pi_\varepsilon$ that (i) is $O(\varepsilon)$-suboptimal, and (ii) emulates certain properties of a gap. Together, these properties facilitate analysis similar to **Setup I**. Overall, this argument is similar to the "virtual policy iteration" analysis in Yin et al. [65].

**Analysis of RVFS^exo.** The analysis of RVFS^exo for ExBMDPs (Theorem B.1) uses the same idea as the analysis for **Setup II**, except that we can only realize $V^\pi$ for *endogenous policies* that act on $\phi^\star(\boldsymbol{x}_h)$. To address this, we use the randomized rounding scheme in RVFS^exo, and the crux of the proof is to show that with high probability, the rounded policies in RVFS^exo "snap" onto endogenous policies, facilitating an argument similar to **Setup II**.

**Generalizing the analysis.** We mention in passing that RVFS can be slightly modified to recover other existing sample complexity guarantees for RL with linear function approximation and local simulator access that do not require pushforward coverability, including linear-$Q^\star$ realizability with gap [40] and $Q^\pi$-realizability [65]; we leave a more general treatment for future work.

## H.2 Benchmark Policy Class and Randomized Policies

As described above, central to our analysis is a set of $O(\varepsilon)$-suboptimal policies against which we benchmark the policies returned RVFS, which emulate certain consequences of the $\Delta$-gap assumption (Assumption 4.4). Before introducing this concept formally, we first define the notion of a *randomized policy*.

**Induced stochastic policies.** Given an arbitrary collection of independent random variables $\widetilde{\boldsymbol{Q}} = (\widetilde{\boldsymbol{Q}}_h(x,a))_{(h,x,a)\in[H]\times\mathcal{X}\times\mathcal{A}}$, we say that a policy $\pi$ is *induced* by $\widetilde{\boldsymbol{Q}}$ if $\pi$ satisfies

$$\forall h \in [H], \forall x \in \mathcal{X}, \quad \boldsymbol{\pi}_h(x) \in \arg\max_{a'\in\mathcal{A}} \widetilde{\boldsymbol{Q}}_h(x,a'), \tag{20}$$

where we use the bold notation $\boldsymbol{\pi}_h(x)$ as shorthand for the random variable $\boldsymbol{a}_h \sim \pi_h(x) \in \Delta(\mathcal{A})$; in other words, for each $x \in \mathcal{X}$, $\pi_h(x) \in \Delta(\mathcal{A})$ is the distribution induced by sampling $\mathbf{Q}_h(x,\cdot)$ and playing the action $\boldsymbol{\pi}_h(x) \in \arg\max_{a'\in\mathcal{A}} \widetilde{\boldsymbol{Q}}_h(x,a')$. If there are ties in (20), we break them by picking the action with the smallest index; we assume without loss of generality that actions in $\mathcal{A}$ are index from $1, \ldots, |\mathcal{A}|$.

**Benchmark policy class.** We now define the benchmark policy class as follows.

**Definition H.1** (Benchmark policy class). *For $\varepsilon \in (0,1)$, let $\Pi_\varepsilon \subseteq \Pi_S$ be the set of stochastic policies such that $\pi \in \Pi_\varepsilon$ if and only if there exists a collection of* independent *random variables* $\widetilde{\boldsymbol{Q}} = (\widetilde{\boldsymbol{Q}}_h(x,a))_{(h,x,a)\in[H]\times\mathcal{X}\times\mathcal{A}}$ *in* $[0,H]$ *such that:*

- *$\pi$ is induced by $\widetilde{\boldsymbol{Q}}$ (i.e. Eq. (20) is satisfied); and*

- *For all $(h, x, a) \in [H] \times \mathcal{X} \times \mathcal{A}$, we have $|(\widetilde{\boldsymbol{Q}}_h - Q_h^\pi)(x, a)| \leq \varepsilon$, almost surely under the draw of $\widetilde{\boldsymbol{Q}}$.*

Intuitively, the set $\Pi_\varepsilon$ contains the set of all (stochastic) policies corresponding to (randomized) state-action value functions that are point-wise $O(\varepsilon)$ close to $Q^\star$. We formalize this claim in the next lemma.

**Lemma H.1** (Suboptimality of benchmark policies)**.** *Let $\varepsilon \in (0, 1)$ be given. Let $\tilde{\pi} \in \Pi_\varepsilon$ be a stochastic policy induced by a collection of (independent) random state-action value functions $(\widetilde{\boldsymbol{Q}}_h(x, a))_{(h,x,a) \in [H] \times \mathcal{X} \times \mathcal{A}} \subset [0, H]$ such that for all $h \in [H]$ and all $(x, a) \in \mathcal{X} \times \mathcal{A}$:*

$$|\widetilde{\boldsymbol{Q}}_h(x, a) - Q_h^{\tilde{\pi}}(x, a)| \leq \varepsilon \quad \text{almost surely, and} \quad \boldsymbol{\tilde{\pi}}_h(x) \in \arg\max_{a' \in \mathcal{A}} \widetilde{\boldsymbol{Q}}_h(x, a').$$

*Then, for all $h \in [H]$,*

$$\forall x \in \mathcal{X}, \quad V_h^\star(x) \leq V_h^{\tilde{\pi}}(x) + 3H\varepsilon. \tag{21}$$

**Proof of Lemma H.1.** Using backward induction over $\ell = H, \dots, 1$, we start by showing that all $\ell$:

$$\forall (x, a) \in \mathcal{X} \times \mathcal{A}, \quad Q_\ell^\star(x, a) \leq \widetilde{\boldsymbol{Q}}_\ell(x, a) + 2\varepsilon \cdot (H - \ell + 1). \tag{22}$$

almost surely. We then instantiate this with $\ell = 1$ and use the fact that $\|\widetilde{\boldsymbol{Q}}_h - Q_h^{\tilde{\pi}}\|_\infty \leq \varepsilon$ to get the desired result.

**Base case** $[\ell = H]$**.** By definition of the state-action value function, we have, for all $\pi \in \Pi_s$, $Q_H^\star \equiv Q_H^\pi$. Thus, since $\sup_{(x,a) \in \mathcal{X} \times \mathcal{A}} |(\widetilde{\boldsymbol{Q}}_H - Q_H^{\tilde{\pi}})(x, a)| \leq \varepsilon$ almost surely (by definition of $\widetilde{\boldsymbol{Q}}_{1:H}$), we get that

$$\forall (x, a) \in \mathcal{X} \times \mathcal{A}, \quad |\widetilde{\boldsymbol{Q}}_H(x, a) - Q_H^\star(x, a)| \leq \varepsilon,$$

almost surely. This implies (21) for $\ell = H$.

**General case** $[\ell < h]$**.** Fix $h \in [H - 1]$ and suppose that (22) holds for all $\ell \in [h + 1, \dots, H]$ almost surely. We show that it holds for $\ell = h$ almost surely. Fix $(x, a) \in \mathcal{X} \times \mathcal{A}$. We have

$$Q_h^\star(x, a) - \widetilde{\boldsymbol{Q}}_h(x, a) = \mathcal{T}_h[Q_{h+1}^\star](x, a) - \mathcal{T}_h[\widetilde{\boldsymbol{Q}}_{h+1}](x, a) + \mathcal{T}_h[\widetilde{\boldsymbol{Q}}_{h+1}](x, a) - \widetilde{\boldsymbol{Q}}_h(x, a),$$
$$\leq 2\varepsilon \cdot (H - h) + \mathcal{T}_h[\widetilde{\boldsymbol{Q}}_{h+1}](x, a) - \widetilde{\boldsymbol{Q}}_h(x, a), \tag{23}$$

almost surely, where the last step follows by the induction hypothesis. We now bound $\mathcal{T}_h[\widetilde{\boldsymbol{Q}}_{h+1}](x, a) - \widetilde{\boldsymbol{Q}}_h(x, a)$. We have, almost surely, that

$$\mathcal{T}_h[\widetilde{\boldsymbol{Q}}_{h+1}](x, a) - \widetilde{\boldsymbol{Q}}_h(x, a) = \mathcal{T}_h[\widetilde{\boldsymbol{Q}}_{h+1}](x, a) - \mathcal{P}_h[V_{h+1}^{\tilde{\pi}}](x, a) + \mathcal{P}_h[V_{h+1}^{\tilde{\pi}}](x, a) - \widetilde{\boldsymbol{Q}}_h(x, a),$$
$$= \mathcal{T}_h[\widetilde{\boldsymbol{Q}}_{h+1}](x, a) - \mathcal{P}_h[V_{h+1}^{\tilde{\pi}}](x, a) + Q_h^{\tilde{\pi}}(x, a) - \widetilde{\boldsymbol{Q}}_h(x, a),$$
$$= \mathcal{T}_h[\widetilde{\boldsymbol{Q}}_{h+1}](x, a) - \mathcal{P}_h[V_{h+1}^{\tilde{\pi}}](x, a) + \varepsilon, \quad \text{(by the assumption on } \widetilde{\boldsymbol{Q}}_h)$$
$$= \mathbb{E}\left[\max_{a' \in \mathcal{A}} \widetilde{\boldsymbol{Q}}_{h+1}(\boldsymbol{x}_{h+1}, a') - Q_{h+1}^{\tilde{\pi}}(\boldsymbol{x}_{h+1}, \boldsymbol{\tilde{\pi}}_{h+1}(\boldsymbol{x}_{h+1})) \mid \boldsymbol{x}_h = x, \boldsymbol{a}_h = a\right] + \varepsilon,$$
$$= \mathbb{E}\left[\widetilde{\boldsymbol{Q}}_{h+1}(\boldsymbol{x}_{h+1}, \boldsymbol{\tilde{\pi}}_{h+1}(\boldsymbol{x}_{h+1})) - Q_{h+1}^{\tilde{\pi}}(\boldsymbol{x}_{h+1}, \boldsymbol{\tilde{\pi}}_{h+1}(\boldsymbol{x}_{h+1})) \mid \boldsymbol{x}_h = x, \boldsymbol{a}_h = a\right] + \varepsilon,$$
$$\leq 2\varepsilon, \tag{24}$$

where the penultimate inequality follows by the definition of $\boldsymbol{\tilde{\pi}}_{h+1}$, and the last inequality follows by the fact that $\|\widetilde{\boldsymbol{Q}}_{h+1} - Q_{h+1}^{\tilde{\pi}}\|_\infty \leq \varepsilon$ almost surely, by assumption. Plugging (24) into (23) completes the induction, and so we have that

$$\forall (x, a) \in \mathcal{X} \times \mathcal{A}, \quad Q_1^\star(x, a) \leq \widetilde{\boldsymbol{Q}}_1(x, a) \leq 2H\varepsilon.$$

In particular, taking the max over $a$ on both sides and using the definition of $\tilde{\pi}$, we get that

$$\forall x \in \mathcal{X}, \quad V_1^\star(x) \leq \widetilde{\boldsymbol{Q}}_1(x, \boldsymbol{\tilde{\pi}}_1(x)) \leq 2H\varepsilon,$$

almost surely. Combining this with the fact that $\widetilde{\boldsymbol{Q}}_1(x, \boldsymbol{\tilde{\pi}}_1(x)) \leq Q_1^{\tilde{\pi}}(x, \boldsymbol{\tilde{\pi}}_1(x)) + \varepsilon$, almost surely (since $\|\widetilde{\boldsymbol{Q}}_1 - Q_1^{\tilde{\pi}}\|_\infty \leq \varepsilon$ almost surely by assumption) implies that

$$V_1^\star(x) \leq Q_1^\pi(x, \boldsymbol{\tilde{\pi}}_1(x)) + 2H\varepsilon + \varepsilon.$$

Taking expectation over $\boldsymbol{\tilde{\pi}}_1(x)$ and bounding $2H\varepsilon + \varepsilon$ by $3H\varepsilon$ leads to the desired result.

$\square$

### H.3 Additional Preliminaries

The following lemma gives a guarantee for the Bellman backup approximation algorithm $\widehat{\mathcal{P}}$ (Algorithm 7) that is tailored to the analysis of RVFS.

**Lemma H.2.** *Let $\varepsilon, \delta, \delta' \in (0, 1)$, $B > 0$, and $h \in [H]$, be given and let $\mathcal{V}$ be a finite function class. For any sequence $(x_i)_{i \geq 1} \subset \mathcal{X}$ of state action pairs, the outputs $(\widehat{\mathcal{P}}_{h,\varepsilon,\delta'}[f](x_i, a))_{i \geq 1, a \in \mathcal{A}}$ of Algorithm 7 satisfy, with probability at least $1 - \delta$,*

$$\forall i \geq 1, \forall f \in \mathcal{V}, \forall a \in \mathcal{A}, \quad |\widehat{\mathcal{P}}_{h,\varepsilon,\delta'}[f](x_i, a) - \mathcal{P}_h[f](x_i, a)| \leq \varepsilon \cdot \sqrt{2 \log_{1/\delta'}(2Ai^2|\mathcal{V}|/\delta)}.$$

**Proof of Lemma H.2.** By Hoeffding's inequality [28] and the union bound over $a \in \mathcal{A}$ and $f \in \mathcal{V}$, we have that for any $i \geq 1$, with probability at least $1 - \delta/(2i^2)$,

$$\forall f \in \mathcal{V}, \forall a \in \mathcal{A}, \quad |\widehat{\mathcal{P}}_{h,\varepsilon,\delta'}[f](x_i, a) - \mathcal{P}_h[f](x_i, a)| \leq \varepsilon \cdot \sqrt{2 \log_{1/\delta'}(2Ai^2|\mathcal{V}|/\delta)}.$$

The desired result follows by the union bound over $i \geq 1$ and the fact that $\sum_{i \geq 1} 1/i^2 = \pi^2/6 \leq 2$. $\quad\square$

## I Guarantee under $V^\pi$-Realizability (Proof of Theorem 4.1, Setup II)

In this section, we prove Theorem 4.1 under **Setup II**. First, in Appendix I.1 we state a number of supporting technical lemmas, then use them to prove a more general version of Theorem 4.1, Theorem I.2, which holds under a weaker realizability assumption (informally, $V^\pi$-realizability only for *near-optimal* policies $\pi$); Theorem 4.1 follows as an immediate consequence. The remainder of the section (Appendix I.2 through to Appendix I.6) contains the proofs for the intermediate results.

### I.1 Analysis: Proof of Theorem 4.1 (Setup II)

We analyze RVFS in the setting of Theorem 4.1 (**Setup II**), where we have a function class $\mathcal{V}$ satisfying $V^\pi$-realizability (Assumption 4.5). We will actually show that the conclusion of Theorem 4.1 holds under a weaker function approximation setup we refer to as *relaxed $V^\pi$-realizability*: instead of requiring $V^\pi$-realizability for all $\pi \in \Pi_S$, we only require it for policies $\pi$ in the set of near-optimal policies corresponding to the *benchmark policy class* $\Pi_{\varepsilon_{\mathrm{real}}}$ for some $\varepsilon_{\mathrm{real}} > 0$ ($\Pi_\varepsilon$ is defined in Appendix H).

**Assumption I.1** (Relaxed $V^\pi$-realizability). *For $\varepsilon_{\mathrm{real}} > 0$ and all $\pi \in \Pi_{\varepsilon_{\mathrm{real}}}$ and $h \in [H]$, we have $V_h^\pi(x) \in \mathcal{V} \subseteq \{f : \mathcal{X} \to [0, H]\}$.*

We will analyze RVFS under Assumption I.1 and Assumption 4.1. However, it turns out that all of the main results for RVFS can be derived under this assumption: As we will see in Appendix J in the sequel, when the $\Delta$-gap assumption (Assumption 4.4) is satisfied, then $\Pi_{\varepsilon_{\mathrm{real}}} = \{\pi^\star\}$ for all $\varepsilon_{\mathrm{real}} < \Delta$, allowing us to prove Theorem 4.1 under **Setup I** as a special case of relaxed $V^\pi$-realizability. Our analysis for the ExBMDP setting in Appendix K requires more work, but uses that for ExBMDPs, Assumption I.1 is satisfied for a subset of $\Pi_{\varepsilon_{\mathrm{real}}}$ corresponding to endogenous policies.

We begin with our analysis under **Setup II** by bounding the number of times the test in Line 14 fails. Since the sizes of the core sets $\mathcal{C}_{1:H}$ in RVFS are directly related to the number of test failures, the next result, which bounds $|\mathcal{C}_h|$ for $h \in [H]$, allows us to show that RVFS terminates in polynomial iterations with high probability. The proof is in Appendix I.2.

**Lemma I.1** (Bounding the number of test failures). *Let $\delta, \varepsilon \in (0, 1)$ be given, and suppose that Assumption 4.1 (pushforward coverability) holds with parameter $C_{\mathrm{push}} > 0$. Further, let $f \in \mathcal{V}$, be given, where $\mathcal{V}$ is an arbitrary function class. Then there is an event $\mathcal{E}$ of probability at least $1 - \delta$ under which any call $\mathrm{RVFS}_0(f, \mathcal{V}_{1:H}, \varnothing, \varnothing, 0; \mathcal{V}, \varepsilon, \delta)$ (Algorithm 5) terminates, and throughout the execution of $\mathrm{RVFS}_0$, we have*

$$\forall h \in [H], \quad |\mathcal{C}_h| \leq \lceil 8\varepsilon^{-1} C_{\mathrm{push}} H \rceil. \tag{25}$$

In particular, Lemma I.1 ensures that with high probability, every call to $\mathrm{RVFS}_h$ (made recursively via the call to $\mathrm{RVFS}_0$) terminates in polynomial iterations. When $\mathrm{RVFS}_h$ terminates, all the tests in Line 14 must have passed for all $\ell > h$. Using this and a standard concentration argument, we get the following guarantee for the estimated value functions and confidence sets returned by each call to $\mathrm{RVFS}_h$. The proof is in Appendix I.3.

**Lemma I.2** (Consequence of passing the tests). *Let $h \in [0..H]$ and $\varepsilon, \delta \in (0,1)$ be given and consider a call to $\mathsf{RVFS}_0$ in the setting of Lemma I.1. Further, let $\mathcal{E}$ be the event of Lemma I.1. There exists an event $\mathcal{E}'_h$ of probability at least $1 - \delta/H$ such that under $\mathcal{E} \cap \mathcal{E}'_h$, if a call to $\mathsf{RVFS}_h$ within the execution of $\mathsf{RVFS}_0$ terminates and returns $(\widehat{V}_{h:H}, \widehat{\mathcal{V}}_{h:H}, \mathcal{C}_{h:H}, \mathcal{B}_{h:H}, t_{h:H})$, then for any $(x_{h-1}, a_{h-1}) \in \mathcal{C}_h$ and $\ell \in [h+1..H+1]$:*

$$\mathbb{P}^{\widehat{\pi}}\left[\sup_{f \in \widehat{\mathcal{V}}_\ell} \max_{a \in \mathcal{A}} \left|\mathcal{P}_{\ell-1}[\widehat{V}_\ell] - \mathcal{P}_{\ell-1}[f_\ell](\boldsymbol{x}_{\ell-1}, a)\right| > 3\varepsilon \mid \boldsymbol{x}_{h-1} = x_{h-1}, \boldsymbol{a}_{h-1} = a_{h-1}\right] \leq \frac{4\log(8M^6 N_{\mathsf{test}}^2 H^8/\delta)}{N_{\mathsf{test}}},$$

(26)

*where $(\widehat{\pi}_\tau)_{\tau \geq h}$ is the stochastic policy induced by $\widehat{V}_{h:H}$ and $M$ and $N_{\mathsf{test}}$ are defined as in Algorithm 5. Furthermore, under the event $\mathcal{E}$, the total number of times the operator $\widehat{\mathcal{P}}$ is called in the test of Line 14 of Algorithm 5 is at most $O(C_{\mathsf{push}} N_{\mathsf{test}} H^4 \varepsilon^{-1})$.*

We now give a guarantee for the estimated value functions $\widehat{V}_{1:H}$ computed within RVFS in Line 26 (the proof is in Appendix I.4).

**Lemma I.3** (Value function regression guarantee). *Let $h \in [0..H]$ and $\delta, \varepsilon' \in (0,1)$ be given, and consider a call to $\mathsf{RVFS}_0$ in the setting of Lemma I.1. Further, let $\Pi' \subseteq \Pi_{\mathsf{S}}$ be a finite policy class such that the class $\mathcal{V}$ realizes the value functions $V^\pi$ for $\pi \in \Pi'$ (i.e. $\mathcal{V}$ satisfies Assumption I.1 with $\Pi_{\varepsilon_{\mathsf{real}}}$ replaced by $\Pi'$). Then, there is an event $\mathcal{E}''_h$ of probability at least $1 - \delta/H$ under which for all $k \geq 1$, if*

1. *$\mathsf{RVFS}_h$ gets called for the $k$th time during the execution of $\mathsf{RVFS}_0$; and*

2. *this $k$th call terminates and returns $(\widehat{V}_{h:H}, \widehat{\mathcal{V}}_{h:H}, \mathcal{C}_{h:H}, \mathcal{B}_{h:H}, t_{h:H})$,*

*then if $(\widehat{\pi}_\tau)_{\tau \geq h}$ is the policy induced by $\widehat{V}_{h:H}$ and $N_{\mathsf{reg}}$ is set as in Algorithm 5, we have that for all $\pi \in \Pi'$,*

$$\sum_{(x_{h-1}, a_{h-1}) \in \mathcal{C}_h} \frac{1}{N_{\mathsf{reg}}} \sum_{(x_h, -) \in \mathcal{D}_h(x_{h-1}, a_{h-1})} \left(\widehat{V}_h(x_h) - V_h^\pi(x_h)\right)^2$$

$$\leq \frac{9kH^2 \log(8k^2 H |\Pi'||\mathcal{V}|/\delta)}{N_{\mathsf{reg}}} + 8H^2 \sum_{(x_{h-1}, a_{h-1}) \in \mathcal{C}_h} \sum_{\tau=h}^H \mathbb{E}^{\widehat{\pi}}\left[D_{\mathsf{tv}}(\widehat{\pi}_\tau(\boldsymbol{x}_\tau), \pi_\tau(\boldsymbol{x}_\tau)) \mid \boldsymbol{x}_{h-1} = x_{h-1}, \boldsymbol{a}_{h-1} = a_{h-1}\right],$$

*where the datasets $\{\mathcal{D}_h(x, a) : (x, a) \in \mathcal{C}_h\}$ are as in the definition of $\widehat{\mathcal{V}}_h$ in (15).*

Next, we use this result to show that the confidence sets $\widehat{\mathcal{V}}_{1:H}$ returned by $\mathsf{RVFS}_0$ are "valid" in the sense that they contain a value function $(V_h^{\tilde{\pi}})$ corresponding to a near-optimal stochastic policy $\tilde{\pi}$ in the benchmark class $\Pi_{4\varepsilon}$. In the sequel, we use this fact to substitute $V_\ell^{\tilde{\pi}}$ for $f_\ell$ in Eq. (26) and bound the suboptimality of the learned policy $\widehat{\pi}$.

**Lemma I.4** (Confidence sets). *Let $\varepsilon, \delta \in (0,1)$ be given and suppose that Assumption 4.1 (push-forward coverability) holds with parameter $C_{\mathsf{push}} > 0$. Let $f \in \mathcal{V}$ be arbitrary, and suppose that $\mathcal{V}$ satisfies Assumption I.1 with $\varepsilon_{\mathsf{real}} = 4\varepsilon$. Then, there is an event $\mathcal{E}'''$ of probability at least $1 - 3\delta$ under which a call to $\mathsf{RVFS}_0(f, \mathcal{V}, \varnothing, \varnothing, 0; \mathcal{V}, \varepsilon, \delta)$ (Algorithm 5) terminates and returns tuple $(\widehat{V}_{1:H}, \widehat{\mathcal{V}}_{1:H}, \mathcal{C}_{1:H}, \mathcal{B}_{1:H}, t_{1:H})$ such that*

$$\forall h \in [H], \quad V_h^{\tilde{\pi}} \in \widehat{\mathcal{V}}_h,$$

*where $\tilde{\pi}_{1:H} \in \Pi_{\mathsf{S}}$ is the stochastic policy defined recursively via*

$$\forall x \in \mathcal{X}, \quad \tilde{\pi}_\tau(x) \in \arg\max_{a \in \mathcal{A}} \begin{cases} \widehat{Q}_\tau(x, a), & \text{if } \|\widehat{Q}_\tau(x, \cdot) - \mathcal{P}_\tau[V_{\tau+1}^{\tilde{\pi}}](x, \cdot)\|_\infty \leq 4\varepsilon, \\ \mathcal{P}_\tau[V_{\tau+1}^{\tilde{\pi}}](x, a), & \text{otherwise,} \end{cases} \quad \text{for } \tau = H, \ldots, 1,$$

(27)

*where $\widehat{Q}_\tau(x, a) \coloneqq \widehat{\mathcal{P}}_{\tau, \varepsilon, \delta'}[\widehat{V}_{\tau+1}](x, a)$ is a realization of the stochastic output of the $\widehat{\mathcal{P}}$ operator in Algorithm 7 given input $(x, a)$, and $\delta'$ is as in Algorithm 5. Furtherore, we have $\tilde{\pi} \in \Pi_{4\varepsilon}$.*

The proof of the lemma is in Appendix I.5.

Equipped with the preceding lemmas, we now state the main technical result of this section, Theorem I.1, a generalization of Theorem 4.1 which holds under relaxed $V^\pi$-realizability (Assumption I.1). The proof is in Appendix I.6.

**Theorem I.1** (Guarantee for RVFS under relaxed $V^\pi$-realizability). *Let $\delta, \varepsilon \in (0, 1)$ be given, and suppose that [Assumption 4.1](pushforward coverability) holds with parameter $C_{\mathsf{push}} > 0$. Let $f \in \mathcal{V}$ be arbitrary, and assume that $\mathcal{V}$ that satisfies [Assumption I.1](with $\varepsilon_{\mathsf{real}} = 4\varepsilon$). Then, with probability at least $1 - 5\delta$, $\mathsf{RVFS}_0(f, \mathcal{V}, \varnothing, \varnothing, 0; \mathcal{V}, \varepsilon, \delta)$ ([Algorithm 5](.)) terminates and returns value functions $\widehat{V}_{1:H}$ that satisfy*

$$\forall h \in [H], \quad \mathbb{E}^{\widehat{\pi}}[D_{\mathsf{tv}}(\widehat{\pi}_h(\boldsymbol{x}_h), \tilde{\pi}_h(\boldsymbol{x}_h))] \le \frac{\varepsilon}{4H^3 C_{\mathsf{push}}},$$

*where $\widehat{\pi}_h(x) \in \arg\max_{a \in \mathcal{A}} \widehat{\mathcal{P}}_{h, \varepsilon, \delta'}[\widehat{V}_{h+1}](x, a)$ for all $h \in [H]$, with $\tilde{\pi} \in \Pi_{4\varepsilon}$ defined as in [Lemma I.4](.) and $\delta'$ defined as in [Algorithm 5](.). Furthermore, the number of episodes is bounded by*

$$\widetilde{O}(C_{\mathsf{push}}^8 H^{10} A \cdot \varepsilon^{-13}).$$

Next, we state a guarantee for the outer-level algorithm, $\mathsf{RVFS.bc}$, under relaxed $V^\pi$-realizability. Recall that $\mathsf{RVFS.bc}$ invokes $\mathsf{RVFS}_0$, then extracts an executable policy by applying the $\mathsf{BehaviorCloning}$ algorithm (see [Appendix M](.)), with the "expert" policy set to be the output of RVFS.

**Theorem I.2** (Main guarantee of $\mathsf{RVFS.bc}$). *Let $\delta, \varepsilon \in (0, 1)$ be given, and define $\varepsilon_{\mathsf{RVFS}} = \varepsilon H^{-1}/48$. Suppose that*

- *[Assumption 4.1](pushforward coverability) holds with parameter $C_{\mathsf{push}} > 0$;*
- *the function class $\mathcal{V}$ satisfies [Assumption I.1](with $\varepsilon_{\mathsf{real}} = 1$ (i.e. all $\pi$-realizability); and*
- *the policy class $\Pi$ satisfies [Assumption 4.3](.).*

*Then, with probability at least $1 - \delta$, $\widehat{\pi}_{1:H} = \mathsf{RVFS.bc}(\Pi, \mathcal{V}, \varepsilon, \delta)$ ([Algorithm 6](.)) satisfies*

$$J(\pi^\star) - J(\widehat{\pi}_{1:H}) \le \varepsilon. \tag{28}$$

*Furthermore, the total number sample complexity in the RLLS framework is bounded by*

$$\widetilde{O}\left(C_{\mathsf{push}}^8 H^{23} A \epsilon^{-13}\right).$$

The proof is in [Appendix I.7](.). Note that [Theorem I.2](.) is a restatement of [Theorem 4.1](.) in **Setup II** (restated for convenience). As a result, [Theorem 4.1](.) is an immediate corollary.

**Proof of [Theorem 4.1](.).** The result follows from [Theorem I.2](.), since [Assumption 4.5](.) is stronger than [Assumption I.1](.). $\qquad\square$

## I.2  Proof of Lemma I.1 (Number of Test Failures)

**Proof of [Lemma I.1](.).** Fix $h \in [H]$. We note that the size of $\mathcal{C}_h$ corresponds to the number of times the test in [Line 14](.) fails for $\ell = h$ throughout the execution of $\mathsf{RVFS}_0(f, \mathcal{V}, \varnothing, \varnothing; \mathcal{V}, \varepsilon, \delta)$.

Let $M \coloneqq \lceil 8\varepsilon^{-1} C_{\mathsf{push}} H \rceil$ denote the desired upper bound on $|\mathcal{C}_h|$. Suppose that the test in [Line 14](.) fails at least twice for $\ell = h$ (if the test fails at most twice, then $|\mathcal{C}_h| \le 2$ and so [(25)](.) holds for $\ell = h$ trivially), and let

$$(x_{h-1}^{(1)}, a_{h-1}^{(1)}, \widehat{V}_h^{(1)}, \widehat{\mathcal{V}}_h^{(1)}, t_h^{(1)}), (x_{h-1}^{(2)}, a_{h-1}^{(2)}, \widehat{V}_h^{(2)}, \widehat{\mathcal{V}}_h^{(2)}, t_h^{(2)}), \dots$$

denote the elements of the set $\mathcal{B}_h$ in the order at which they are added to the latter in [Line 15](.) of [Algorithm 5](.). Note that $|\mathcal{B}_h| = |\mathcal{C}_h|$. Note also that $t_h^{(i)}$ represents the number of times the subroutine $\widehat{\mathcal{P}}_{h-1, \varepsilon, \delta'}$ has been called in the test of [Line 14](.) throughout the execution of $\mathsf{RVFS}_0$ and up to the time the test failed for $(x_{h-1}^{(i)}, a_{h-1}^{(i)})$. We will use this fact in a concentration argument in the sequel.

By definition of $(\widehat{\mathcal{V}}_h^{(i)})$ and [Lemma C.4](Freedman's inequality) instantiated with

- $\mathcal{Q} = \{\widehat{V}_h^{(i)} - f_h : f \in \widehat{\mathcal{V}}_h^{(i)}\}$;
- $\boldsymbol{y}_h = \boldsymbol{x}_h$;
- $B = H$; and

- $n = N_{\mathsf{reg}} \cdot i$;

and the union bound over $i \in [M \wedge |\mathcal{C}_h|]$, we get that there is an event $\mathcal{E}_h$ of probability at least $1 - \delta/(2H)$ under which

$$\forall i \in [M \wedge |\mathcal{C}_h|], \forall f \in \widehat{\mathcal{V}}_h^{(i)}, \quad \sum_{j<i} \mathbb{E}\big[(\widehat{V}_h^{(i)}(\boldsymbol{x}_h) - f_h(\boldsymbol{x}_h))^2 \mid \boldsymbol{x}_{h-1} = x_{h-1}^{(j)}, \boldsymbol{a}_{h-1} = a_{h-1}^{(j)}\big]$$

$$\leq \tilde{\varepsilon}_{\mathsf{reg}}^2 := 2\varepsilon_{\mathsf{reg}}^2 + \frac{4H^2 \log(4MH|\mathcal{V}|/\delta)}{N_{\mathsf{reg}}}. \tag{29}$$

Now, define $f_h^{(i)} \in \arg\max_{f \in \widehat{\mathcal{V}}_h^{(i)}} \big|\mathbb{E}\big[\widehat{V}_h^{(i)}(\boldsymbol{x}_h) - f_h(\boldsymbol{x}_h) \mid \boldsymbol{x}_{h-1} = x_{h-1}^{(i)}, \boldsymbol{a}_{h-1} = a_{h-1}^{(i)}\big]\big|$. From (29), we have that under $\mathcal{E}_h$:

$$\forall i \in [M \wedge |\mathcal{C}_h|], \quad \sum_{j<i} \mathbb{E}\big[(\widehat{V}_h^{(i)}(\boldsymbol{x}_h) - f_h^{(i)}(\boldsymbol{x}_h))^2 \mid \boldsymbol{x}_{h-1} = x_{h-1}^{(j)}, \boldsymbol{a}_{h-1} = a_{h-1}^{(j)}\big] \leq \tilde{\varepsilon}_{\mathsf{reg}}^2. \tag{30}$$

We now use this to bound the number of times the test in Line 14 fails for $\ell = h$. Suppose for the sake of contradiction that the test fails at least $N$ times for some $N \geq M$ (i.e. $|\mathcal{C}_h| = N \geq M$). Conditioned on $\mathcal{E}_h$, we have by Lemma C.8 and Eq. (30),

$$\min_{i \in [M]} \sup_{f \in \widehat{\mathcal{V}}_h^{(i)}} \big|\mathbb{E}\big[\widehat{V}_h^{(i)}(\boldsymbol{x}_h) - f_h(\boldsymbol{x}_h) \mid \boldsymbol{x}_{h-1} = x_{h-1}^{(i)}, \boldsymbol{a}_{h-1} = a_{h-1}^{(i)}\big]\big|$$

$$= \min_{i \in [M]} \big|\mathbb{E}\big[\widehat{V}_h^{(i)}(\boldsymbol{x}_h) - f_h^{(i)}(\boldsymbol{x}_h) \mid \boldsymbol{x}_{h-1} = x_{h-1}^{(i)}, \boldsymbol{a}_{h-1} = a_{h-1}^{(i)}\big]\big|,$$

$$\leq 2\left(\frac{C_{\mathsf{push}}}{M^2} M \tilde{\varepsilon}_{\mathsf{reg}}^2 \log(2M)\right)^{1/2} + \frac{2C_{\mathsf{push}}H}{M}.$$

Now, substituting the expression of $\tilde{\varepsilon}_{\mathsf{reg}}^2$ in (29) and using the definition of $\varepsilon_{\mathsf{reg}}^2$ in Line 6 of Algorithm 5, we get

$$= 2\left(\frac{C_{\mathsf{push}}}{M} \cdot \left(2\varepsilon_{\mathsf{reg}}^2 + \frac{4MH^2 \log(4MH|\mathcal{V}|/\delta)}{N_{\mathsf{reg}}}\right)\right)^{1/2} + \frac{2C_{\mathsf{push}}H}{M},$$

$$\leq 2\left(\frac{C_{\mathsf{push}}}{M} \cdot \left(\frac{22MH^2 \log(8M^2H|\mathcal{V}|^2/\delta)}{N_{\mathsf{reg}}} + \frac{68MH^3 \log(8M^6 N_{\mathsf{test}}^2 H^8/\delta)}{N_{\mathsf{test}}}\right)\right)^{1/2} + \frac{2C_{\mathsf{push}}H}{M},$$

$$\leq \varepsilon, \tag{31}$$

where the last inequality uses that $M = \lceil 8\varepsilon^{-1} C_{\mathsf{push}} H \rceil$ and

$$N_{\mathsf{reg}} = 2^8 M^2 \varepsilon^{-1} \log(8|\mathcal{V}|^2 H M^2 \delta^{-1}) \text{ and } N_{\mathsf{test}} = 2^8 M^2 H \varepsilon^{-1} \log(8M^6 H^8 \varepsilon^{-2} \delta^{-1});$$

see Line 5 of Algorithm 5.

On the other hand, by Lemma H.2, there is an event $\mathcal{E}_h'$ of probability at least $1 - \delta/(2MH)$ under which for all $f \in \mathcal{V}$, all $i \in [M]$, and $\delta'$ as in Algorithm 5:

$$\big|\widehat{\mathcal{P}}_{h-1,\varepsilon,\delta'}[\widehat{V}_h^{(i)}](x_{h-1}^{(i)}, a_{h-1}^{(i)}) - \widehat{\mathcal{P}}_{h-1,\varepsilon,\delta'}[f_h](x_{h-1}^{(i)}, a_{h-1}^{(i)})\big|$$

$$\leq \big|\mathcal{P}_{h-1}[\widehat{V}_h^{(i)} - f_h](x_{h-1}^{(i)}, a_{h-1}^{(i)})\big| + \varepsilon \cdot \sqrt{2 \log_{1/\delta'}(4MAH|\mathcal{V}|(t_h^{(i)})^2/\delta)},$$

$$= \big|\mathcal{P}_{h-1}[\widehat{V}_h^{(i)} - f_h](x_{h-1}^{(i)}, a_{h-1}^{(i)})\big| + \varepsilon \cdot \beta(t_h^{(i)}), \tag{32}$$

where $\beta(t_h^{(i)})$ is as in Algorithm 5. Thus, under $\mathcal{E}_h'$, the test in Line 14 fails for $\ell = h$ at least $M$ times only if

$$\forall i \in [M], \quad \varepsilon < \sup_{f \in \widehat{\mathcal{V}}_h^{(i)}} \big|(\widehat{\mathcal{P}}_{h-1,\varepsilon,\delta'}[\widehat{V}_h^{(i)}] - \widehat{\mathcal{P}}_{h-1,\varepsilon,\delta'}[f_h])(x_{h-1}^{(i)}, a_{h-1}^{(i)})\big|$$

$$- \varepsilon \cdot \beta(t_h^{(i)}),$$

$$\leq \sup_{f \in \widehat{\mathcal{V}}_h^{(i)}} \big|\mathbb{E}\big[\widehat{V}_h^{(i)}(\boldsymbol{x}_h) - f_h(\boldsymbol{x}_h) \mid \boldsymbol{x}_{h-1} = x_{h-1}^{(i)}, \boldsymbol{a}_{h-1} = a_{h-1}^{(i)}\big]\big| \quad \text{(by (32))},$$

$$< \sup_{f \in \widehat{\mathcal{V}}_h^{(i)}} \big|\mathbb{E}\big[\widehat{V}_h^{(i)}(\boldsymbol{x}_h) - f_h(\boldsymbol{x}_h) \mid \boldsymbol{x}_{h-1} = x_{h-1}^{(i)}, \boldsymbol{a}_{h-1} = a_{h-1}^{(i)}\big]\big|.$$

Unless $N < M$, this is a contradiction to Eq. (31). We conclude that under the event $\mathcal{E}_h \cap \mathcal{E}'_h$, the test in Line 14 fails at most $N < M = \lceil 8\varepsilon^{-1}C_{\mathsf{push}}H\rceil$ times for $\ell = h$, and so under $\mathcal{E}_1 \cap \mathcal{E}'_1 \cap \cdots \cap \mathcal{E}_H \cap \mathcal{E}_H$, we have

$$\forall h \in [H], \quad |\mathcal{C}_h| \le \lceil 8\varepsilon^{-1}C_{\mathsf{push}}H\rceil.$$

By the union bound, we have $\mathbb{P}[\mathcal{E}_1 \cap \mathcal{E}'_1 \cap \cdots \cap \mathcal{E}_H \cap \mathcal{E}'_H] \ge 1 - \delta$, which completes the proof. $\qquad\square$

### I.3  Proof of Lemma I.2 (Consequence of Passing the Tests)

**Proof of Lemma I.2.** Let $h \in [H]$ be given. Fix $\ell \in [h+1..H]$ and let $\boldsymbol{x}^{(1)}_{\ell-1}, \boldsymbol{x}^{(2)}_{\ell-1}, \ldots$ denote the sequence of states used in the tests of Line 14 throughout the execution of $\mathsf{RVFS}_0$; we assume that the sequence is *ordered* in the sense that if $i < j$, then $\boldsymbol{x}^{(i)}_{\ell-1}$ is used in the test of Line 14 before $\boldsymbol{x}^{(j)}_{\ell-1}$. Let $\boldsymbol{T}_\ell \in \mathbb{N} \cup \{+\infty\}$ be the random variable representing the total number of times the operator $\widehat{\boldsymbol{\mathcal{P}}}_{\ell-1,\varepsilon,\delta'}$ is invoked in Line 14 throughout the execution of $\mathsf{RVFS}_0$ ($\boldsymbol{T}_\ell$ is also the random length of the sequence $\boldsymbol{x}^{(1)}_{\ell-1}, \boldsymbol{x}^{(2)}_{\ell-1}, \ldots$; if $\mathsf{RVFS}_0$ terminates, then $\boldsymbol{T}_\ell$ is finite. The first step of the proof will be to show that under the event $\mathcal{E}$ of Lemma I.1, $\boldsymbol{T}_\ell$ is no larger than $M^3 N_{\mathsf{test}}H^3$ at any point during the execution of $\mathsf{RVFS}_0$. This will help us establish key concentration results, leading to the desired inequality (26).

**Bounding $\boldsymbol{T}_\ell$ under $\mathcal{E}$.**  First, note that under the event $\mathcal{E}$ of Lemma I.1, we have that for any $\tau \in [H]$,

$$|\mathcal{C}_\tau| \le M \coloneqq \lceil 8\varepsilon^{-1}C_{\mathsf{push}}H\rceil, \tag{33}$$

and so $\mathsf{RVFS}_\tau$ gets called at most $M$ times throughout the execution of $\mathsf{RVFS}_0$. For the rest of this paragraph, we condition on $\mathcal{E}$ and fix $\tau \in [0..H]$. Within any given call to $\mathsf{RVFS}_\tau$ (throughout the execution of $\mathsf{RVFS}_0$), the operator $\widehat{\boldsymbol{\mathcal{P}}}_{\ell-1,\varepsilon,\delta'}$ is invoked at most

$$\underbrace{|\mathcal{C}_\tau|N_{\mathsf{test}}H}_{\text{Due to the for-loops in Line 8, Line 9, \& Line 10}} \times \underbrace{HM}_{\text{Number of times } \mathsf{RVFS}_\tau \text{ returns to Line 8 (see below)}} \le M^2 N_{\mathsf{test}}H^2$$

times. This is because the for-loop in Line 8 of $\mathsf{RVFS}_\tau$ resumes whenever a test in Line 14 fails for one of the layers $\tau+1, \ldots, H$ (see Line 18) once the recursive calls return, and the total number of test failures across all these layers is bounded by $HM$ (by (33)). Now, since $\mathsf{RVFS}_\tau$ gets called at most $M$ times throughout the execution of $\mathsf{RVFS}_0$ (as argued in the prequel), the total number of times the operator $\widehat{\boldsymbol{\mathcal{P}}}_{\ell-1,\varepsilon,\delta'}$ is invoked in Line 14 within $\mathsf{RVFS}_\tau$ is at most

$$M^3 N_{\mathsf{test}}H^2.$$

Finally, the total number of times the operator $\widehat{\boldsymbol{\mathcal{P}}}_{\ell-1,\varepsilon,\delta'}$ is called in Line 14 throughout the execution of $\mathsf{RVFS}_0$ is at most $H$ times larger (accounting for the contributions from $\mathsf{RVFS}_\tau$ for all $\tau \in [H]$); that is, it is at most $M^3 N_{\mathsf{test}}H^3$. We conclude that the random variable $\boldsymbol{T}_\ell$ satisfies

$$\boldsymbol{T}_\ell \le M^3 N_{\mathsf{test}}H^3 \tag{34}$$

under $\mathcal{E}$.

**Specifying $\mathcal{E}'_h$.**  In this paragraph, we no longer condition on $\mathcal{E}$. We will specify the event $\mathcal{E}'_h$ in the lemma statement. Let $\delta'$ be defined as in Algorithm 5. By Lemma H.2, we have that there is an event $\mathcal{E}'_{h,\ell}$ of probability at least $1 - \delta/(2H^2)$ under which:

$$\forall i \in [\boldsymbol{T}_\ell], \forall a_{\ell-1} \in \mathcal{A} : \sup_{f \in \widehat{\mathcal{V}}_\ell} |(\widehat{\boldsymbol{\mathcal{P}}}_{\ell-1,\varepsilon,\delta'}[\widehat{V}_\ell] - \widehat{\boldsymbol{\mathcal{P}}}_{\ell-1,\varepsilon,\delta'}[f_\ell])(\boldsymbol{x}^{(i)}_{\ell-1}, a_{\ell-1})| - \varepsilon - \varepsilon \cdot \beta(\boldsymbol{T}_\ell)$$

$$= \sup_{f \in \widehat{\mathcal{V}}_\ell} |(\widehat{\boldsymbol{\mathcal{P}}}_{\ell-1,\varepsilon,\delta'}[\widehat{V}_\ell] - \widehat{\boldsymbol{\mathcal{P}}}_{\ell-1,\varepsilon,\delta'}[f_\ell])(\boldsymbol{x}^{(i)}_{\ell-1}, a_{\ell-1})| - \varepsilon - \varepsilon \cdot \sqrt{2\log_{1/\delta'}(8AH^2M|\mathcal{V}|\boldsymbol{T}_\ell^2/\delta)},$$

$$\ge \sup_{f \in \widehat{\mathcal{V}}_\ell} |(\mathcal{P}_{\ell-1}[\widehat{V}_\ell] - \mathcal{P}_{\ell-1}[f_\ell])(\boldsymbol{x}^{(i)}_{\ell-1}, a_{\ell-1})| - \varepsilon - \varepsilon \cdot \sqrt{2\log_{1/\delta'}(8AH^2M|\mathcal{V}|\boldsymbol{T}_\ell^2/\delta)}$$

$$\quad - \varepsilon \cdot \sqrt{2\log_{1/\delta'}(4AH^2M|\mathcal{V}|i^2/\delta)}, \quad \text{(Lemma H.2)}$$

$$\ge \sup_{f \in \widehat{\mathcal{V}}_\ell} |(\mathcal{P}_{\ell-1}[\widehat{V}_\ell] - \mathcal{P}_{\ell-1}[f_\ell])(\boldsymbol{x}^{(i)}_{\ell-1}, a_{\ell-1})| - \varepsilon - 2\varepsilon \cdot \sqrt{2\log_{1/\delta'}(8AH^2M|\mathcal{V}|\boldsymbol{T}_\ell^2/\delta)}. \tag{35}$$

On the other hand, for $k \in [\boldsymbol{T}_\ell - N_{\mathsf{test}} + 1]$, we have by Lemma C.4 (Freedman's inequality) instantiated with

- $n = N_{\text{test}}$ and $\boldsymbol{y}_i = \mathbb{I}\big\{\sup_{f \in \widehat{\mathcal{V}}_\ell} \max_{a \in \mathcal{A}} \big|(\mathcal{P}_{\ell-1}[\widehat{V}_\ell] - \mathcal{P}_{\ell-1})[f_\ell](\boldsymbol{x}_{\ell-1}^{(k+i)}, a)\big| > 3\varepsilon\big\}$, for all $i \in [N_{\text{test}}]$;

- $\mathcal{Q} = \{\text{id}\}$;

- $B = 1$; and

- $\lambda = 1$;

that there is an event $\mathcal{E}''_{h,\ell,k}$ of probability at least $1 - \delta/(4k^2 H^2)$ under which

$$\sum_{0 \le i < N_{\text{test}}} \mathbb{P}\left[\sup_{f \in \widehat{\mathcal{V}}_\ell} \max_{a \in \mathcal{A}} \big|(\mathcal{P}_{\ell-1}[\widehat{V}_\ell] - \mathcal{P}_{\ell-1}[f_\ell])(\boldsymbol{x}_{\ell-1}^{(k+i)}, a)\big| > 3\varepsilon\right]$$

$$\le 4\log(8H^2 \boldsymbol{T}_\ell^2/\delta) + \sum_{0 \le i < N_{\text{test}}} \mathbb{I}\left\{\sup_{f \in \widehat{\mathcal{V}}_\ell, a \in \mathcal{A}} |(\mathcal{P}_{\ell-1}[\widehat{V}_\ell] - \mathcal{P}_{\ell-1}[f_\ell])(\boldsymbol{x}_{\ell-1}^{(k+i)}, a)| > 3\epsilon\right\}.$$

Now, let $\mathcal{E}''_{h,\ell} \coloneqq \bigcap_{k \in [\boldsymbol{T}_\ell - N_{\text{test}}+1]} \mathcal{E}''_{h,\ell,k}$. By the union bound and the fact that $\sum_{k \ge 1} 1/k^2 = \pi^2/6 \le 2$, we have that $\mathbb{P}[\mathcal{E}''_{h,\ell}] \ge 1 - \delta/(2H^2)$. Furthermore, under $\mathcal{E}''_{h,\ell}$, we have

$$\forall k \in [\boldsymbol{T}_\ell - N_{\text{test}} + 1],$$

$$\sum_{0 \le i < N_{\text{test}}} \mathbb{P}\left[\sup_{f \in \widehat{\mathcal{V}}_\ell} \max_{a \in \mathcal{A}} \big|(\mathcal{P}_{\ell-1}[\widehat{V}_\ell] - \mathcal{P}_{\ell-1}[f_\ell])(\boldsymbol{x}_{\ell-1}^{(k+i)}, a)\big| > 3\varepsilon\right]$$

$$\le 4\log(8H^2 \boldsymbol{T}_\ell^2/\delta) + \sum_{0 \le i < N_{\text{test}}} \mathbb{I}\left\{\sup_{f \in \widehat{\mathcal{V}}_\ell, a \in \mathcal{A}} |(\mathcal{P}_{\ell-1}[\widehat{V}_\ell] - \mathcal{P}_{\ell-1}[f_\ell])(\boldsymbol{x}_{\ell-1}^{(k+i)}, a)| > 3\epsilon\right\}. \qquad (36)$$

We define $\mathcal{E}'_h \coloneqq \mathcal{E}'_{h,1} \cap \mathcal{E}''_{h,1} \cap \cdots \cap \mathcal{E}'_{h,H} \cap \mathcal{E}''_{h,H}$. Note that by the union bound, we have $\mathbb{P}[\mathcal{E}'_h] \ge 1 - \frac{\delta}{H}$ as desired.

**Termination of RVFS$_h$ under $\mathcal{E} \cap \mathcal{E}'_h$.** We now show that under $\mathcal{E} \cap \mathcal{E}'_h$, if RVFS$_h$ terminates, its output satisfies (26). For the rest of the proof, we condition on $\mathcal{E} \cap \mathcal{E}'_h$. Suppose that RVFS$_h$ terminates and returns $(\widehat{V}_{h:H}, \widehat{\mathcal{V}}_{h:H}, \mathcal{C}_{h:H}, \mathcal{B}_{h:H}, t_{h:H})$. In this case, the value function $\widehat{V}_\ell$ must have passed the tests in Line 14 for all $(x_{h-1}, a_{h-1}) \in \mathcal{C}_h$, $n \in N_{\text{test}}$, and $a_{\ell-1} \in \mathcal{A}$. Fix $(x_{h-1}, a_{h-1}) \in \mathcal{C}_h$ and let $k \in [\boldsymbol{T}_\ell - N_{\text{test}} \cdot A + 1]$ be such that $(\boldsymbol{x}_{\ell-1}^{k+j})_{j \in [0..N_{\text{test}}-1]}$ represents a subsequence of states that pass the tests in Line 14 at layer $\ell$ for $(x_{h-1}, a_{h-1})$ within the call to RVFS$_h$. The fact that the sequence $(\boldsymbol{x}_{\ell-1}^{(i)})_{i \ge 1}$ is ordered (see definition in the first paragraph of this proof) and that $(\boldsymbol{x}_{\ell-1}^{k+j})_{j \in [0..N_{\text{test}}-1]}$ pass the tests imply that

1. The states $(\boldsymbol{x}_{\ell-1}^{(k+i)})_{i \in [0..N_{\text{test}}-1]}$ at layer $\ell - 1$ are i.i.d., and are obtained by rolling out with $\widehat{\pi}_{h:H}$ starting from $(x_{h-1}, a_{h-1})$; and

2. The test in Line 14 succeeds for all $(\boldsymbol{x}_{\ell-1}^{(k+j)})_{j \in [0..N_{\text{test}}-1]}$; that is

$$\forall j \in [0..N_{\text{test}}-1], \forall a_{\ell-1} \in \mathcal{A}, \quad \sup_{f \in \widehat{\mathcal{V}}_\ell} |(\widehat{\boldsymbol{\mathcal{P}}}_{\ell-1,\varepsilon,\delta'}[\widehat{V}_\ell] - \widehat{\boldsymbol{\mathcal{P}}}_{\ell-1,\varepsilon,\delta'}[f_\ell])(\boldsymbol{x}_{\ell-1}^{(k+j)}, a_{\ell-1})|$$

$$\le \varepsilon + \varepsilon \cdot \beta(k+j),$$

$$\le \varepsilon + \varepsilon \cdot \sqrt{2\log_{1/\delta'}(8AM|\mathcal{V}|(k+j)^2/\delta)},$$

$$\le \varepsilon + \varepsilon \cdot \sqrt{2\log_{1/\delta'}(8AM|\mathcal{V}|\boldsymbol{T}_\ell^2/\delta)}.$$

This implies that

$$\forall i \in [0 \, .. \, N_{\text{test}} - 1], \forall a_{\ell-1} \in \mathcal{A} :$$

$$\sup_{f \in \widehat{\mathcal{V}}_\ell} |\mathcal{P}_{\ell-1}[\widehat{V}_\ell - f_\ell](\boldsymbol{x}_{\ell-1}^{(k+i)}, a_{\ell-1})| - 3\varepsilon$$

$$\leq \sup_{f \in \widehat{\mathcal{V}}_\ell} |\mathcal{P}_{\ell-1}[\widehat{V}_\ell - f_\ell](\boldsymbol{x}_{\ell-1}^{(k+i)}, a_{\ell-1})| - \varepsilon - 2\varepsilon \cdot \sqrt{2\log_{1/\delta'}(4AMH^2|\mathcal{V}|\boldsymbol{T}_\ell^2\delta)}, \tag{37}$$

$$\overset{(35)}{\leq} \sup_{f \in \widehat{\mathcal{V}}_\ell} |(\widehat{\boldsymbol{\mathcal{P}}}_{\ell-1,\varepsilon,\delta'}[\widehat{V}_\ell] - \widehat{\boldsymbol{\mathcal{P}}}_{\ell-1,\varepsilon,\delta'}[f_\ell])(\boldsymbol{x}_{\ell-1}^{(k+i)}, a_{\ell-1})| - \varepsilon - \varepsilon \cdot \sqrt{2\log_{1/\delta'}(4AMH^2|\mathcal{V}|\boldsymbol{T}_\ell^2/\delta)},$$

$$\leq 0. \quad \text{(by Item 2)} \tag{38}$$

where (37) follows by (34) and the choice of $\delta'$ in Algorithm 5.

Now, by Item 1, we have that $\boldsymbol{x}_{\ell-1}^{(k+i)}$ has probability law $\mathbb{P}^{\widehat{\pi}_{h:H}}[\cdot \mid \boldsymbol{x}_{h-1} = x_{h-1}, \boldsymbol{a}_{h-1} = a_{h-1}]$ for all $i \in [0 \, .. \, N_{\text{test}} - 1]$, and so by (36), we have:

$$\mathbb{P}^{\widehat{\pi}}\left[ \sup_{f \in \widehat{\mathcal{V}}_\ell} \max_{a \in \mathcal{A}} |(\mathcal{P}_{\ell-1}[\widehat{V}_\ell] - \mathcal{P}_{\ell-1}[f_\ell])(\boldsymbol{x}_{\ell-1}, a)| > 3\varepsilon \mid \boldsymbol{x}_{h-1} = x_{h-1}, \boldsymbol{a}_{h-1} = a_{h-1} \right]$$

$$\leq \frac{4\log(8H^2\boldsymbol{T}_\ell^2/\delta)}{N_{\text{test}}} + \frac{1}{N_{\text{test}}} \sum_{0 \leq i < N_{\text{test}}} \mathbb{I}\left\{ \sup_{f \in \widehat{\mathcal{V}}_\ell, a \in \mathcal{A}} |\mathcal{P}_{\ell-1}[\widehat{V}_\ell] - \mathcal{P}_{\ell-1}[f_\ell])(\boldsymbol{x}_{\ell-1}^{(k+i)}, a)| > 3\epsilon \right\},$$

$$\leq \frac{4\log(8M^6N_{\text{test}}^2H^8/\delta)}{N_{\text{test}}} \quad \text{(using (34) and the fact that all the tests pass, i.e. (38)).}$$

**Concluding.** We have established that under $\mathcal{E} \cap \mathcal{E}_h'$, we have for all $\ell \in [h+1 \, .. \, H]$ and all $(x_{h-1}, a_{h-1}) \in \mathcal{C}_h$:

$$\mathbb{P}^{\widehat{\pi}}\left[ \sup_{f \in \widehat{\mathcal{V}}_\ell} \max_{a \in \mathcal{A}} |(\mathcal{P}_{\ell-1}[\widehat{V}_\ell] - \mathcal{P}_{\ell-1}[f_\ell])(\boldsymbol{x}_{\ell-1}, a)| > 3\varepsilon \mid \boldsymbol{x}_{h-1} = x_{h-1}, \boldsymbol{a}_{h-1} = a_{h-1} \right] \leq \frac{4\log(8M^6N_{\text{test}}^2H^8/\delta)}{N_{\text{test}}},$$

Furthermore, we have $\mathbb{P}[\mathcal{E}_h'] \geq 1 - \delta/H$. This completes the proof. $\qquad \square$

## I.4 Proof of Lemma I.3 (Value Function Regression Guarantee)

**Proof of Lemma I.3.** Fix $\pi \in \Pi' \subseteq \Pi_S$ and $k \geq 1$, and consider the $k$th call to $\mathsf{RVFS}_h$ as per the lemma statement, and let $\mathcal{S}_k$ be the state of $\mathsf{RVFS}_0$ during the $k^{\text{th}}$ call to $\mathsf{RVFS}_h$ and immediately before Line 20, i.e. immediately before gathering data for the regression step in $\mathsf{RVFS}_h$.

**Relating the regression targets to $V_h^{\widehat{\pi}}$.** Observe that $\widehat{V}_h$ is the least-squares solution of the objective in Line 26, where the targets are empirical estimates of $V_h^{\widehat{\pi}}$. In particular, if we let $\{\mathcal{D}_h(x, a) : (x, a) \in \mathcal{C}_h\}$ be the datasets in the definition of $\widehat{\mathcal{V}}_h$ in (15), then for any $(x_{h-1}, a_{h-1}) \in \mathcal{C}_h$ and $(x_h, v_h) \in \mathcal{D}_h(x_{h-1}, a_{h-1})$, the target $v_h$ satisfies

$$v_h = \widehat{V}_h(x_h),$$

where $\widehat{V}_h(x_h)$ is an empirical estimate of $V_h^{\widehat{\pi}}(x_h)$ obtained by sampling $N_{\text{est}}(|\mathcal{C}_h|) = N_{\text{est}}(k)$ episodes (for $N_{\text{est}}(\cdot)$ defined as in Algorithm 5) by rolling out with $\widehat{\pi}$ after starting from $x_h$ and playing action $a$ at layer $h$; note that $|\mathcal{C}_h| = k$ because we are considering the $k$th call to $\mathsf{RVFS}_h$. Thus, by Hoeffding's inequality and the union bound over $(x_{h-1}, a_{h-1}) \in \mathcal{C}_h$ and $(x_h, -) \in \mathcal{D}_h(x_{h-1}, a_{h-1})$, there is an event $\mathcal{E}_{h,k}''(\mathcal{S}_k)$ of probability at least $1 - \delta/(8k^2H)$ under which

$$\forall (x_{h-1}, a_{h-1}) \in \mathcal{C}_h, \forall (x_h, -) \in \mathcal{D}_h(x_{h-1}, a_{h-1}) :$$

$$|V_h^{\widehat{\pi}}(x_h) - \widehat{V}_h(x_h)| \leq H\sqrt{\frac{2\log(8|\mathcal{C}_h|N_{\text{reg}}Hk^2/\delta)}{N_{\text{est}}(k)}} \leq H\sqrt{\frac{2\log(8N_{\text{reg}}Hk^3/\delta)}{N_{\text{est}}(k)}}, \tag{39}$$

where $N_{\text{reg}}$ is as in Line 5, and the last inequality follows by $|\mathcal{C}_h| \leq k$ since we are considering the $k$th call to $\mathsf{RVFS}_h$. Thus, under $\mathcal{E}''_{h,k}$, we have

$$\forall (x_{h-1}, a_{h-1}) \in \mathcal{C}_h, \forall (x_h, v_h) \in \mathcal{D}_h(x_{h-1}, a_{h-1}):$$

$$\left| V_h^{\widehat{\pi}}(x_h) - v_h \right| = \left| V_h^{\widehat{\pi}}(x_h) - \widehat{V}_h(x_h) \right| \leq H \sqrt{\frac{2 \log(8 N_{\text{reg}} H k^3 / \delta)}{N_{\text{est}}(k)}} \leq \frac{H}{N_{\text{reg}}}, \tag{40}$$

where the second-to-last inequality is by (39) and the last inequality follows by the choice of $N_{\text{est}}$ in Algorithm 5.

**Bounding the discrepancy $V_h^{\widehat{\pi}} - V_h^{\pi}$.** On the other hand, by the performance difference lemma, the value function $V_h^{\widehat{\pi}}$ satisfies:

$$\forall x \in \mathcal{X}, \quad |V_h^{\widehat{\pi}}(x) - V_h^{\pi}(x)| \leq \sum_{\tau=h}^{H} \mathbb{E}^{\widehat{\pi}} \left[ |Q_\tau^{\pi}(\boldsymbol{x}_\tau, \boldsymbol{\pi}_\tau(\boldsymbol{x}_\tau)) - Q_\tau^{\pi}(\boldsymbol{x}_\tau, \widehat{\boldsymbol{\pi}}_\tau(\boldsymbol{x}_\tau))| \mid \boldsymbol{x}_h = x \right],$$

$$\leq H \sum_{\tau=h}^{H} \mathbb{E}^{\widehat{\pi}} \left[ D_{\text{tv}}(\widehat{\pi}_\tau(\boldsymbol{x}_\tau), \pi_\tau(\boldsymbol{x}_\tau)) \mid \boldsymbol{x}_h = x \right]. \tag{41}$$

Now, let $(x_{h-1}^{(1)}, a_{h-1}^{(1)}), (x_{h-1}^{(2)}, a_{h-1}^{(2)}), \dots$ denote the elements of $\mathcal{C}_h$ in the order in which they are added to the latter in Line 15. By Lemma C.2 (Freedman's inequality) instantiated with

- $n = N_{\text{reg}} \cdot k$.

- $\boldsymbol{w}_i = |V_h^{\pi}(\boldsymbol{x}_h^{(i)}) - V_h^{\widehat{\pi}}(\boldsymbol{x}_h^{(i)})| - \mathbb{E}[|V_h^{\pi}(\boldsymbol{x}_h) - V_h^{\widehat{\pi}}(\boldsymbol{x}_h)| \mid \boldsymbol{x}_{h-1} = x_{h-1}^{(j)}, \boldsymbol{a}_{h-1} = a_{h-1}^{(j)}]$, for all $i \in [n]$ and $j = \lfloor i/N_{\text{reg}} \rfloor + 1$, where $\boldsymbol{x}_h^{(N_{\text{reg}} \cdot j)}, \dots, \boldsymbol{x}_h^{(N_{\text{reg}} \cdot j + N_{\text{reg}} - 1)} \overset{\text{i.i.d.}}{\sim} T_h(\cdot \mid \boldsymbol{x}_{h-1} = x_{h-1}^{(j)}, \boldsymbol{a}_{h-1} = a_{h-1}^{(j)})$;

- $\mathcal{H}_i = \sigma(\boldsymbol{x}_h^{(1)}, \dots \boldsymbol{x}_h^{(i-1)})$, for all $i \in [n]$;

- $R = H$; and

- $\lambda = 1/H$;

we get that there is an event $\widetilde{\mathcal{E}}''_{h,k,\pi}(\mathcal{S}_k)$ of probability at least $1 - \delta/(8k^2 H |\Pi'|)$ under which:

$$\sum_{(x_{h-1}, a_{h-1}) \in \mathcal{D}_h} \sum_{(x_h, -) \in \mathcal{D}_h(x_{h-1}, a_{h-1})} |V_h^{\pi}(x_h) - V_h^{\widehat{\pi}}(x_h)|$$

$$= 2 N_{\text{reg}} \sum_{(x_{h-1}, a_{h-1}) \in \mathcal{D}_h} \mathbb{E} \left[ |V_h^{\pi}(\boldsymbol{x}_h) - V_h^{\widehat{\pi}}(\boldsymbol{x}_h)| \mid \boldsymbol{x}_{h-1} = x_{h-1}, \boldsymbol{a}_{h-1} = a_{h-1} \right] + H \log(8k^2 |\Pi'| H/\delta),$$

$$\leq 2 H N_{\text{reg}} \sum_{(x_{h-1}, a_{h-1}) \in \mathcal{D}_h} \sum_{\tau=h}^{H} \mathbb{E}^{\widehat{\pi}} \left[ D_{\text{tv}}(\widehat{\pi}_\tau(\boldsymbol{x}_\tau), \pi_\tau(\boldsymbol{x}_\tau)) \mid \boldsymbol{x}_{h-1} = x_{h-1}, \boldsymbol{a}_{h-1} = a_{h-1} \right] + H \log(8k^2 |\Pi'| H/\delta),$$

$$\tag{42}$$

where the last inequality follows by (41) and the law of total expectation.

**Regression guarantee.** Since $\pi \in \Pi' \subseteq \Pi_S$ and Assumption I.1 holds, Lemma C.5 (regression guarantee) instantiated with

- $f_\star(x) = V_h^{\pi}(x)$;

- $B = H$;

- $\boldsymbol{b}_i = \boldsymbol{v}_h - V_h^{\pi}(\boldsymbol{x}_h)$ (where $\boldsymbol{v}_h := \max_{a \in \mathcal{A}} \widehat{Q}_h(\boldsymbol{x}_h, a)$); and

- $\xi = H$;

implies that there is an event $\breve{\mathcal{E}}''_{h,k,\pi}(\mathcal{S}_k)$ of probability at least $1 - \delta/(4k^2 H|\Pi'|)$ under which we have:

$$\sum_{(x_{h-1},a_{h-1})\in\mathcal{C}_h} \frac{1}{N_{\text{reg}}} \sum_{(x_h,-)\in\mathcal{D}_h(x_{h-1},a_{h-1})} \left(\widehat{V}_h(x_h) - V_h^\pi(x_h)\right)^2$$

$$\leq \frac{4kH^2 \log(4k^2 H|\Pi'||\mathcal{V}|/\delta)}{N_{\text{reg}}} + \frac{4H}{N_{\text{reg}}} \sum_{(x_{h-1},a_{h-1})\in\mathcal{C}_h} \sum_{(x_h,v_h)\in\mathcal{D}_h(x_{h-1},a_{h-1})} |V_h^\pi(x_h) - v_h|,$$

$$\leq \frac{4kH^2 \log(4k^2 H|\Pi'||\mathcal{V}|/\delta)}{N_{\text{reg}}} + \frac{4H}{N_{\text{reg}}} \sum_{(x_{h-1},a_{h-1})\in\mathcal{C}_h} \sum_{(x_h,v_h)\in\mathcal{D}_h(x_{h-1},a_{h-1})} |V_h^{\widehat{\pi}}(x_h) - v_h|$$

$$+ \frac{4H}{N_{\text{reg}}} \sum_{(x_{h-1},a_{h-1})\in\mathcal{C}_h} \sum_{(x_h,v_h)\in\mathcal{D}_h(x_{h-1},a_{h-1})} |V_h^\pi(x_h) - V_h^{\widehat{\pi}}(x_h)|, \tag{43}$$

where the last step follows by the triangle inequality. Thus, by plugging (42) and (40) into (43), we get that under $\mathcal{E}''_{h,k}(\mathcal{S}_k) \cap \widetilde{\mathcal{E}}''_{h,k,\pi}(\mathcal{S}_k) \cap \breve{\mathcal{E}}''_{h,k,\pi}(\mathcal{S}_k)$:

$$\sum_{(x_{h-1},a_{h-1})\in\mathcal{C}_h} \frac{1}{N_{\text{reg}}} \sum_{(x_h,-)\in\mathcal{D}_h(x_{h-1},a_{h-1})} \left(\widehat{V}_h(x_h) - V_h^\pi(x_h)\right)^2$$

$$\leq \frac{9kH^2 \log(8k^2 H|\Pi'||\mathcal{V}|/\delta)}{N_{\text{reg}}} + 8H^2 \sum_{(x_{h-1},a_{h-1})\in\mathcal{C}_h} \sum_{\tau=h}^{H} \mathbb{E}^{\widehat{\pi}} \left[ D_{\text{tv}}\left(\widehat{\pi}_\tau(\boldsymbol{x}_\tau), \pi_\tau(\boldsymbol{x}_\tau)\right) \mid \boldsymbol{x}_{h-1} = x_{h-1}, \boldsymbol{a}_{h-1} = a_{h-1} \right]. \tag{44}$$

**Applying the union bound to conclude.** Let $\mathcal{S}_k$ be the random state of $\mathsf{RVFS}_0$ during the $k^{\text{th}}$ call to $\mathsf{RVFS}_h$ and immediately before Line 20, i.e. immediately before gathering data for the regression step in $\mathsf{RVFS}_h$. Further, let $\mathcal{S}_k^+$ be the random state of $\mathsf{RVFS}_0$ during the $k^{\text{th}}$ call to $\mathsf{RVFS}_h$ and immediately before Line 26, i.e. immediately before the regression step in $\mathsf{RVFS}_h$. If $\mathsf{RVFS}_0$ terminates before the $k^{th}$ call to $\mathsf{RVFS}_h$, we use the convention that $\mathcal{S}_k = \mathcal{S}_k^+ = \mathfrak{t}$, where $\mathfrak{t}$ denotes a terminal state, and define $\mathcal{E}''_{h,k}(\mathfrak{t}) = \widetilde{\mathcal{E}}''_{h,k,\pi}(\mathfrak{t}) = \breve{\mathcal{E}}''_{h,k,\pi}(\mathfrak{t}) = \{\mathfrak{t}\}$. Further, we define

$$\mathcal{E}''_h := \left\{ \prod_{k\in\mathbb{N},\pi\in\Pi'} \mathbb{I}\{\boldsymbol{\mathcal{S}}_k^+ \in \mathcal{E}''_{h,k}(\boldsymbol{\mathcal{S}}_k) \cap \widetilde{\mathcal{E}}''_{h,k,\pi}(\boldsymbol{\mathcal{S}}_k) \cap \breve{\mathcal{E}}''_{h,k,\pi}(\boldsymbol{\mathcal{S}}_k)\} = 1 \right\}.$$

Note that by the argument in the sequel and the union bound, we have that

$$\forall k \geq 1, \forall \mathcal{S}_k, \quad \mathbb{P}[\boldsymbol{\mathcal{S}}_k^+ \in \mathcal{E}''_{h,k}(\mathcal{S}_k) \cap \widetilde{\mathcal{E}}''_{h,k,\pi}(\mathcal{S}_k) \cap \breve{\mathcal{E}}''_{h,k,\pi}(\mathcal{S}_k)] \geq 1 - \frac{\delta}{2k^2 H}, \tag{45}$$

where $\mathcal{S}_k$ denotes the state of $\mathsf{RVFS}_0$ during the $k^{\text{th}}$ call to $\mathsf{RVFS}_h$ and immediately before Line 20. By letting $\boldsymbol{\mathcal{S}}'_1, \boldsymbol{\mathcal{S}}'_2, \ldots$ denote an identical, independent copy of the sequence $\boldsymbol{\mathcal{S}}_1, \boldsymbol{\mathcal{S}}_2, \ldots$, we have by the chain rule:

$$\mathbb{P}[\mathcal{E}''_h] = \mathbb{E}_{\boldsymbol{\mathcal{S}}'_1, \boldsymbol{\mathcal{S}}'_2, \ldots} \left[ \prod_{k\geq 1} \mathbb{P}[\boldsymbol{\mathcal{S}}_k^+ \in \mathcal{E}''_{h,k}(\boldsymbol{\mathcal{S}}_k) \cap \widetilde{\mathcal{E}}''_{h,k,\pi}(\boldsymbol{\mathcal{S}}_k) \cap \breve{\mathcal{E}}''_{h,k,\pi}(\boldsymbol{\mathcal{S}}_k) \mid \boldsymbol{\mathcal{S}}_k = \boldsymbol{\mathcal{S}}'_k] \right],$$

$$\geq \prod_{k\geq 1} \left(1 - \frac{\delta}{2k^2 H}\right), \quad \text{(by (45))}$$

$$\geq 1 - \frac{\delta}{H}, \tag{46}$$

where the last inequality follows from the fact that for any sequence $x_1, x_2, \cdots \in (0,1)$, we have $\prod_{k\geq 1}(1 - x_k) \geq 1 - \sum_{k\geq 1} x_k$. Combining (46) with (44) implies that $\mathcal{E}''_h$ gives the desired result. $\quad\square$

## I.5 Proof of Lemma I.4 (Guarantee for Confidence Sets)

To prove Lemma I.4, we need one additional result pertaining to the order in which the instances $(\mathsf{RVFS}_h)_{h\in[H]}$ are called.

**Lemma I.5.** *Let $h \in [0 .. H]$ be given, and consider the setting of Lemma I.4. Further, consider a call to $\mathrm{RVFS}_0(f, \mathcal{V}, \varnothing, \varnothing; \mathcal{V}, \varepsilon, \delta)$ that terminates, and let $h \in [H]$ be any layer such that $\mathrm{RVFS}_h$ is called during the execution of $\mathrm{RVFS}_0$. Then, after the last call to $\mathrm{RVFS}_h$ terminates, no instance of $\mathrm{RVFS}$ in $(\mathrm{RVFS}_\tau)_{\tau > h}$ is called before $\mathrm{RVFS}_0$ terminates.*

**Proof of Lemma I.5.** Suppose there is an instance of RVFS in $(\mathrm{RVFS}_\tau)_{\tau > h}$ that is called after the last call to $\mathrm{RVFS}_h$ terminates. Let $\tau > h$ be the lowest layer where $\mathrm{RVFS}_\tau$ is called after the last call to $\mathrm{RVFS}_h$ terminates. Let $\mathrm{RVFS}_\tau^{\mathrm{last}}$ denote the corresponding instance of $\mathrm{RVFS}_\tau$. Further, let $\ell < \tau$ be such that $\mathrm{RVFS}_\ell$ is the parent instance of $\mathrm{RVFS}_\tau^{\mathrm{last}}$ (i.e. the instance that called $\mathrm{RVFS}_\tau^{\mathrm{last}}$). Note that we cannot have $\ell = h$ as this would imply that an instance of $\mathrm{RVFS}_h$ terminates after $\mathrm{RVFS}_\tau^{\mathrm{last}}$, and we have assumed that $\mathrm{RVFS}_\tau^{\mathrm{last}}$ terminates after the last call $\mathrm{RVFS}_h$. It is also not possible to have $\ell > h$ as this would imply that $\tau$ is not the lowest layer where $\mathrm{RVFS}_\tau$ is called after the last call to $\mathrm{RVFS}_h$ terminates. Thus, we must have that $\ell < h$. Now, the for-loop in Line 16 ensures that that there is an instance of $\mathrm{RVFS}_h$ that is called after $\mathrm{RVFS}_\tau^{\mathrm{last}}$ terminates and before $\mathrm{RVFS}_\ell$ does. This contradicts the assumption that $\mathrm{RVFS}_\tau^{\mathrm{last}}$ is called after the last call to $\mathrm{RVFS}_h$. $\qquad\square$

**Proof of Lemma I.4.** We start by showing that $\tilde\pi \in \Pi_{4\varepsilon}$ by constructing the corresponding collection of random state-action value functions $(\widehat{\boldsymbol{Q}}_h(x,a))_{(h,x,a) \in [H] \times \mathcal{X} \times \mathcal{A}} \subseteq [0, H]$ in the definition of $\Pi_{4\varepsilon}$. In particular, for $(h,x,a) \in [H] \times \mathcal{X} \times \mathcal{A}$, we define

$$\widetilde{\boldsymbol{Q}}_h(x,a) = \begin{cases} \widehat{\boldsymbol{Q}}_h(x,a), & \text{if } \|\widehat{\boldsymbol{Q}}_h(x,\cdot) - \mathcal{P}_h[V_{h+1}^{\tilde\pi}](x,\cdot)\|_\infty \le 4\varepsilon, \\ \mathcal{P}_h[V_{h+1}^{\tilde\pi}](x,a), & \text{otherwise,} \end{cases} \quad \text{for } h = H, \ldots, 1.$$

where $\widehat{\boldsymbol{Q}}_\tau(x,a) \coloneqq \widehat{\boldsymbol{\mathcal{P}}}_{\tau,\varepsilon,\delta'}[\widehat{V}_{\tau+1}](x,a)$. Note that $\widetilde{\boldsymbol{Q}}_h(x,a)$ only depends on the randomness of $\widehat{\boldsymbol{\mathcal{P}}}_{h,\varepsilon,\delta'}[\widehat{V}_{h+1}](x,a)$, and so $(\widetilde{\boldsymbol{Q}}_h(x,a))_{(h,x,a) \in [H] \times \mathcal{X} \times \mathcal{A}}$ are independent random variables. Furthermore, since $\mathcal{P}_h[V_{h+1}^{\tilde\pi}] \equiv Q_h^{\tilde\pi}$, we have that

$$\forall (x,a) \in \mathcal{X} \times \mathcal{A}, \quad \|\widetilde{\boldsymbol{Q}}_h(x,a) - Q_h^{\tilde\pi}(x,a)\| \le 4\varepsilon.$$

Finally, since $\tilde{\boldsymbol{\pi}}_h(\cdot) \in \arg\max_{a \in \mathcal{A}} \widetilde{\boldsymbol{Q}}_h(\cdot, a)$, we have that $\tilde\pi \in \Pi_{4\varepsilon}$.

**We show $V_h^{\tilde\pi} \in \widehat{\mathcal{V}}_h$.** We prove that $V_h^{\tilde\pi} \in \widehat{\mathcal{V}}_h$, for all $h \in [H]$, under the event $\mathcal{E}''' \coloneqq \mathcal{E} \cap \mathcal{E}_1' \cap \mathcal{E}_1'' \cap \cdots \cap \mathcal{E}_H' \cap \mathcal{E}_H''$, where $\mathcal{E}, (\mathcal{E}_h')$, and $(\mathcal{E}_h'')$ are the events defined in Lemma I.1, Lemma I.2, and Lemma I.3, respectively. Throughout, we condition on $\mathcal{E}'''$. First, note that by Lemma I.1, $\mathrm{RVFS}_0$ terminates. Let $(\widehat{V}_{1:H}, \widehat{\mathcal{V}}_{1:H}, \mathcal{C}_{1:H}, \mathcal{B}_{1:H}, t_{1:H})$ be the tuple it returns.

We will show via backwards induction over $\ell = H + 1, \ldots, 1$, that

$$V_\ell^{\tilde\pi} \in \widehat{\mathcal{V}}_\ell, \tag{47}$$

where $\tilde\pi_{1:H}$ is the stochastic policy defined recursively via

$$\tilde{\boldsymbol{\pi}}_\tau(x) \in \arg\max_{a \in \mathcal{A}} \begin{cases} \widehat{\boldsymbol{Q}}_\tau(x,a) \coloneqq \widehat{\boldsymbol{\mathcal{P}}}_{\tau,\varepsilon,\delta'}[\widehat{V}_{\tau+1}](x,a), & \text{if } \|\widehat{\boldsymbol{Q}}_\tau(x,\cdot) - \mathcal{P}_\tau[V_{\tau+1}^{\tilde\pi}](x,\cdot)\|_\infty \le 4\varepsilon, \\ \mathcal{P}_\tau[V_{\tau+1}^{\tilde\pi}](x,a), & \text{otherwise,} \end{cases} \quad \text{for } \tau = H, \ldots, 1,$$

where $\widehat{\boldsymbol{Q}}_\tau(x,a) \coloneqq \widehat{\boldsymbol{\mathcal{P}}}_{\tau,\varepsilon,\delta'}[\widehat{V}_{\tau+1}](x,a)$.

**Base case [$\ell = H + 1$].** This holds trivially because $V_{H+1}^\pi \equiv 0$ for any $\pi \in \Pi_{\mathrm{S}}$ by convention.

**General case [$\ell \le H$].** Fix $h \in [H]$ and suppose that (47) holds for all $\ell \in [h + 1 .. H + 1]$. We show as a consequence that (47) holds for $\ell = h$. First, note that if $\mathrm{RVFS}_h$ is never called during the execution of $\mathrm{RVFS}_0$, then $\widehat{\mathcal{V}}_h = \mathcal{V}$, and so (47) trivially holds for $\ell = h$ under Assumption I.1 with $\varepsilon_{\mathrm{real}} = 4\varepsilon$.

Now, suppose that $\mathrm{RVFS}_h$ is called at least once, and let $(\widehat{V}_{h:H}^+, \widehat{\mathcal{V}}_{h:H}^+, \mathcal{C}_{h:H}^+, \mathcal{B}_{h:H}^+, t_{h:H}^+)$ be the output of the last call to $\mathrm{RVFS}_h$ during the execution of $\mathrm{RVFS}_0$. We claim that

$$(\widehat{V}_{h:H}^+, \widehat{\mathcal{V}}_{h:H}^+, \mathcal{C}_{h:H}^+) = (\widehat{V}_{h:H}, \widehat{\mathcal{V}}_{h:H}, \mathcal{C}_{h:H}). \tag{48}$$

To see this, first note that the for-loop in Line 16 ensures that no instance of $(\mathrm{RVFS}_\tau)_{\tau > h}$ can be called after the last call to $\mathrm{RVFS}_h$ (by Lemma I.5). Thus, the estimated value functions, confidence sets, and core sets for layers $h + 1, \ldots, H$ remain unchanged after the last call to $\mathrm{RVFS}_h$; that is, (48) holds.

Thus, by Lemma I.2, and since we are conditioning on $\mathcal{E}'_{h+1:H}$, we have that for all $(x_{h-1}, a_{h-1}) \in \mathcal{C}_h$ and $\ell \in [h+1 .. H+1]$:

$$\mathbb{P}^{\widehat{\pi}}\left[\sup_{f \in \widehat{\mathcal{V}}_\ell} \max_{a \in \mathcal{A}} \left|(\mathcal{P}_{\ell-1}[\widehat{V}_\ell] - \mathcal{P}_{\ell-1}[f_\ell])(\boldsymbol{x}_{\ell-1}, a)\right| > 3\varepsilon \mid \boldsymbol{x}_{h-1} = x_{h-1}, \boldsymbol{a}_{h-1} = a_{h-1}\right] \le \frac{4\log(8M^6 N_{\text{test}}^2 H^8/\delta)}{N_{\text{test}}},$$
(49)

where $M = \lceil 8\varepsilon^{-1} C_{\text{push}} H \rceil$. Now, by the induction hypothesis, we have $V_\ell^{\tilde{\pi}} \in \widehat{\mathcal{V}}_\ell$, and so substituting $V_\ell^{\tilde{\pi}}$ for $f_\ell$ in (49), we get that for all $(x_{h-1}, a_{h-1}) \in \mathcal{C}_h$ and $\ell \in [h+1 .. H+1]$:

$$\mathbb{P}^{\widehat{\pi}}\left[\max_{a \in \mathcal{A}} \left|(\mathcal{P}_{\ell-1}[\widehat{V}_\ell] - \mathcal{P}_{\ell-1}[V_\ell^{\tilde{\pi}}])(\boldsymbol{x}_{\ell-1}, a)\right| > 3\varepsilon \mid \boldsymbol{x}_{h-1} = x_{h-1}, \boldsymbol{a}_{h-1} = a_{h-1}\right] \le \frac{4\log(8M^6 N_{\text{test}}^2 H^8/\delta)}{N_{\text{test}}}.$$

Therefore, by Lemma L.1 (instantiated with $\mu[\cdot] = \mathbb{P}^{\widehat{\pi}}[\cdot \mid \boldsymbol{x}_{h-1} = x_{h-1}, \boldsymbol{a}_{h-1} = a_{h-1}]$, $\tau = \ell - 1$, and $V_{\tau+1} = V_\ell^{\tilde{\pi}}$), we have that $(x_{h-1}, a_{h-1}) \in \mathcal{C}_h$ and $\ell \in [h+1 .. H+1]$:

$$\mathbb{E}^{\widehat{\pi}}\left[D_{\text{tv}}(\widehat{\pi}_{\ell-1}(\boldsymbol{x}_{\ell-1}), \tilde{\pi}_{\ell-1}(\boldsymbol{x}_{\ell-1})) \mid \boldsymbol{x}_{h-1} = x_{h-1}, \boldsymbol{a}_{h-1} = a_{h-1}\right] \le \frac{4\log(8M^6 N_{\text{test}}^2 H^8/\delta)}{N_{\text{test}}} + \delta',$$
(50)

where $\delta'$ is as in Algorithm 5.

**Applying the regression guarantee to conclude the induction.** Note that $\tilde{\pi} \in \Pi'$, where $\Pi' \subset \Pi_{\text{S}}$ is the set of stochastic policies such that $\pi \in \Pi'$ if and only if there exists $V_{1:H} \in \mathcal{V}$ such that $\pi$ is defined recursively as

$$\boldsymbol{\pi}_\tau(x) \in \arg\max_{a \in \mathcal{A}} \begin{cases} \boldsymbol{Q}_\tau(x, a) \coloneqq \widehat{\boldsymbol{\mathcal{P}}}_{\tau, \varepsilon, \delta'}[V_{\tau+1}](x, a), & \text{if } \|\boldsymbol{Q}_\tau(x, \cdot) - \mathcal{P}_\tau[V_{\tau+1}^\pi](x, \cdot)\|_\infty \le 4\varepsilon, \\ \mathcal{P}_\tau[V_{\tau+1}^\pi](x, a), & \text{otherwise,} \end{cases}$$

for $\tau = H, \ldots, 1$, where $\boldsymbol{Q}_\tau(x, a) \coloneqq \widehat{\boldsymbol{\mathcal{P}}}_{\tau, \varepsilon, \delta'}[V_{\tau+1}](x, a)$. The policy class $\Pi'$ is finite and $|\Pi'| \le |\mathcal{V}|$. Furthermore, we have $\Pi' \subseteq \Pi_{4\varepsilon}$ as shown at the beginning of this proof. Therefore, if we let $\{\mathcal{D}_h(x, a) : (x, a) \in \mathcal{C}_h\}$ be the datasets in the definition of $\widehat{\mathcal{V}}_h$ in (15), we have by Lemma I.3 (under Assumption I.1 with $\varepsilon_{\text{real}} = 4\varepsilon$) and the conditioning on $\mathcal{E}''_{h+1:H}$ and $\mathcal{E}$:

$$\sum_{(x_{h-1}, a_{h-1}) \in \mathcal{C}_h} \frac{1}{N_{\text{reg}}} \sum_{(x_h, -) \in \mathcal{D}_h(x_{h-1}, a_{h-1})} \left(\widehat{V}_h(x_h) - V_h^{\tilde{\pi}}(x_h)\right)^2$$

$$\le \frac{9|\mathcal{C}_h|H^2 \log(8|\mathcal{C}_h|^2 H |\mathcal{V}|^2/\delta)}{N_{\text{reg}}} + 8H^2 \sum_{(x_{h-1}, a_{h-1}) \in \mathcal{C}_h} \sum_{\tau=h}^H \mathbb{E}^{\widehat{\pi}}\left[D_{\text{tv}}(\widehat{\pi}_\tau(\boldsymbol{x}_\tau), \tilde{\pi}_\tau(\boldsymbol{x}_\tau)) \mid \boldsymbol{x}_{h-1} = x_{h-1}, \boldsymbol{a}_{h-1} a_{h-1}\right],$$

$$\le \frac{9MH^2 \log(8M^2 H |\mathcal{V}|^2/\delta)}{N_{\text{reg}}} + 8H^2 \sum_{(x_{h-1}, a_{h-1}) \in \mathcal{C}_h} \sum_{\tau=h}^H \mathbb{E}^{\widehat{\pi}}\left[D_{\text{tv}}(\widehat{\pi}_\tau(\boldsymbol{x}_\tau), \tilde{\pi}_\tau(\boldsymbol{x}_\tau)) \mid \boldsymbol{x}_{h-1} = x_{h-1}, \boldsymbol{a}_{h-1} = a_{h-1}\right],$$
(51)

where the last inequality follows by the fact that $|\mathcal{C}_h| \le M$ under $\mathcal{E}$. Combining (51) with (50), implies that

$$\sum_{(x_{h-1}, a_{h-1}) \in \mathcal{C}_h} \frac{1}{N_{\text{reg}}} \sum_{(x_h, -) \in \mathcal{D}_h(x_{h-1}, a_{h-1})} \left(\widehat{V}_h(x_h) - V_h^{\tilde{\pi}}(x_h)\right)^2$$

$$\le \frac{9MH^2 \log(8M^2 H |\mathcal{V}|^2/\delta)}{N_{\text{reg}}} + 8MH^3 \cdot \frac{4\log(8M^6 N_{\text{test}}^2 H^8/\delta)}{N_{\text{test}}} + 8MH^3 \delta',$$

$$= \frac{9MH^2 \log(8M^2 H |\mathcal{V}|^2/\delta)}{N_{\text{reg}}} + 8MH^3 \cdot \frac{4\log(8M^6 N_{\text{test}}^2 H^8/\delta)}{N_{\text{test}}} + 8MH^3 \frac{\delta}{4M^7 N_{\text{test}}^2 H^8 |\mathcal{V}|},$$

$$\le \varepsilon_{\text{reg}}^2,$$
(52)

where the last inequality follows by the fact that $\delta \in (0, 1)$ and the definition of $\varepsilon_{\text{reg}}^2$ in Algorithm 5. By the definition of $\widehat{\mathcal{V}}_h$ in (15), (52) implies that $V_h^{\tilde{\pi}} \in \widehat{\mathcal{V}}_h$, which completes the induction. $\qquad\square$

## I.6 Proof of Theorem I.1 (Main Guarantee of RVFS)

**Proof of Theorem I.1.** We condition on the event $\widetilde{\mathcal{E}} \coloneqq \mathcal{E} \cap \mathcal{E}''' \cap \mathcal{E}'_1 \cap \cdots \cap \mathcal{E}'_H$, where $\mathcal{E}$, $\mathcal{E}'''$, and $(\mathcal{E}'_h)$ are the events in Lemma I.1, Lemma I.4, and Lemma I.2, respectively. Note that by the union bound, we have $\mathbb{P}[\widetilde{\mathcal{E}}] \geq 1 - 5\delta$. By Lemma I.2, we have that

$$\forall h \in [H], \quad \mathbb{P}^{\widehat{\pi}}\left[\sup_{f \in \widehat{\mathcal{V}}_h} \max_{a \in \mathcal{A}} \left|(\mathcal{P}_{h-1}[\widehat{V}_h] - \mathcal{P}_{h-1}[f_h])(\boldsymbol{x}_{h-1}, a)\right| > 3\varepsilon\right] \leq \frac{4\log(8M^6 N_{\mathsf{test}}^2 H^8/\delta)}{N_{\mathsf{test}}}, \tag{53}$$

where $M = \lceil 8\varepsilon^{-1} C_{\mathsf{push}} H \rceil$ and $N_{\mathsf{test}} = 2^8 M^2 H \varepsilon^{-1} \log(8M^6 H^8 \varepsilon^{-2}\delta^{-1})$. On the other hand, by Lemma I.4, we have

$$\forall h \in [H], \quad V_h^{\tilde{\pi}} \in \widehat{\mathcal{V}}_h.$$

Thus, substituting $V_h^{\tilde{\pi}}$ for $f_h$ in (53) we get that for all $h \in [H+1]$.

$$\mathbb{P}^{\widehat{\pi}}\left[\max_{a \in \mathcal{A}} \left|(\mathcal{P}_{h-1}[\widehat{V}_h] - \mathcal{P}_{h-1}[V_h^{\tilde{\pi}}])(\boldsymbol{x}_{h-1}, a)\right| > 3\varepsilon\right] \leq \frac{4\log(8M^6 N_{\mathsf{test}}^2 H^8/\delta)}{N_{\mathsf{test}}}.$$

This together with Lemma L.1, instantiated with $\mu[\cdot] = \mathbb{P}^{\widehat{\pi}}[\cdot]$; $\tau = h-1$; $V_{\tau+1} = V_h^{\tilde{\pi}}$; and $\delta = \delta'$ (with $\delta'$ as in Algorithm 5), translates to:

$$\forall h \in [H], \quad \mathbb{E}^{\widehat{\pi}}\left[D_{\mathsf{tv}}(\widehat{\pi}_h(\boldsymbol{x}_h), \tilde{\pi}_h(\boldsymbol{x}_h))\right] \leq \frac{4\log(8M^6 N_{\mathsf{test}}^2 H^8/\delta)}{N_{\mathsf{test}}} + \delta',$$

$$= \frac{4\log(8M^6 N_{\mathsf{test}}^2 H^8/\delta)}{N_{\mathsf{test}}} + \frac{\delta}{4M^7 N_{\mathsf{test}}^2 H^8 |\mathcal{V}|},$$

$$\leq \frac{\varepsilon}{4H^3 C_{\mathsf{push}}},$$

where the last step follows from the fact that $N_{\mathsf{test}} = 2^8 M^2 H \varepsilon^{-1} \log(8M^6 H^8 \varepsilon^{-2}\delta^{-1})$ (with $M$ as in Line 3).

**Bounding the sample complexity.** We now bound the number of episodes used by Algorithm 5 under the event $\widetilde{\mathcal{E}}$. First, we fix $h \in [H]$, and focus on the number of episodes used within a to call $\mathsf{RVFS}_h$, excluding any episodes used by any subsequent calls to $\mathsf{RVFS}_\tau$ for $\tau > h$. We start by counting the number of episodes used to test the fit of the estimated value functions $\widehat{V}_{h+1:H}$. Starting from Line 8, there are for-loops over $(x_{h-1}, a_{h-1}) \in \mathcal{C}_h$, $\ell = H, \ldots, h+1$, and $n \in [N_{\mathsf{test}}]$ to collect partial episodes using the learned policy $\widehat{\pi}$ in Algorithm 5, where $N_{\mathsf{test}} = 2^8 M^2 H \varepsilon^{-1} \log(8M^6 H^8 \varepsilon^{-2}\delta^{-1})$ and $M = \lceil 8\varepsilon^{-1} C_{\mathsf{push}} H \rceil$. Note that executing $\widehat{\pi}$ requires the local simulator and uses $N_{\mathsf{sim}} = 2 \log(4M^7 N_{\mathsf{test}}^2 H^2 |\mathcal{V}|/\delta)/\varepsilon^2$ local simulator queries to output an action at each layer (since Algorithm 5 calls Algorithm 7 with confidence level $\delta' = \delta/(8M^7 N_{\mathsf{test}}^2 H^8 |\mathcal{V}|)$). Also, note that whenever a test fails in Line 14 and the recursive RVFS calls return, the for-loop in Line 8 resumes. We also know (by Lemma I.1) that the number of times the test fails in Line 14 is at most $M$. Thus, the number of times the for-loop in Line 8 resumes is bounded by $HM$; here, $H$ accounts for possible test failures across all layers $\tau \in [h+1 \mathinner{.\,.} H]$. Thus, the total sample complexity required to generate episodes between lines Line 8 and Line 11 is bounded by

$$\text{\# episodes for roll-outs} \leq \underbrace{MH}_{\text{\# of times Line 8 resumes}} \cdot \underbrace{MH^2 N_{\mathsf{test}} N_{\mathsf{sim}}}_{\text{Sample complexity in case of no test failures}}. \tag{54}$$

Note that the test in Line 14 also uses episodes because it calls the operator $\widehat{\mathcal{P}}$ for every $a \in \mathcal{A}$. Thus, the number of episodes used for the test in Line 14 is bounded by

$$\text{\# episodes for the tests} \leq \underbrace{MH}_{\text{\# of times Line 8 resumes}} \cdot \underbrace{MHAN_{\mathsf{test}} N_{\mathsf{sim}}}_{\text{Sample complexity for Line 14}}. \tag{55}$$

We now count the number of episodes used to re-fit the value function in Line 16 and onwards. Note that starting from Line 16, there are for-loops over $(x_{h-1}, a_{h-1}) \in \mathcal{C}_h$ and $i \in [N_{\mathsf{reg}}]$ to generate $A \cdot N_{\mathsf{est}}(|\mathcal{C}_h|) \leq A \cdot N_{\mathsf{est}}(M)$ partial episodes using $\widehat{\pi}$, where $N_{\mathsf{est}}(k) = 2N_{\mathsf{reg}}^2 \log(8AN_{\mathsf{reg}} H k^3/\delta)$

is defined as in Algorithm 5. Since $\widehat{\pi}$ uses the local simulator and requires $N_{\mathsf{est}}$ samples (see Algorithm 7) to output an action at each layer, the number of episodes used to refit the value function is bounded by

$$\text{\# episodes for } V\text{-refitting} \le MN_{\mathsf{reg}}AN_{\mathsf{est}}(M)HN_{\mathsf{sim}}. \tag{56}$$

Therefore, by (54), (55), and (56), the number of episodes used within a single call to $\mathsf{RVFS}_h$ (not accounting for episodes used by recursive calls to $\mathsf{RVFS}_\tau$, for $\tau > h$) is bounded by

$$\text{\# episodes used locally within } \mathsf{RVFS}_h \le M^2H(H+A)N_{\mathsf{test}}N_{\mathsf{sim}} + MN_{\mathsf{reg}}AN_{\mathsf{est}}(M)HN_{\mathsf{sim}}. \tag{57}$$

Finally, by Lemma I.1, $\mathsf{RVFS}_h$ may be called at most $M$ times throughout the execution of $\mathsf{RVFS}_0$. Using this together with (57) and accounting for the number of episodes from all layers $h \in [H]$, we get that the total number of episodes is bounded by

$$M^3H^2(H+A)N_{\mathsf{test}}N_{\mathsf{sim}} + M^2H^2N_{\mathsf{reg}}AN_{\mathsf{est}}(M)N_{\mathsf{sim}}.$$

Substituting the expressions of $M$, $N_{\mathsf{test}}$, $N_{\mathsf{est}}$, $N_{\mathsf{sim}}$, and $N_{\mathsf{reg}}$ from Algorithm 5 and Algorithm 7, we obtain the desired number of episodes, which concludes the proof. $\qquad\square$

## I.7 Proof of Theorem I.2 (Guarantee of RVFS.bc)

**Proof of Theorem I.2.** Let $\widehat{V}_{1:H}$ be the value function estimates produced by $\mathsf{RVFS}_0$ within Algorithm 6, and let $\widehat{\boldsymbol{\pi}}_h^{\mathsf{RVFS}}(\cdot) \in \arg\max_{a \in \mathcal{A}} \widehat{\boldsymbol{\mathcal{P}}}_{h,\varepsilon_{\mathsf{RVFS}},\delta'}[\widehat{V}_{h+1}](\cdot, a)$, for all $h \in [H]$ with $\widehat{V}_{H+1} \equiv 0$ with $\varepsilon_{\mathsf{RVFS}}$ and $\delta'$ as in Algorithm 6. Further, let and let $\tilde{\pi}_{1:H} \in \Pi_{\mathsf{S}}$ be the stochastic policy defined recursively via

$$\forall x \in \mathcal{X}, \quad \tilde{\boldsymbol{\pi}}_\tau(x) \in \arg\max_{a \in \mathcal{A}} \begin{cases} \widehat{\boldsymbol{Q}}_\tau(x,a), & \text{if } \|\widehat{\boldsymbol{Q}}_\tau(x,\cdot) - \mathcal{P}_\tau[V_{\tau+1}^{\tilde{\pi}}](x,\cdot)\|_\infty \le 4\varepsilon_{\mathsf{RVFS}}, \\ \mathcal{P}_\tau[V_{\tau+1}^{\tilde{\pi}}](x,a), & \text{otherwise,} \end{cases} \tag{58}$$

for $\tau = H, \ldots, 1$, where $\widehat{\boldsymbol{Q}}_\tau(x,a) \coloneqq \widehat{\boldsymbol{\mathcal{P}}}_{\tau,\varepsilon_{\mathsf{RVFS}},\delta'}[\widehat{V}_{\tau+1}](x,a)$. By Theorem I.1, there is an event $\widetilde{\mathcal{E}}$ of probability at least $1 - \delta/2$ under which:

$$\tilde{\pi} \in \Pi_{4\varepsilon_{\mathsf{RVFS}}}, \tag{59}$$

and

$$\forall h \in [H], \quad \mathbb{E}^{\widehat{\pi}^{\mathsf{RVFS}}}\left[D_{\mathsf{tv}}(\widehat{\pi}_h^{\mathsf{RVFS}}(\boldsymbol{x}_h), \tilde{\pi}_h(\boldsymbol{x}_h))\right] \le \frac{\varepsilon_{\mathsf{RVFS}}}{4H^3C_{\mathsf{push}}} \le \frac{\varepsilon}{4H^2}, \tag{60}$$

where the last inequality follows by the choice of $\varepsilon_{\mathsf{RVFS}}$ in Algorithm 6.

For the rest of the proof, we condition on $\widetilde{\mathcal{E}}$. By (59), (63), and Proposition M.1 instantiated with $\varepsilon_{\mathsf{mis}} = 0$ (due to all $\pi$-realizability), we have that there is an event $\widetilde{\mathcal{E}'}$ of probability at least $1 - \delta/2$ under which the policy $\widehat{\pi}_{1:H}$ produced by BehaviorCloning ensures that

$$J(\widehat{\pi}_{1:H}^{\mathsf{RVFS}}) - J(\widehat{\pi}_{1:H}) \le \frac{\varepsilon}{2}. \tag{61}$$

Now, by Lemma C.6 (the performance difference lemma), we have for $\tilde{\pi}$ as in (58):

$$J(\tilde{\pi}) - J(\widehat{\pi}_{1:H}^{\mathsf{RVFS}}) = \sum_{h=1}^{H} \mathbb{E}^{\widehat{\pi}^{\mathsf{RVFS}}}\left[Q_h^{\tilde{\pi}}(\boldsymbol{x}_h, \tilde{\boldsymbol{\pi}}_h(\boldsymbol{x}_h)) - Q_h^{\tilde{\pi}}(\boldsymbol{x}_h, \widehat{\boldsymbol{\pi}}_h^{\mathsf{RVFS}}(\boldsymbol{x}_h))\right],$$

$$\le H \sum_{h=1}^{H} \mathbb{E}^{\widehat{\pi}^{\mathsf{RVFS}}}\left[D_{\mathsf{tv}}(\widehat{\pi}_h^{\mathsf{RVFS}}(\boldsymbol{x}_h), \tilde{\pi}_h(\boldsymbol{x}_h))\right],$$

and so by (63), we have

$$J(\tilde{\pi}) - J(\widehat{\pi}_{1:H}^{\mathsf{RVFS}}) \le \varepsilon/4. \tag{62}$$

Finally, since $\tilde{\pi} \in \Pi_{4\varepsilon_{\mathsf{RVFS}}}$ (see (59)), we have by Lemma H.1,

$$J(\pi^\star) - J(\tilde{\pi}) \le 12H\varepsilon_{\mathsf{RVFS}} \le \varepsilon/4,$$

where the last inequality follows by the choice $\varepsilon_{\mathsf{RVFS}}$ in Algorithm 6. Combining this with (61) and (62), we conclude that under $\widetilde{\mathcal{E}} \cap \widetilde{\mathcal{E}'}$:

$$J(\pi^\star) - J(\widehat{\pi}_{1:H}) \le \varepsilon.$$

By the union bound, we have $\mathbb{P}[\widetilde{\mathcal{E}} \cap \widetilde{\mathcal{E}'}] \ge 1 - \delta$, and so the desired suboptimality guarantee in (28) holds with probability at least $1 - \delta$.

**Bounding the sample complexity.** The sample complexity is dominated by the call to $\mathsf{RVFS}_0$ within $\mathsf{RVFS.bc}$ (Algorithm 6). Since $\mathsf{RVFS.bc}$ calls $\mathsf{RVFS}_0$ with $\varepsilon = \varepsilon_{\mathsf{RVFS}} = \varepsilon H^{-1}/48$, we conclude from Theorem I.1 that the total sample complexity is bounded by

$$\widetilde{O}\left(C_{\mathsf{push}}^8 H^{23} A \cdot \varepsilon^{-13}\right).$$

$\square$

## J   Guarantee under $V^\star$-Realizability (Proof of Theorem 4.1, Setup I)

In this section, we prove Theorem 4.1 under **Setup I** ($V^\star/\pi^\star$-realizability (Assumptions 4.2 and 4.3) and $\Delta$-gap (Assumption 4.4)). We prove this result as a consequence of the more general results (Theorem I.2) in Appendix I by appealing to the relaxed $V^\pi$-realizability condition in Assumption I.1.

### J.1   Analysis: Proof of Theorem 4.1 (Setup I)

We begin by showing that Assumption 4.2 and Assumption 4.4 together imply that Assumption I.1 holds for any $\varepsilon_{\mathsf{real}} \le \Delta/2$; we prove this by showing that the benchmark policy class $\Pi_{\varepsilon'}$ (Appendix H.2) reduces to $\{\pi^\star\}$ when $\varepsilon' \le \Delta/2$.

**Lemma J.1.** *Assume that $\mathcal{V}$ satisfies Assumption 4.2 ($V^\star$-realizability), and that Assumption 4.4 (gap) holds with $\Delta > 0$. Then, for all $\varepsilon' \le \Delta/2$, we have $\Pi_{\varepsilon'} = \{\pi^\star\}$ and $\mathcal{V}$ satisfies Assumption I.1 with $\varepsilon_{\mathsf{real}} = \varepsilon'$.*

Informally Lemma J.1, whose proof is in Appendix J.2, states that under Assumption 4.2, Assumption 4.4 with $\Delta > 0$, and Assumption 4.1 (pushforward coverability) with $C_{\mathsf{push}} > 0$, we are essentially in the setting of Theorem I.1 (guarantee of $\mathsf{RVFS}$ under relaxed $V^\pi$-realizability), as long as we choose $\varepsilon_{\mathsf{real}} \le \Delta/2$. With this, we now state and prove a central guarantee for $\mathsf{RVFS}$ under $V^\star$-realizability with a gap.

**Lemma J.2** (Intermediate guarantee for $\mathsf{RVFS}$ under **Setup I**). *Let $\delta \in (0, 1)$ be given, and suppose that:*

- *Assumption 4.1 (pushforward coverability) holds with parameter $C_{\mathsf{push}} > 0$;*

- *Assumption 4.4 (gap) holds with parameter $\Delta > 0$;*

- *The function class $\mathcal{V}$ satisfies Assumption 4.2 ($V^\star$-realizability).*

*Then, for any $f \in \mathcal{V}$ and $\varepsilon \in (0, \Delta/8)$, with probability at least $1 - \delta$, $\mathsf{RVFS}_0(f, \mathcal{V}, \varnothing, \varnothing; \mathcal{V}, \varepsilon, \delta)$ (Algorithm 5) terminates and returns value functions $\widehat{V}_{1:H}$ that satisfy*

$$\forall h \in [H], \quad \mathbb{P}^{\widehat{\pi}}\left[\widehat{\boldsymbol{\pi}}_h(\boldsymbol{x}_h) \ne \pi_h^\star(\boldsymbol{x}_h)\right] \le \frac{\varepsilon}{4C_{\mathsf{push}}H^3},$$

*where $\widehat{\boldsymbol{\pi}}_h(x) \in \arg\max_{a \in \mathcal{A}} \widehat{\boldsymbol{\mathcal{P}}}_{\tau, \varepsilon, \delta'}[\widehat{V}_{h+1}](x, a)$, for all $h \in [H]$, with $\delta'$ is defined as in Algorithm 5.*

**Proof of Lemma J.2.** From Lemma J.1, we have that $\Pi_{4\varepsilon} = \{\pi^\star\}$, and so Theorem I.1 implies that with probability at least $1 - \delta$,

$$\forall h \in [H], \quad \frac{\varepsilon}{4C_{\mathsf{push}}H^3} \ge \mathbb{E}^{\widehat{\pi}}\left[D_{\mathsf{tv}}\left(\widehat{\pi}_h(\boldsymbol{x}_h), \pi_h^\star(\boldsymbol{x}_h)\right)\right] = \mathbb{P}^{\widehat{\pi}}\left[\widehat{\boldsymbol{\pi}}_h(\boldsymbol{x}_h) \ne \pi_h^\star(\boldsymbol{x}_h)\right],$$

where the equality follows by the fact that $\pi^\star$ is deterministic. $\square$

From here, Theorem 4.1 follows swiftly as a consequence.

**Proof of Theorem 4.1** (Setup I). Let $\widehat{V}_{1:H}$ be the value function estimates produced by $\mathsf{RVFS}_0$ within Algorithm 6, and let $\widehat{\boldsymbol{\pi}}_h^{\mathsf{RVFS}}(\cdot) \in \arg\max_{a \in \mathcal{A}} \widehat{\mathcal{P}}_{h, \varepsilon_{\mathsf{RVFS}}, \delta'}[\widehat{V}_{h+1}](\cdot, a)$, for all $h \in [H]$ with $\widehat{V}_{H+1} \equiv 0$ with $\varepsilon_{\mathsf{RVFS}}$ and $\delta'$ as in Algorithm 6. By Lemma J.2, there is an event $\widetilde{\mathcal{E}}$ of probability at least $1 - \delta/2$ under which:

$$\mathbb{P}^{\widehat{\pi}}\left[\widehat{\boldsymbol{\pi}}_h(\boldsymbol{x}_h) \ne \pi_h^\star(\boldsymbol{x}_h)\right] \le \frac{\varepsilon_{\mathsf{RVFS}}}{4H^3 C_{\mathsf{push}}} \le \frac{\varepsilon}{4H^2}, \tag{63}$$

where the last inequality follows by the choice of $\varepsilon_{\mathsf{RVFS}}$ in Algorithm 6.

For the rest of the proof, we condition on $\widetilde{\mathcal{E}}$. By (63) and Assumption 4.3 ($\pi^\star$-realizability), the policy $\widehat{\pi}_{1:H}^{\mathsf{RVFS}}$ returned by $\mathsf{RVFS}_0$ satisfies the condition in Proposition M.1 with $\varepsilon_{\mathsf{mis}} = \varepsilon/(4 C_{\mathsf{push}} H^3)$. Thus, by Proposition M.1, there is an event $\widetilde{\mathcal{E}}'$ of probability at least $1 - \delta/2$ under which the policies $\widehat{\pi}_{1:H}$ produced by $\mathsf{RVFS.bc}$ satisfy

$$J(\widehat{\pi}_{1:H}^{\mathsf{RVFS}}) - J(\widehat{\pi}_{1:H}) \le \frac{\varepsilon}{H} + \frac{\varepsilon}{2} \le \frac{3\varepsilon}{2}.$$

We now condition on $\widetilde{\mathcal{E}} \cap \widetilde{\mathcal{E}}'$. By Lemma C.6 (performance difference lemma), we have

$$J(\pi^\star) - J(\widehat{\pi}_{1:H}^{\mathsf{RVFS}}) = \sum_{h=1}^H \mathbb{E}^{\widehat{\pi}^{\mathsf{RVFS}}}\big[Q_h^{\pi^\star}(\boldsymbol{x}_h, \pi_h^\star(\boldsymbol{x}_h)) - Q_h^{\pi^\star}(\boldsymbol{x}_h, \widehat{\boldsymbol{\pi}}_h^{\mathsf{RVFS}}(\boldsymbol{x}_h))\big],$$

$$\le H \sum_{h=1}^H \mathbb{P}^{\widehat{\pi}}\big[\widehat{\boldsymbol{\pi}}_h(\boldsymbol{x}_h) \ne \pi_h^\star(\boldsymbol{x}_h)\big],$$

$$\le \varepsilon/(4H),$$

where the last inequality follows by (63).

Finally, by the union bound, we have $\mathbb{P}[\widetilde{\mathcal{E}} \cap \widetilde{\mathcal{E}}'] \ge 1 - \delta$, and so the desired suboptimality guarantee in (28) holds with probability at least $1 - \delta$.

**Bounding the sample complexity.** The sample complexity is dominated by the call to $\mathsf{RVFS}_0$ within $\mathsf{RVFS.bc}$ (Algorithm 6). Since $\mathsf{RVFS.bc}$ calls $\mathsf{RVFS}_0$ with $\varepsilon = \varepsilon_{\mathsf{RVFS}} = \varepsilon H^{-1}/48$, we conclude from Theorem I.1 that the total number of episodes is bounded by

$$\widetilde{O}\big(C_{\mathsf{push}}^8 H^{23} A \cdot \varepsilon^{-13}\big).$$

**Bounding the number of oracle calls.** We now bound the number of calls to the oracle described in Remark 4.1 that RVFS makes. To do this, note that bounding the number of calls to the oracle is equivalent to bounding the number of executions of Line 14 in Algorithm 5, or, equivalently, the number of times the operator $\widehat{\mathcal{P}}$ in Line 14 is invoked during the entire execution of RVFS, including all subsequent recursive calls to RVFS. The proof of Lemma I.2 establishes a direct bound on the number of times the operator $\widehat{\mathcal{P}}$ is called, giving an upper limit of $O(C_{\mathsf{push}} N_{\mathsf{test}} H^4 \varepsilon^{-1})$, where $N_{\mathsf{test}}$ is a polynomial in the problem parameters (see Algorithm 5) and does *not* depend on $\mathcal{X}$. $\quad\square$

## J.2 Proof of Lemma J.1 (Relaxed $V^\pi$-Realizability under Gap)

**Proof of Lemma J.1.** Fix $\varepsilon' \in (0, 1)$ and $\pi \in \Pi_{\varepsilon'}$. Let $(\widetilde{\boldsymbol{Q}}_h(x, a))_{(h,x,a) \in [H] \times \mathcal{X} \times \mathcal{A}}$ be independent random variables such that

$$\forall h \in [H], \quad \boldsymbol{\pi}_h(\cdot) \in \arg\max_{a' \in \mathcal{A}} \widetilde{\boldsymbol{Q}}_h(\cdot, a') \quad \text{and} \quad \|\widetilde{\boldsymbol{Q}}_h - Q_h^\pi\|_\infty \le \varepsilon', \text{ almost surely.}$$

Such a collection of random state-action value functions $(\widetilde{\boldsymbol{Q}}_h(x, a))_{(h,x,a) \in [H] \times \mathcal{X} \times \mathcal{A}}$ is guaranteed to exist for $pi$ by the definition of $\Pi_{\varepsilon'}$. We will show via backward induction over $\ell = H + 1, \dots, 1$ that

$$\forall x \in \mathcal{X}_\ell, \quad \boldsymbol{\pi}_\ell(x) = \pi_\ell^\star(x) \tag{64}$$

almost surely, with the convention that $\pi_{H+1} \equiv \pi_{H+1}^\star \equiv \pi_{\mathsf{unif}}$. This convention makes the base case, $\ell = H + 1$, hold trivially.

Now, we consider the general case. Fix $h \in [H]$ and suppose that (64) holds for all $\ell \in [h+1 .. H+1]$. We will show that it holds for $\ell = h$.

Thanks to the induction hypothesis, we have for all $x \in \mathcal{X}_{h+1}$ and $a \in \mathcal{A}$:

$$V_{h+1}^\pi(x) = V_{h+1}^\star(x),$$

and so

$$Q_h^\pi \equiv \mathcal{P}_h[V_{h+1}^\pi] \equiv \mathcal{P}_h[V_{h+1}^\star] = V_h^\star. \tag{65}$$

Fix $x \in \mathcal{X}$. We will show that $\pi_h(x) = \pi_h^\star(x)$ almost surely. Note that thanks to (65), the fact that $\|\widetilde{\boldsymbol{Q}}_h - \mathcal{T}_h[Q_{h+1}^\pi]\|_\infty \le \varepsilon'$ almost surely, implies that

$$\|\widetilde{\boldsymbol{Q}}_h - Q_h^\star\|_\infty \le \varepsilon',$$

almost surely. Using this, we have, almost surely

$$
\begin{aligned}
Q_h^\star(x, \boldsymbol{\pi}_h(x)) &\ge \widetilde{Q}_h(x, \boldsymbol{\pi}_h(x)) - \varepsilon', \\
&\ge \widetilde{Q}_h(x, \pi_h^\star(x)) - \varepsilon', \\
&\ge Q_h^\star(x, \pi_h^\star(x)) - 2\varepsilon' = Q_h^\star(x, \pi_h^\star(x)) - \Delta.
\end{aligned}
\tag{66}
$$

On the other hand, if $\boldsymbol{\pi}_h(x) \ne \pi_h^\star(x)$, then

$$
Q_h^\star(x, \boldsymbol{\pi}_h(x)) < Q_h^\star(x, \pi_h^\star(x)) - \Delta,
$$

which would contradict (66). Thus, $\boldsymbol{\pi}_h(x) = \pi_h^\star(x)$, which concludes the induction and shows that $\pi \equiv \pi^\star$. We conclude that $\Pi_{\varepsilon'} = \{\pi^\star\}$. $\qquad\square$

# K    Guarantee for Weakly Correlated ExBMDPs (Proof of Theorem B.1)

In this section, we prove Theorem B.1, the main guarantee for RVFS$^{\mathsf{exo}}$. First, in Appendix K.1 we state a number of supporting technical lemmas and use them to prove Theorem B.1. The remainder of the section (Appendix K.2 through Appendix K.6) contains the proofs for the intermediate results.

## K.1    Analysis: Proof of Theorem B.1

Recall that the the $V^\pi$-realizability assumption required by RVFS for Theorem 4.1 is not satisfied in ExBMDPs, as the value functions for policies that act on the exogenous noise variables cannot be realized as a function of the true decoder $\phi^\star$. In RVFS$^{\mathsf{exo}}$, we address this issue by applying the randomized rounding technique in Line 11 to the learned value functions. The crux of the analysis will be to show that for an appropriate choice of the rounding parameters $\zeta_{1:H}$, the policies produced by RVFS$^{\mathsf{exo}}$ are endogenous in the sense that we can write $\pi(x) = \pi(\phi^\star(x))$ for all $x \in \mathcal{X}$. This will allow us to leverage the decoder realizability (Assumption 3.3), which implies that the function class $\mathcal{V} = \mathcal{V}_{1:H}$ given by

$$
\mathcal{V}_h \coloneqq \{x \mapsto f(\phi(x)) : f \in [0, H]^S, \phi \in \Phi\},
\tag{67}
$$

satisfies $V^\pi$-realizability for all endogenous policies $\pi$.

In what follows, we first motivate the randomized rounding approach in RVFS$^{\mathsf{exo}}$ in detail and prove that it succeeds, then use this result to proceed with an analysis similar to that of Theorem 4.1 (**Setup II**), re-using many of the technical tools developed for Theorem 4.1.

### K.1.1    Randomized Rounding for Endogeneity

Naively, to ensure that the policies we execute are endogenous, it would seem that we require knowledge of the true decoder $\phi^\star$. Alas, knowing $\phi^\star$ trivializes the ExBMDP problem by reducing it to the tabular setting. To avoid requiring knowledge of $\phi^\star$, we apply a *randomized rounding* to the policies learned by RVFS$^{\mathsf{exo}}$ to ensure their endogeneity.

Let $\varepsilon > 0$ be fixed going forward. Recall that compared to RVFS, RVFS$^{\mathsf{exo}}$ (Algorithm 8) takes an additional input $\zeta_{1:H} \subset (0, 1/2)$ and executes the following coarsened policies:

$$
\widehat{\boldsymbol{\pi}}_h(\cdot) \in \arg\max_{a \in \mathcal{A}} \lceil \widehat{\boldsymbol{\mathcal{P}}}_{h,\varepsilon,\delta}[\widehat{V}_{h+1}](\cdot, a)/\varepsilon + \zeta_h \rceil.
$$

The *rounding parameters* $\zeta_{1:H}$, which can be thought of as an offset, are chosen randomly; this will be elucidated in the sequel.

Following a similar analysis to Appendix I (**Setup II**), we can associate a near-optimal benchmark policy $\tilde{\pi} \in \Pi_{2\varepsilon}$ with $\widehat{\pi}$ in order to emulate certain properties of the $\Delta$-gap assumption. In particular, generalizing the construction in Eq. (27), we define a near-optimal benchmark policy $\tilde{\pi}$ recursively via:

$$
\forall x \in \mathcal{X}, \ \tilde{\boldsymbol{\pi}}_\tau(x; \zeta_{1:H}, \varepsilon, \delta) \in \arg\max_{a \in \mathcal{A}}
\begin{cases}
\lceil \widehat{Q}_\tau(x, a)/\varepsilon + \zeta_\tau \rceil, & \text{if } \|\widehat{Q}_\tau(x, \cdot) - \mathcal{P}_\tau[V_{\tau+1}^{\tilde{\pi}}](x, \cdot)\|_\infty \le 4\varepsilon^2, \\
\lceil \mathcal{P}_\tau[V_{\tau+1}^{\tilde{\pi}}](x, a)/\varepsilon + \zeta_\tau \rceil, & \text{otherwise,}
\end{cases}
\tag{68}
$$

for $\tau = H, \dots, 1$, where $\widehat{Q}_\tau(\cdot, a) \coloneqq \widehat{\boldsymbol{\mathcal{P}}}_{\tau,\varepsilon,\delta}[\widehat{V}_{\tau+1}](\cdot, a)$.

Naively, to use the benchmark policy $\tilde{\pi}$ within the analysis based on relaxed $V^\pi$-realizability (Assumption I.1) in Appendix I , we would require the function class $\mathcal{V}$ to realize $(V_h^{\tilde{\pi}})$. However, as argued earlier, this is not feasible unless $\tilde{\pi}$ is an endogenous policy. Fortunately, it turns out that if $\zeta_{1:H}$ (the additional input to RVFS$^{\text{exo}}$) are drawn randomly from uniform distribution over $[0, 1/2]$, then with constant probability, $\tilde{\pi}$ is indeed endogenous. What's more, under such an event, and for all possible choices of $(\widehat{V}_h)$ in (68) uniformly, $\tilde{\pi}$ "snaps" onto the stochastic endogenous policy $\bar{\pi}(\cdot; \zeta_{1:H}, \varepsilon)$ defined recursively as follows:

$$\bar{\pi}_h(\cdot; \zeta_{1:H}, \varepsilon) \in \arg\max_{a \in \mathcal{A}} \lceil \mathcal{P}_h[V_{h+1}^{\bar{\pi}}](x, a)/\varepsilon + \zeta_h \rceil, \tag{69}$$

for $h = H, \ldots, 1$. Informally, this happens because, as long as $\zeta_{1:H} \subset (0, 1/2)$ avoid certain pathological locations in $(0, 1/2)$, the coarsened state-action value functions $\varepsilon \cdot \lceil \mathcal{P}_\tau[V_{\tau+1}^{\bar{\pi}}](x, a)/\varepsilon + \zeta_h \rceil$ defining $\bar{\pi}$ exhibit a "gap" of order $\Theta(\varepsilon^2)$ separating optimal actions from the rest. This "snapping" behavior is analogous to what happens in **Setup I** with $V^\star$-realizability and $\Delta$-gap, where $\Pi_\varepsilon$ reduces to $\{\pi^\star\}$ for all $\varepsilon < \Delta/2$ (see Lemma J.1). We formalize these claims in the next two lemmas. We start by showing that $\bar{\pi}$ is endogenous and that $\bar{\pi} \in \Pi_{2\varepsilon}$. The proof is in Appendix K.2.

**Lemma K.1** (Endogenous Benchmark policies). *For any $\delta \in (0, 1)$, $\varepsilon \in (0, 1/2)$, and $\zeta_{1:H} \subset (0, 1/2)$, the stochastic policy $\bar{\pi}(\cdot; \zeta_{1:H}, \varepsilon)$ defined in Eq. (69) is endogenous, and we have $\bar{\pi}(\cdot; \zeta_{1:H}, \varepsilon) \in \Pi_{2\varepsilon}$.*

Next, we show that $\tilde{\pi}$ "snaps" onto $\bar{\pi}$ for the certain choices of $\zeta_{1:H}$. The proof is in Appendix K.3.

**Lemma K.2** (Snapping probability). *Let $\delta \in (0, 1)$, $\varepsilon \in (0, 1/2)$ be given, and $\mathbb{P}^\zeta$ denote the probability law of $\zeta_1, \ldots, \zeta_H \sim \text{unif}([0, 1/2])$. Then, there is an event $\mathcal{E}_{\text{rand}}$ of probability at least $1 - 24SAH\varepsilon$ under $\zeta_{1:H} \sim \mathbb{P}^\zeta$ such that for all $\widetilde{V} \in (\mathcal{X} \times [H] \to [0, H])$ simultaneously,*

$$\forall h \in [H], \quad \tilde{\pi}_h(\cdot; \widetilde{V}, \zeta_{1:H}, \varepsilon, \delta) = \bar{\pi}_h(\cdot; \zeta_{1:H}, \varepsilon),$$

*where $\tilde{\pi}_h(\cdot; \widetilde{V}, \zeta_{1:H}, \varepsilon, \delta)$ is defined as in (68) with $\widehat{V} = \widetilde{V}$, and $\bar{\pi}$ is defined as in (69).*

The lemma together, with Lemma K.1, implies that with constant probability under $\mathbb{P}^\zeta$, the benchmark policies $(\tilde{\pi}_h)$ used in the analysis of RVFS$^{\text{exo}}$ are endogenous and satisfy $\tilde{\pi} \in \Pi_{2\varepsilon}$.

### K.1.2 Pushforward Coverability

In order to proceed with the analysis strategy in Appendix I, we need to verify that pushforward coverability is satisfied for ExBMDPs under the weak correlation assumption. We do so in the next lemma; see Appendix K.4 for a proof.

**Lemma K.3** (Pushforward coverability). *A weakly correlated ExBMDP with constant $C_{\text{exo}}$ (see Assumption B.1) satisfies $C_{\text{push}}$-pushforward coverability (Assumption 4.1) with $C_{\text{push}} = C_{\text{exo}} \cdot SA$, where $S \in \mathbb{N}$ is the number of endogenous states.*

Equipped with the preceding lemmas, we proceed with an analysis similar to the approach for Theorem 4.1 (**Setup II**) in Appendix I. In what follows, we state a number of technical lemmas that apply the relevant results from Appendix I to the ExBMDP setting we consider here.

### K.1.3 Bounding the Number of Test Failures

Since the size of the core sets $\mathcal{C}_{1:H}$ in RVFS$^{\text{exo}}$ is directly proportional to the number of test failures, the next result, which bounds $|\mathcal{C}_h|$ for all $h \in [H]$, allows us to show that RVFS$^{\text{exo}}$ (Algorithm 8) terminates in a polynomial number of iterations.

**Lemma K.4** (Bounding the number of test failures). *Let $\delta, \varepsilon \in (0, 1)$ and $\zeta_{1:H} \in [0, 1/2]$ be given, and suppose that Assumption B.1 (weak correlation) holds with $C_{\text{exo}} > 0$. Let $f \in \mathcal{V}$, be given, where $\mathcal{V}$ is an arbitrary function class. Then, there is an event $\mathcal{E}$ of probability at least $1 - \delta$ under which the call to RVFS$_0^{\text{exo}}(f, \mathcal{V}^H, \varnothing, \varnothing, 0; \mathcal{V}, \varepsilon, \zeta_{1:H}, \delta)$ (Algorithm 8) terminates, and throughout the execution of RVFS$_0^{\text{exo}}$, we have*

$$|\mathcal{C}_h| \le \lceil 8\varepsilon^{-2} C_{\text{exo}} SAH \rceil.$$

**Proof of Lemma K.4.** The results follows from Lemma K.3 and Lemma I.1. $\qquad\square$

### K.1.4 Value Function Regression Guarantee

We next give a guarantee for the estimated value functions $\widehat{V}_{1:H}$ computed within $\mathsf{RVFS}^{\mathsf{exo}}$ in Line 26 of Algorithm 8.

**Lemma K.5** (Value function regression guarantee). *Let $h \in [0 .. H]$, $\delta, \varepsilon \in (0,1)$, and $\zeta_{1:H} \in [0, 1/2]$ be given, and consider a call to $\mathsf{RVFS}_0^{\mathsf{exo}}$ in the setting of Lemma K.4. Further, let $\mathcal{V}$ be defined as in Eq. (67), and assume that $\Phi$ satisfies Assumption 3.3. Then, for any endogenous policy $\pi$ in $\Pi_S$, there is an event $\mathcal{E}_h''$ of probability at least $1 - \delta/H$ under which for any $k \geq 1$, if*

- *$\mathsf{RVFS}_h^{\mathsf{exo}}$ gets called for the $k$th time during the execution of $\mathsf{RVFS}_0^{\mathsf{exo}}$; and*

- *this $k$th call terminates and returns $(\widehat{V}_{h:H}, \widehat{\mathcal{V}}_{h:H}, \mathcal{C}_{h:H}, \mathcal{B}_{h:H}, t_{h:H})$,*

*then if $(\widehat{\pi}_\tau)_{\tau \geq h}$ is the policy induced by $\widehat{V}_{h:H}$ and $N_{\mathsf{reg}}$ is set as in Algorithm 8, we have*

$$\sum_{(x_{h-1}, a_{h-1}) \in \mathcal{C}_h} \frac{1}{N_{\mathsf{reg}}} \sum_{(x_h, -) \in \mathcal{D}_h(x_{h-1}, a_{h-1})} \left(\widehat{V}_h(x_h) - V_h^\pi(x_h)\right)^2$$

$$\leq \frac{9kH^2 \log(8k^2 H |\mathcal{V}|/\delta)}{N_{\mathsf{reg}}} + 8H^2 \sum_{(x_{h-1}, a_{h-1}) \in \mathcal{C}_h} \sum_{\tau=h}^{H} \mathbb{E}^{\widehat{\pi}} \left[ D_{\mathsf{tv}}(\widehat{\pi}_\tau(\boldsymbol{x}_\tau), \pi_\tau(\boldsymbol{x}_\tau)) \mid \boldsymbol{x}_{h-1} = x_{h-1}, \boldsymbol{a}_{h-1} = a_{h-1} \right],$$

*where the datasets $\{\mathcal{D}_h(x, a) : (x, a) \in \mathcal{C}_h\}$ are as in the definition of $\widehat{\mathcal{V}}_h$ in (16).*

**Proof of Lemma K.5.** Since $\Phi$ satisfies Assumption 3.3, the function class $\mathcal{V} = \mathcal{V}_{1:H}$ satisfies $V^\pi$-realizability for all endogenous policies $\pi$ (see Lemma D.1). Thus, the proof of Lemma K.5 follows from that of Lemma I.3 (see Appendix I.4). $\square$

### K.1.5 Confidence Sets

We now state a version of the confidence set validity lemma (Lemma I.4) that supports the ExBMDP setting.

**Lemma K.6** (Confidence sets). *Let $\varepsilon \in (0, 1/2)$ and $\zeta_{1:H} \subset [0, 1/2]$ be given, and suppose that*

- *Assumption B.1 holds with $C_{\mathsf{exo}} > 0$;*

- *The decoder class $\Phi$ satisfies Assumption 3.3;*

- *$\zeta_{1:H} \in \mathcal{E}_{\mathsf{rand}}$, where $\mathcal{E}_{\mathsf{rand}}$ is the event in Lemma K.2.*

*Let $f \in \mathcal{V}$ be arbitrary. There is an event $\mathcal{E}'''$ of probability at least $1 - 3\delta$ under which a call to $\mathsf{RVFS}_0^{\mathsf{exo}}(f, \mathcal{V}, \varnothing, \varnothing, 0; \mathcal{V}, \varepsilon, \zeta_{1:H}, \delta)$ terminates and returns tuple $(\widehat{V}_{1:H}, \widehat{\mathcal{V}}_{1:H}, \mathcal{C}_{1:H}, \mathcal{B}_{1:H}, t_{1:H})$ such that*

$$\forall h \in [H], \quad V_h^{\bar{\pi}} \in \widehat{\mathcal{V}}_h,$$

*where $\bar{\pi}_{1:H}$ is the policy defined recursively via*

$$\bar{\pi}_\tau(x) \in \arg\max_{a \in \mathcal{A}} \lceil \mathcal{P}_\tau[V_{\tau+1}^{\bar{\pi}}](x, a)/\varepsilon + \zeta_h \rceil, \quad \text{for } \tau = H, \ldots, 1. \tag{70}$$

While the proof of this lemma is very similar to that of Lemma I.4, we need a dedicated treatment to handle the rounding in $\mathsf{RVFS}^{\mathsf{exo}}$. The fully proof of Lemma K.8 is in Appendix I.5.

### K.1.6 Main Guarantee for $\mathsf{RVFS}^{\mathsf{exo}}$

We now state the central technical guarantee for $\mathsf{RVFS}^{\mathsf{exo}}$, Lemma K.7, which shows that the base invocation of the algorithm returns a set of value functions $\widehat{V}_{1:H}$ that induce a near-optimal policy $\widehat{\pi}$, as long as the randomized rounding parameters $\zeta_{1:H}$ satisfy $\zeta_{1:H} \in \mathcal{E}_{\mathsf{rand}}$, where $\mathcal{E}_{\mathsf{rand}}$ is the success event in Lemma K.2. The proof of the theorem is in Appendix K.6.

**Lemma K.7** (Main guarantee for $\mathsf{RVFS}^{\mathsf{exo}}$). *Let $\delta, \varepsilon \in (0, 1)$ and $\zeta_{1:H} \subset [0, \frac{1}{2}]$ be given, and suppose that*

- *Assumption B.1 holds with $C_{\mathsf{exo}} > 0$;*

- *The decoder class $\Phi$ satisfies Assumption 3.3;*

- *$\zeta_{1:H} \in \mathcal{E}_{\mathsf{rand}}$, where $\mathcal{E}_{\mathsf{rand}}$ is the event in Lemma K.2.*

*Then, for any $f \in \mathcal{V}$, with probability at least $1 - 5\delta$, a call to $\mathsf{RVFS}_0^{\mathsf{exo}}(f, \mathcal{V}^H, \varnothing, \varnothing, 0; \mathcal{V}, \varepsilon, \zeta_{1:H}, \delta)$ (Algorithm 8) terminates and returns value functions $\widehat{V}_{1:H}$ such that*

$$\forall h \in [H], \quad \mathbb{P}^{\widehat{\pi}}[\widehat{\boldsymbol{\pi}}_h(\boldsymbol{x}_h) \neq \bar{\pi}_h(\boldsymbol{x}_h)] \leq \frac{\varepsilon^2}{4H^3 SAC_{\mathsf{exo}}},$$

*where $\widehat{\boldsymbol{\pi}}_h(x) \in \arg\max_{a \in \mathcal{A}}[\widehat{\boldsymbol{\mathcal{P}}}_{h,\varepsilon,\delta'}[\widehat{V}_{h+1}](x,a)/\varepsilon + \zeta_h]$, for all $h \in [H]$, $\bar{\pi}$ is defined as in Eq. (69), and $\delta'$ is defined as in Algorithm 8. Furthermore, the number of episodes used by $\mathsf{RVFS}_0^{\mathsf{exo}}$ is bounded by*

$$\widetilde{O}\left(C_{\mathsf{exo}}^8 S^8 H^{10} A^9 \cdot \varepsilon^{-26}\right).$$

### K.1.7 Concluding: Main Guarantee for $\mathsf{RVFS}^{\mathsf{exo}}.\mathsf{bc}$

To conclude, we prove Theorem B.1, which shows that $\mathsf{RVFS}^{\mathsf{exo}}.\mathsf{bc}$ succeeds with high probability. Recall that $\mathsf{RVFS}^{\mathsf{exo}}.\mathsf{bc}$ (i) invokes $\mathsf{RVFS}^{\mathsf{exo}}$ multiple times for random samples $\zeta_{1:H}$ to ensure that the success event for Lemma K.7 occurs for at least one invocation, and (ii) extracts an executable policy using behavior cloning. Regarding the former point, note that the probability of the success event of Lemma K.2 can be boosted by sampling i.i.d. $\boldsymbol{\zeta}_{1:H}^{(1)}, \ldots, \boldsymbol{\zeta}_{1:H}^{(n)} \sim \mathbb{P}^\zeta$ inputs to $\mathsf{RVFS}^{\mathsf{exo}}$ for $n \geq 1$; as long as $n$ is polynomially large, with high probability at least one of the inputs $\boldsymbol{\zeta}_{1:H}^{(1)}, \ldots, \boldsymbol{\zeta}_{1:H}^{(n)}$ will satisfy the conclusion of Lemma K.2. Thus, it suffices to pick the policy with the highest value function among the different calls to $\mathsf{RVFS}^{\mathsf{exo}}$. Using this, we prove Theorem B.1.

**Proof of Theorem B.1.** Recall that Algorithm 9 picks the final policy $\widehat{\pi}_{1:H}^{(i_{\mathsf{opt}})}$ based on empirical value function estimates. In particular, for every $i \in [N_{\mathsf{boost}}]$ (with $N_{\mathsf{boost}}$ as in Algorithm 9), the estimate $\widehat{J}(\widehat{\pi}_{1:H}^{(i)})$ for $J(\widehat{\pi}_{1:H}^{(i)})$ is computed using $N_{\mathsf{eval}}$ episodes. Thus, by Hoeffding's inequality and the union bound, we have that there is an event $\breve{\mathcal{E}}$ of probability at least $1 - \delta/4$ under which

$$\forall i \in [N_{\mathsf{boost}}], \quad |J(\widehat{\pi}_{1:H}^{(i)}) - \widehat{J}(\widehat{\pi}_{1:H}^{(i)})| \leq \sqrt{2\log(2N_{\mathsf{boost}}/\delta)/N_{\mathsf{eval}}}.$$

Therefore, by definition of $i_{\mathsf{opt}}$ in Algorithm 9, we have that under $\breve{\mathcal{E}}$:

$$\forall i \in [N_{\mathsf{boost}}], \quad J(\widehat{\pi}_{1:H}^{(i)}) \leq J(\widehat{\pi}_{1:H}^{(i_{\mathsf{opt}})}) + \sqrt{2\log(2N_{\mathsf{boost}}/\delta)/N_{\mathsf{eval}}},$$
$$\leq J(\widehat{\pi}_{1:H}^{(i_{\mathsf{opt}})}) + \varepsilon/8, \tag{71}$$

where the last inequality follows by the choice of $N_{\mathsf{eval}}$ in Algorithm 9. On the other hand, by Lemma K.2, there is an event $\mathcal{E}^{\mathsf{success}}$ of $\mathbb{P}^\zeta$-probability at least

$$1 - (24SAH\varepsilon)^{N_{\mathsf{boost}}} \geq 1 - \delta/4 \quad \text{(by the choice of } N_{\mathsf{boost}} \text{ in Algorithm 9)}$$

under which there exists $j \in [N_{\mathsf{boost}}]$ such that $\boldsymbol{\zeta}_{1:H}^{(j)} \in \mathcal{E}_{\mathsf{rand}}$, where $\mathcal{E}_{\mathsf{rand}}$ is defined as in Lemma K.2. In what follows, we condition on the event $\mathcal{E}^{\mathsf{success}}$ and let $j \in [N_{\mathsf{boost}}]$ be such that $\boldsymbol{\zeta}_{1:H}^{(j)} \in \mathcal{E}_{\mathsf{rand}}$. Further, we use $\widehat{\pi}_{1:H}^{\mathsf{RVFS}}$ to denote the policy returned by the instance of $\mathsf{RVFS}^{\mathsf{exo}}$ that is used by $\mathsf{RVFS}^{\mathsf{exo}}.\mathsf{bc}$ to learn $\widehat{\pi}_{1:H}^{(j)}$.

By Proposition M.1 (instantiated with $\varepsilon_{\mathsf{mis}} = 0$), there is an event $\widetilde{\mathcal{E}}'$ of probability at least $1 - \delta/4$ under which the policy $\widehat{\pi}^{(j)}$ produced by $\texttt{BehaviorCloning}$ satisfies

$$J(\widehat{\pi}_{1:H}^{\mathsf{RVFS}}) - J(\widehat{\pi}_{1:H}^{(j)}) \leq \frac{\varepsilon}{2}. \tag{72}$$

By Lemma K.7 and the fact that $\boldsymbol{\zeta}_{1:H}^{(j)} \in \mathcal{E}_{\mathsf{rand}}$, there is an event $\widetilde{\mathcal{E}}$ of probability at least $1 - \delta/2$ under which:

$$\forall h \in [H], \quad \forall h \in [H], \quad \mathbb{P}^{\widehat{\pi}}[\widehat{\boldsymbol{\pi}}_h(\boldsymbol{x}_h) \neq \bar{\pi}_h(\boldsymbol{x}_h)] \leq \frac{\varepsilon_{\mathsf{RVFS}}^2}{4H^3 SAC_{\mathsf{exo}}} \leq \frac{\varepsilon}{4H^2}, \tag{73}$$

where $\bar{\pi}(\cdot) \coloneqq \bar{\pi}(\cdot; \boldsymbol{\zeta}_{1:H}^{(j)}, \varepsilon_{\mathsf{RVFS}})$ which is defined in (69).

Moving forward, we condition on $\breve{\mathcal{E}} \cap \widetilde{\mathcal{E}} \cap \widetilde{\mathcal{E}}'$. By Lemma C.6 (the performance difference lemma), we have

$$J(\bar{\pi}) - J(\widehat{\pi}_{1:H}^{\mathsf{RVFS}}) = \sum_{h=1}^{H} \mathbb{E}^{\widehat{\pi}^{\mathsf{RVFS}}}[Q_h^{\bar{\pi}}(\boldsymbol{x}_h, \bar{\pi}_h(\boldsymbol{x}_h)) - Q_h^{\bar{\pi}}(\boldsymbol{x}_h, \widehat{\boldsymbol{\pi}}_h^{\mathsf{RVFS}}(\boldsymbol{x}_h))],$$

$$\leq H \sum_{h=1}^{H} \mathbb{P}^{\widehat{\pi}}[\widehat{\boldsymbol{\pi}}_h(\boldsymbol{x}_h) \neq \bar{\pi}_h(\boldsymbol{x}_h)],$$

$$\leq \varepsilon/4, \tag{74}$$

where the last inequality follows by (73). Now, by Lemma K.1, we have $\bar{\pi} \in \Pi_{2\varepsilon_{\mathsf{RVFS}}}$, and so by Lemma H.1,

$$J(\pi^\star) - J(\bar{\pi}) \le 6H\varepsilon_{\mathsf{RVFS}} \le \varepsilon/8, \tag{75}$$

where the last inequality follows by the choice of $\varepsilon_{\mathsf{RVFS}}$ in Algorithm 9. Combining (75) with (71), (72), and (74), we get that

$$J(\pi^\star) - J(\widehat{\pi}_{1:H}) = J(\pi^\star) - J(\widehat{\pi}_{1:H}^{(i_{\mathrm{opt}})}) \le \varepsilon.$$

Finally, by the union bound, we have $\mathbb{P}[\check{\mathcal{E}} \cap \widetilde{\mathcal{E}} \cap \widetilde{\mathcal{E}}'] \ge 1 - \delta$, and so the desired suboptimality guarantee holds with probability at least $1 - \delta$.

**Bounding the sample complexity.**  The sample complexity is dominated by the calls to $\mathsf{RVFS}_0^{\mathsf{exo}}$ within $\mathsf{RVFS}^{\mathsf{exo}}$.bc (Algorithm 9). Since $\mathsf{RVFS}^{\mathsf{exo}}$.bc calls $\mathsf{RVFS}_0^{\mathsf{exo}}$ with suboptimality parameter $\varepsilon_{\mathsf{RVFS}} = \varepsilon H^{-1}/48$, we get by Lemma K.7 that the total sample complexity is bounded by

$$\widetilde{O}\left(C_{\mathsf{exo}}^8 S^8 H^{36} A^9 \cdot \varepsilon^{-26}\right).$$

$\square$

## K.2    Proof of Lemma K.1 (Endogenous Benchmark Policies)

**Proof of Lemma K.1.**  Fix $\delta \in (0,1)$, $\varepsilon \in (0, 1/2)$, and $\zeta'_{1:H} \subset [0, 1/2]$. We show via backward induction over $\ell = H + 1, \ldots, 1$ that $\bar{\pi}_\tau(\cdot; \zeta'_{1:H}, \varepsilon)$ is endogenous for all $\tau \in [\ell .. H + 1]$, with the convention that $\bar{\pi}_{H+1} = \pi_{\mathsf{unif}}$. The base case holds trivially by convention.

Fix $h \in [H]$ and suppose that the induction hypothesis holds for all $\ell \in [h+1 .. H+1]$. We show that it holds for $\ell = h$. First, by the induction hypothesis, $\bar{\pi}_\ell(\cdot; \zeta'_{1:H}, \varepsilon)$ is endogenous for all $\ell \in [h+1 .. H]$. Thus, there exists a function $f_{h+1} : \mathcal{S} \to [0, H-h]$ such that

$$V_{h+1}^{\bar{\pi}}(x') = f_{h+1}(\phi^\star(x')), \quad \forall x' \in \mathcal{X}.$$

Therefore, we have for all $(x, a) \in \mathcal{X} \times \mathcal{A}$:

$$\begin{aligned}
\mathcal{P}_h[V_{h+1}^{\bar{\pi}}](x,a) &= r_h(x,a) + \mathbb{E}[f_{h+1}(\phi^\star(\boldsymbol{x}_{h+1})) \mid \boldsymbol{x}_h = x, \boldsymbol{a}_h = a], \\
&= r_h(x,a) + \mathbb{E}[f_{h+1}(\boldsymbol{s}_{h+1}) \mid \boldsymbol{x}_h = x, \boldsymbol{a}_h = a], \\
&= r_h(x,a) + \mathbb{E}[f_{h+1}(\boldsymbol{s}_{h+1}) \mid \boldsymbol{s}_h = \phi^\star(x), \boldsymbol{a}_h = a], \tag{76}
\end{aligned}$$

where the last equality follows by the ExBMDP transition structure. Eq. (76) together with the fact that the rewards are endogenous (by assumption) implies that there exists $g_h : \mathcal{S} \times \mathcal{A} \to [0, H-h+1]$ such that

$$\forall (x,a) \in \mathcal{X} \times \mathcal{A}, \quad \mathcal{P}_h[V_{h+1}^{\bar{\pi}}](x,a) = g_h(\phi^\star(x), a),$$

which in turn implies that $x \mapsto \lceil \mathcal{P}_h[V_{h+1}^{\bar{\pi}}](x,a)/\varepsilon + \zeta'_h \rceil$ is only a function of $x$ through $\phi^\star(x)$ for all $a \in \mathcal{A}$. Thus, $\bar{\pi}_h$ is an endogenous policy and the induction is completed.

For the second claim, observe that for the functions $\widetilde{Q}_1, \cdots, \widetilde{Q}_H \in [0, H]^{\mathcal{X} \times \mathcal{A}}$ defined as

$$\forall h \in [H], \forall (x,a) \in \mathcal{X} \times \mathcal{A}, \quad \widetilde{Q}_h(x,a) = \varepsilon \cdot \lceil \mathcal{P}_h[V_{h+1}^{\bar{\pi}}](x,a)/\varepsilon + \zeta'_h \rceil,$$

we have

$$\forall h \in [H], \quad \bar{\pi}_h(\cdot; \zeta'_{1:H}, \varepsilon) \in \arg\max_{a \in \mathcal{A}} \widetilde{Q}_h(\cdot, a) \quad \text{and} \quad \|\widetilde{Q}_h - Q_h^{\bar{\pi}}\|_\infty \le 2\varepsilon,$$

which implies that $\bar{\pi}(\cdot; \zeta'_{1:H}, \varepsilon) \in \Pi_{2\varepsilon}$.  $\square$

## K.3    Proof of Lemma K.2 (Snapping Probability)

**Proof of Lemma K.2.**  Fix $\varepsilon \in (0,1)$ and $\delta \in (0, 1/2)$. For $\tau \le \ell \in [H]$, let $\mathbb{P}_{\tau:\ell}^\zeta$ denote the probability law of $\boldsymbol{\zeta}_\tau, \ldots, \boldsymbol{\zeta}_\ell$. We also use the shorthand $\mathbb{P}_\tau^\zeta$ for $\mathbb{P}_{\tau:\tau}^\zeta$, for all $\tau \in [H]$. We show via backward induction over $\ell = H + 1, \ldots, 1$ that there exists an event $\mathcal{E}_\ell$ of $\mathbb{P}_{\ell:H}^\zeta$-probability at least $1 - 24SA(H - \ell + 1)\varepsilon$ under which for all $\widetilde{V} \in (\mathcal{X} \times [H] \to [0, H])$:

$$\forall \tau \in [\ell .. H], \quad \tilde{\boldsymbol{\pi}}_\tau(\cdot; \widetilde{V}, \boldsymbol{\zeta}_{1:H}, \varepsilon, \delta) = \bar{\pi}_\tau(\cdot; \boldsymbol{\zeta}_{1:H}, \varepsilon),$$

with the convention that $\bar{\pi}_{H+1} \equiv \tilde{\pi}_{H+1} \equiv \pi_{\mathsf{unif}}$. We then set $\mathcal{E}_{\mathsf{rand}} = \mathcal{E}_1$.

The base case follows trivially by convention.

We now proceed with the inductive step. Fix $h \in [H]$ and suppose that the induction hypothesis holds for all $\ell \in [h + 1 .. H]$. We show that it holds for $\ell = h$. Throughout, we condition on $\mathcal{E}_{h+1}$. By definition of $\mathcal{E}_{h+1}$, we have for all $\widetilde{V} \in (\mathcal{X} \times [H] \to [0, H])$:

$$\forall \ell \in [h + 1 .. H], \quad \bar{\pi}_\ell(\cdot; \boldsymbol{\zeta}_{1:H}, \varepsilon) = \tilde{\pi}_\ell(\cdot; \widetilde{V}, \boldsymbol{\zeta}_{1:H}, \varepsilon, \delta).$$

This implies that for all $\widetilde{V} \in (\mathcal{X} \times [H] \to [0, H])$:

$$\forall x \in \mathcal{X}, \quad \tilde{\pi}_h(x; \widetilde{V}, \boldsymbol{\zeta}_{1:H}, \varepsilon, \delta) \in \underset{a \in \mathcal{A}}{\arg\max} \begin{cases} \left[\widehat{\boldsymbol{Q}}_h(x, a)/\varepsilon + \boldsymbol{\zeta}_h\right], & \text{if } \|\widehat{\boldsymbol{Q}}_h(x, \cdot) - \mathcal{P}_h[V_{h+1}^{\bar{\pi}}](x, \cdot)\|_\infty \le 4\varepsilon^2, \\ \left[\mathcal{P}_h[V_{h+1}^{\bar{\pi}}](x, a)/\varepsilon + \boldsymbol{\zeta}_h\right], & \text{otherwise,} \end{cases}$$

by the definition of $\tilde{\pi}_h$ in (68), where $\widehat{\boldsymbol{Q}}_h(\cdot, a) \coloneqq \widehat{\boldsymbol{\mathcal{P}}}_{h,\varepsilon,\delta}[\widetilde{V}_{h+1}](\cdot, a)$. From this, we see that to prove $\tilde{\pi}_h(\cdot; \widetilde{V}, \boldsymbol{\zeta}_{1:H}, \varepsilon, \delta) = \bar{\pi}_h(\cdot; \boldsymbol{\zeta}_{1:H}, \varepsilon)$ for all $\widetilde{V}$, it suffices to show that for all $x \in \mathcal{X}$ and $\widetilde{V}$,

$$\underset{a \in \mathcal{A}}{\arg\max}\left[\widehat{\boldsymbol{Q}}_h(x, a)/\varepsilon + \boldsymbol{\zeta}_h\right] = \underset{a \in \mathcal{A}}{\arg\max}\left[\mathcal{P}_h[V_{h+1}^{\bar{\pi}}](x, a)/\varepsilon + \boldsymbol{\zeta}_h\right], \quad \text{whenever } |\widehat{\boldsymbol{Q}}_h(x, a) - \mathcal{P}_h[V_{h+1}^{\bar{\pi}}](x, a)| \le 4\varepsilon^2.$$

Observe that a sufficient condition for this to hold is that

$$\forall x \in \mathcal{X}, \forall a \in \mathcal{A}, \forall \delta \in [-4\varepsilon^2, 4\varepsilon^2], \quad \left[(\mathcal{P}_h[V_{h+1}^{\bar{\pi}}](x, a) + \delta) \cdot \varepsilon^{-1} + \boldsymbol{\zeta}_h\right] = \left[\mathcal{P}_h[V_{h+1}^{\bar{\pi}}](x, a) \cdot \varepsilon^{-1} + \boldsymbol{\zeta}_h\right], \tag{77}$$

where $\delta$ represents all the possible values that the difference $\widehat{\boldsymbol{Q}}_h(x, a) - \mathcal{P}_h[V_{h+1}^{\bar{\pi}}](x, a)$ is allowed to take. By Lemma K.1, we know that $\bar{\pi}$ is endogenous, and so there exists a function $g_h : \mathcal{S} \times \mathcal{A} \to [0, H - h + 1]$ such that

$$\forall x \in \mathcal{X}, a \in \mathcal{A}, \quad \mathcal{P}_h[V_{h+1}^{\bar{\pi}}](x, a) = g_h(\phi^\star(x), a).$$

Toward proving Eq. (77), observe that for any $(s, a) \in \mathcal{S} \in \mathcal{A}$, if $\boldsymbol{\zeta}_h$ is such that

$$\begin{aligned} &g_h(s, a)/\varepsilon + \boldsymbol{\zeta}_h + 4\varepsilon \le \left[g_h(s, a)/\varepsilon + \boldsymbol{\zeta}_h\right], \\ \text{and} \quad &g_h(s, a)/\varepsilon + \boldsymbol{\zeta}_h - 4\varepsilon > \left[g_h(s, a)/\varepsilon + \boldsymbol{\zeta}_h\right] - 1, \end{aligned} \tag{78}$$

then, for all $\delta \in [-4\varepsilon^2, 4\varepsilon^2]$ and all $x \in \mathcal{X}$ such that $\phi^\star(x) = s$, we have

$$\left[(\mathcal{P}_h[V_{h+1}^{\bar{\pi}}](x, a) + \delta)/\varepsilon + \boldsymbol{\zeta}_h\right] = \left[(g_h(s, a) + \delta)/\varepsilon + \boldsymbol{\zeta}_h\right] = \left[g_h(s, a)/\varepsilon + \boldsymbol{\zeta}_h\right] = \left[\mathcal{P}_h[V_{h+1}^{\bar{\pi}}](x, a)/\varepsilon + \boldsymbol{\zeta}_h\right].$$

Therefore, if we let $\mathcal{E}_h(s, a)$ denote the event in (78), then under $\bigcap_{(s,a) \in \mathcal{S} \times \mathcal{A}} \mathcal{E}_h(s, a)$, the desired condition in (77) holds. At this point, setting $\mathcal{E}_h = (\bigcap_{(s,a) \in \mathcal{S} \times \mathcal{A}} \mathcal{E}_h(s, a)) \cap \mathcal{E}_{h+1}$ would complete the induction as long as $\mathbb{P}_{h:H}^\zeta[\mathcal{E}_h] \ge 1 - 24SA(H - h + 1)\varepsilon$. We now show that this is indeed the case by bounding the probability of the event $\bigcap_{(s,a) \in \mathcal{S} \times \mathcal{A}} \mathcal{E}_h(s, a)$. By the union bound, we have

$$\mathbb{P}_{h:H}^\zeta\left[\bigcap_{(s,a) \in \mathcal{S} \times \mathcal{A}} \mathcal{E}_h(s, a) \mid \mathcal{E}_{h+1}\right] \ge 1 - \sum_{(s,a) \in \mathcal{S} \times \mathcal{A}} \mathbb{P}_{h:H}^\zeta\left[\mathcal{E}_h(s, a)^c \mid \mathcal{E}_{h+1}\right], \tag{79}$$

where $\mathcal{E}_h(s, a)^c$ denotes the complement of $\mathcal{E}_h(s, a)$. We now bound the probability

$$\mathbb{P}_{h:H}^\zeta\left[\mathcal{E}_h(s, a)^c \mid \mathcal{E}_{h+1}\right].$$

Fix $(s, a) \in \mathcal{S} \times \mathcal{A}$. We have that $\boldsymbol{\zeta}_h \in \mathcal{E}(s, a)^c$ if and only if

$$\begin{aligned} &g_h(s, a)/\varepsilon + \boldsymbol{\zeta}_h + 4\varepsilon > \left[g_h(s, a)/\varepsilon + \boldsymbol{\zeta}_h\right], \\ \text{or} \quad &g_h(s, a)/\varepsilon + \boldsymbol{\zeta}_h - 4\varepsilon \le \left[g_h(s, a)/\varepsilon + \boldsymbol{\zeta}_h\right] - 1. \end{aligned} \tag{80}$$

Now, since $\boldsymbol{\zeta}_h \in [0, 1/2]$, Lemma L.3 (instantiated with $(x, \zeta, \nu) = (g_h(s, a)/\varepsilon, \boldsymbol{\zeta}_h, 4\varepsilon)$) implies that (80) holds only if

$$\left[g_h(s, a)/\varepsilon\right] - 4\varepsilon \le g_h(s, a)/\varepsilon + \boldsymbol{\zeta}_h \le \left[g_h(s, a)/\varepsilon\right] + 4\varepsilon \quad \text{or} \quad 0 \le \boldsymbol{\zeta}_h \le 4\varepsilon. \tag{81}$$

Further, note that since $\boldsymbol{\zeta}_h$ is uniformly distributed over $[0, 1/2]$, the $\mathbb{P}_h^\zeta$-probability of the event in (81) is at most the sum of the lengths of the intervals

$$\left[\left[g_h(s, a)/\varepsilon\right] - g_h(s, a)/\varepsilon - 4\varepsilon, \left[g_h(s, a)/\varepsilon\right] - g_h(s, a)/\varepsilon + 4\varepsilon\right] \quad \text{and} \quad [0, 4\varepsilon],$$

multiplied by 2, which is equal to $24\varepsilon$. Therefore, we have

$$\mathbb{P}_{h:H}^{\zeta}\left[\mathcal{E}_h(s,a)^c \mid \mathcal{E}_{h+1}\right]$$
$$\leq \mathbb{P}_h^{\zeta}\left[\left\lceil g_h(s,a)/\varepsilon\right\rceil - 4\varepsilon \leq g_h(s,a)/\varepsilon + \boldsymbol{\zeta}_h \leq \left\lceil g_h(s,a)/\varepsilon\right\rceil + 4\varepsilon \ \text{ or } \ 0 \leq \boldsymbol{\zeta}_h \leq 4\varepsilon\right] \leq 24\varepsilon.$$

Combining this with (79), we obtain

$$\mathbb{P}_{h:H}^{\zeta}\left[\bigcap_{(s,a)\in\mathcal{S}\times\mathcal{A}} \mathcal{E}_h(s,a) \mid \mathcal{E}_{h+1}\right] \geq 1 - \sum_{(s,a)\in\mathcal{S}\times\mathcal{A}} \mathbb{P}_{h:H}^{\zeta}\left[\mathcal{E}_h(s,a)^c \mid \mathcal{E}_{h+1}\right] \geq 1 - 24SA\varepsilon.$$

Thus, by setting $\mathcal{E}_h = \left(\bigcap_{(s,a)\in\mathcal{S}\times\mathcal{A}} \mathcal{E}_h(s,a)\right) \cap \mathcal{E}_{h+1}$, we get that

$$\mathbb{P}_{h:H}^{\zeta}[\mathcal{E}_h] \geq \mathbb{P}_{h+1:H}^{\zeta}[\mathcal{E}_{h+1}] \cdot \mathbb{P}_{h:H}^{\zeta}[\mathcal{E}_h \mid \mathcal{E}_{h+1}] \geq (1 - 24SA(H-h)\varepsilon)(1 - 24SA\varepsilon),$$
$$\geq 1 - 24SA(H-h+1)\varepsilon,$$

which completes the induction. $\qquad\square$

## K.4 Proof of Lemma K.3 (Coverability in Weakly Correlated ExBMDP)

**Proof of Lemma K.3.** Fix $h \in [2..H]$ and define the measure $\mu$ as

$$\mu(x) \coloneqq \sum_{\xi'\in\Xi} q(x' \mid (\phi_h^\star(x'),\xi')) \cdot \mathbb{P}[\boldsymbol{\xi}_h = \xi'] \cdot \mathbb{P}[\boldsymbol{s}_h = \phi_h^\star(x') \mid \boldsymbol{s}_{h-1} = \phi_{h-1}^\star(x), \boldsymbol{a}_{h-1} = a],$$

for all $h \in [H]$ and $x \in \mathcal{X}$. We show that $\mu$ satisfies Assumption 4.1 with $C_{\mathsf{push}} = C_{\mathsf{exo}} \cdot SA$. First, note that $\mu$ is indeed a probability measure over $\mathcal{X}$. Fix $(x,a,x') \in \mathcal{X} \times \mathcal{A} \times \mathcal{X}$. We have

$$\mathbb{P}[\boldsymbol{x}_h = x' \mid \boldsymbol{x}_{h-1} = x, \boldsymbol{a}_{h-1} = a]$$
$$= \frac{\mathbb{P}[\boldsymbol{x}_h = x', \boldsymbol{x}_{h-1} = x \mid \boldsymbol{a}_{h-1} = a]}{\mathbb{P}[\boldsymbol{x}_{h-1} = x \mid \boldsymbol{a}_{h-1} = a]},$$
$$= \frac{\mathbb{P}[\boldsymbol{x}_h = x', \boldsymbol{x}_{h-1} = x \mid \boldsymbol{a}_{h-1} = a]}{\mathbb{P}[\boldsymbol{x}_{h-1} = x]},$$
$$= \frac{\sum_{\xi,\xi'\in\Xi} q(x' \mid (\phi_h^\star(x'),\xi')) \cdot q(x \mid (\phi_{h-1}^\star(x),\xi)) \cdot \mathbb{P}[\boldsymbol{s}_h = \phi_h^\star(x'), \boldsymbol{\xi}_h = \xi', \boldsymbol{s}_{h-1} = \phi_{h-1}^\star(x), \boldsymbol{\xi}_{h-1} = \xi \mid \boldsymbol{a}_{h-1} = a]}{\sum_{\xi\in\Xi} q(x \mid (\phi_{h-1}^\star(x),\xi)) \cdot \mathbb{P}[\boldsymbol{s}_{h-1} = \phi_{h-1}^\star(x), \boldsymbol{\xi}_{h-1} = \xi]},$$

and so by the ExBMDP structure:

$$= \frac{\sum_{\xi,\xi'\in\Xi} q(x' \mid (\phi_h^\star(x'),\xi')) \cdot q(x \mid (\phi_{h-1}^\star(x),\xi)) \cdot \mathbb{P}[\boldsymbol{\xi}_h = \xi', \boldsymbol{\xi}_{h-1} = \xi] \cdot \mathbb{P}[\boldsymbol{s}_h = \phi_h^\star(x') \mid \boldsymbol{s}_{h-1} = \phi_{h-1}^\star(x), \boldsymbol{a}_{h-1} = a]}{\sum_{\xi\in\Xi} q(x \mid (\phi_h^\star(x),\xi)) \cdot \mathbb{P}[\boldsymbol{\xi}_{h-1} = \xi]},$$

and by Assumption B.1

$$\leq C_{\mathsf{exo}} \frac{\sum_{\xi,\xi'\in\Xi} q(x' \mid (\phi_h^\star(x'),\xi')) \cdot q(x \mid (\phi_{h-1}^\star(x),\xi)) \cdot \mathbb{P}[\boldsymbol{\xi}_h = \xi'] \cdot \mathbb{P}[\boldsymbol{\xi}_{h-1} = \xi] \cdot \mathbb{P}[\boldsymbol{s}_h = \phi_h^\star(x') \mid \boldsymbol{s}_{h-1} = \phi_{h-1}^\star(x), \boldsymbol{a}_{h-1} = a]}{\sum_{\xi\in\Xi} q(x \mid (\phi_h^\star(x),\xi)) \cdot \mathbb{P}[\boldsymbol{\xi}_{h-1} = \xi]}$$
$$= C_{\mathsf{exo}} \sum_{\xi'\in\Xi} q(x' \mid (\phi_h^\star(x'),\xi')) \cdot \mathbb{P}[\boldsymbol{\xi}_h = \xi'] \cdot \mathbb{P}[\boldsymbol{s}_h = \phi_h^\star(x') \mid \boldsymbol{s}_{h-1} = \phi_{h-1}^\star(x), \boldsymbol{a}_{h-1} = a],$$
$$= C_{\mathsf{exo}}SA \cdot \mu(x'),$$

This completes the proof. $\qquad\square$

## K.5 Proof of Lemma K.6 (Confidence Sets)

To prove Lemma K.6, we need the following consequence of tests in Line 14 passing for all $\ell \in [h+1..H]$.

**Lemma K.8** (Consequence of passed tests). *Let $h \in [0..H]$, $\varepsilon > 0$, and $\zeta_{1:H} \in [0,1/2]$ be given and consider a call to $\mathsf{RVFS}_0^{\mathsf{exo}}$ (Algorithm 8) in the setting of Lemma K.4. Further, let $\mathcal{E}$ be the event of Lemma K.4. There exists an event $\mathcal{E}_h'$ of probability at least $1 - \delta/H$ such that under $\mathcal{E} \cap \mathcal{E}_h'$, if a call to $\mathsf{RVFS}_h^{\mathsf{exo}}$ during the execution of $\mathsf{RVFS}_0^{\mathsf{exo}}$ terminates and returns $(\widehat{V}_{h:H}, \widehat{\mathcal{V}}_{h:H}, \mathcal{C}_{h:H}, \mathcal{B}_{h:H}, t_{h:H})$, then for any $(x_{h-1}, a_{h-1}) \in \mathcal{C}_h$ and $\ell \in [h+1..H+1]$:*

$$\mathbb{P}^{\widehat{\pi}}\left[\sup_{f\in\widehat{\mathcal{V}}_\ell} \max_{a\in\mathcal{A}} \left|(\mathcal{P}_{\ell-1}[\widehat{V}_\ell] - \mathcal{P}_{\ell-1}[f_\ell])(\boldsymbol{x}_{\ell-1}, a)\right| > 3\varepsilon^2 \mid \boldsymbol{x}_{h-1} = x_{h-1}, \boldsymbol{a}_{h-1} = a_{h-1}\right] \leq \frac{4\log(8M^6 N_{\mathsf{test}}^2 H^8/\delta)}{N_{\mathsf{test}}},$$

*where $(\widehat{\pi}_\tau)_{\tau\geq h} \subset \Pi_{\mathsf{S}}$, $M$, and $N_{\mathsf{test}}$ are as in $\mathsf{RVFS}_h^{\mathsf{exo}}$ (Algorithm 8).*

**Proof of Lemma K.8.** This is just a restatement of Lemma I.2, and the proof is exactly the same as the latter. □

We will also use Lemma I.5; even though this result is stated in section for the $V^\pi$-realizable setting, it is also applicable to the ExBMDP variant of RVFS as it merely says something about the order in which the $(\mathsf{RVFS}_h^{\mathsf{exo}})$ instances are called. With this, we now prove Lemma K.6.

**Proof of Lemma K.6.** The proof is very similar to that of Lemma I.2, with differences to account for the "coarsening" of the learned and benchmark policies.

We prove the desired result for $\mathcal{E}''' \coloneqq \mathcal{E} \cap \mathcal{E}_1' \cap \mathcal{E}_1'' \cap \cdots \cap \mathcal{E}_H' \cap \mathcal{E}_H''$, where $\mathcal{E}$, $(\mathcal{E}_h')$, and $(\mathcal{E}_h'')$ are the events in Lemma K.4, Lemma K.8, and Lemma K.5, respectively. Throughout, we condition on $\mathcal{E}'''$. First, note that by Lemma K.4, $\mathsf{RVFS}_0^{\mathsf{exo}}$ terminates. Let $(\widehat{V}_{1:H}, \widehat{\mathcal{V}}_{1:H}, \mathcal{C}_{1:H}, \mathcal{B}_{1:H}, t_{1:H})$ be its returned tuple.

We show via backward induction over $\ell = H + 1, \ldots, 1$, that

$$V_\ell^{\tilde\pi} \in \widehat{\mathcal{V}}_\ell, \tag{82}$$

where $\tilde\pi_{1:H}$ is the stochastic policy defined recursively via

$$\forall x \in \mathcal{X}, \ \boldsymbol{\tilde\pi}_\tau(x; \zeta_{1:H}, \varepsilon, \delta) \in \arg\max_{a \in \mathcal{A}} \begin{cases} \lceil \widehat{\boldsymbol{Q}}_\tau(x,a)/\varepsilon + \zeta_\tau \rceil, & \text{if } \|\widehat{\boldsymbol{Q}}_\tau(x,\cdot) - \mathcal{P}_\tau[V_{\tau+1}^{\tilde\pi}](x,\cdot)\|_\infty \le 4\varepsilon^2, \\ \lceil \mathcal{P}_\tau[V_{\tau+1}^{\tilde\pi}](x,a)/\varepsilon + \zeta_\tau \rceil, & \text{otherwise,} \end{cases}$$

for $\tau = H, \ldots, 1$, where $\widehat{\boldsymbol{Q}}_\tau(\cdot, a) \coloneqq \widehat{\boldsymbol{\mathcal{P}}}_{\tau,\varepsilon,\delta}[\widehat{V}_{\tau+1}](\cdot, a)$. Note that since $\zeta_{1:H} \in \mathcal{E}_{\mathsf{rand}}$ (for $\mathcal{E}_{\mathsf{rand}}$ is defined in Lemma K.2), we have $\tilde\pi \equiv \bar\pi$, where $\bar\pi$ is as in (70). Thus, instantiating the induction hypothesis with $\ell = h$ and using the definition of the confidence sets $(\widehat{\mathcal{V}}_\ell)$ in (16) together with $V_h^{\tilde\pi} \equiv V_h^{\bar\pi}$ (since $\tilde\pi \equiv \bar\pi$) implies the desired result.

**Base case [$\ell = H + 1$].** Holds trivially since $V_{H+1}^\pi \equiv 0$ for any $\pi \in \Pi_{\mathsf{S}}$ by convention.

**General case [$\ell \le H$].** Fix $h \in [H]$ and suppose that (82) holds for all $\ell \in [h+1 .. H+1]$. We show that this remains true for $\ell = h$. First, note that if $\mathsf{RVFS}_h^{\mathsf{exo}}$ is never called during the execution of $\mathsf{RVFS}_0^{\mathsf{exo}}$, then $\widehat{\mathcal{V}}_h = \mathcal{V}_h$, and so (82) holds for $\ell = h$, since $\tilde\pi = \bar\pi$ is endogenous under $\zeta_{1:H} \in \mathcal{E}_{\mathsf{rand}}$, where $\mathcal{E}_{\mathsf{rand}}$ is the event in Lemma K.2.

Now, suppose that $\mathsf{RVFS}_h^{\mathsf{exo}}$ is called at least once, and let $(\widehat{V}_{h:H}^+, \widehat{\mathcal{V}}_{h:H}^+, \mathcal{C}_{h:H}^+, \mathcal{B}_{h:H}^+, t_{h:H}^+)$ be the output of the last call to $\mathsf{RVFS}_h^{\mathsf{exo}}$ throughout the execution of $\mathsf{RVFS}_0^{\mathsf{exo}}$. Next, we show that

$$(\widehat{V}_{h:H}^+, \widehat{\mathcal{V}}_{h:H}^+, \mathcal{C}_{h:H}^+) = (\widehat{V}_{h:H}, \widehat{\mathcal{V}}_{h:H}, \mathcal{C}_{h:H}). \tag{83}$$

The for-loop in Line 16 ensures that no instance of $(\mathsf{RVFS}_\tau^{\mathsf{exo}})_{\tau > h}$ can be called after the last call to $\mathsf{RVFS}_h^{\mathsf{exo}}$ (see Lemma I.5). Thus, the estimated value functions, confidence sets, and core sets for layers $h+1, \ldots, H$ remain unchanged after the last call to $\mathsf{RVFS}_h^{\mathsf{exo}}$; that is, (83) holds. Thus, by Lemma K.8, and since we are conditioning on $\mathcal{E}_{h+1:H}'$, we have that for all $(x_{h-1}, a_{h-1}) \in \mathcal{C}_h$ and $\ell \in [h+1 .. H+1]$:

$$\mathbb{P}^{\widehat\pi}\left[\sup_{f \in \widehat{\mathcal{V}}_\ell} \max_{a \in \mathcal{A}} \left|(\mathcal{P}_{\ell-1}[\widehat{V}_\ell] - \mathcal{P}_{\ell-1}[f_\ell])(\boldsymbol{x}_{\ell-1}, a)\right| > 3\varepsilon^2 \mid \boldsymbol{x}_{h-1} = x_{h-1}, \boldsymbol{a}_{h-1} = a_{h-1}\right] \le \frac{4\log(8M^6 N_{\mathsf{test}}^2 H^8/\delta)}{N_{\mathsf{test}}}. \tag{84}$$

Now, by the induction hypothesis, we have $V_\ell^{\tilde\pi} \in \widehat{\mathcal{V}}_\ell$, and so substituting $V_\ell^{\tilde\pi}$ for $f_\ell$ in (84), we get that for all $(x_{h-1}, a_{h-1}) \in \mathcal{C}_h$ and $\ell \in [h+1 .. H+1]$:

$$\mathbb{P}^{\widehat\pi}\left[\max_{a \in \mathcal{A}} \left|(\mathcal{P}_{\ell-1}[\widehat{V}_\ell] - \mathcal{P}_{\ell-1}[V_\ell^{\tilde\pi}])(\boldsymbol{x}_{\ell-1}, a)\right| > 3\varepsilon^2 \mid \boldsymbol{x}_{h-1} = x_{h-1}, \boldsymbol{a}_{h-1} = a_{h-1}\right] \le \frac{4\log(8M^6 N_{\mathsf{test}}^2 H^8/\delta)}{N_{\mathsf{test}}}.$$

Therefore, by Lemma L.2 (instantiated with $\mu[\cdot] = \mathbb{P}^{\widehat\pi}[\cdot \mid \boldsymbol{x}_{h-1} = x_{h-1}, \boldsymbol{a}_{h-1} = a_{h-1}]$, $\tau = \ell - 1$, $\varepsilon' = \varepsilon^2$, and $V_{\tau+1} = V_\ell^{\tilde\pi}$), we have that for all $(x_{h-1}, a_{h-1}) \in \mathcal{C}_h$ and $\ell \in [h+1 .. H+1]$:

$$\mathbb{E}^{\widehat\pi}\left[D_{\mathsf{tv}}(\widehat\pi_{\ell-1}(\boldsymbol{x}_{\ell-1}), \tilde\pi_{\ell-1}(\boldsymbol{x}_{\ell-1})) \mid \boldsymbol{x}_{h-1} = x_{h-1}, \boldsymbol{a}_{h-1} = a_{h-1}\right] \le \frac{4\log(8M^6 N_{\mathsf{test}}^2 H^8/\delta)}{N_{\mathsf{test}}} + \delta'. \tag{85}$$

Now, since $\bar{\pi}(\cdot;\zeta_{1:H},\varepsilon)$ is endogenous and $\tilde{\pi} \equiv \bar{\pi}$ (thanks to $\zeta_{1:H} \in \mathcal{E}_{\mathsf{rand}}$), Lemma K.5 (applied with $\pi = \bar{\pi}$) and the conditioning on $\bar{\mathcal{E}}''_{h+1:H}$ imply that:

$$\sum_{(x_{h-1},a_{h-1}) \in \mathcal{C}_h} \frac{1}{N_{\mathsf{reg}}} \sum_{(x_h,-) \in \mathcal{D}_h(x_{h-1},a_{h-1})} \left(\widehat{V}_h(x_h) - V_h^{\tilde{\pi}}(x_h)\right)^2$$

$$\leq \frac{9kH^2 \log(8k^2 H|\mathcal{V}|/\delta)}{N_{\mathsf{reg}}} + 8H^2 \sum_{(x_{h-1},a_{h-1}) \in \mathcal{C}_h} \sum_{\tau=h}^{H} \mathbb{E}^{\widehat{\pi}}\left[D_{\mathsf{tv}}(\widehat{\pi}_\tau(\boldsymbol{x}_\tau), \tilde{\pi}_\tau(\boldsymbol{x}_\tau)) \mid \boldsymbol{x}_{h-1} = x_{h-1}, \boldsymbol{a}_{h-1} = a_{h-1}\right],$$
(86)

where the datasets $\{\mathcal{D}_h(x,a) : (x,a) \in \mathcal{C}_h\}$ are as in the definition of $\widehat{\mathcal{V}}_h$ in (16). Combining (86) with (85), we conclude that

$$\sum_{(x_{h-1},a_{h-1}) \in \mathcal{C}_h} \frac{1}{N_{\mathsf{reg}}} \sum_{(x_h,-) \in \mathcal{D}_h(x_{h-1},a_{h-1})} \left(\widehat{V}_h(x_h) - V_h^{\tilde{\pi}}(x_h)\right)^2$$

$$\leq \frac{9MH^2 \log(8M^2 H|\mathcal{V}|/\delta)}{N_{\mathsf{reg}}} + 8MH^3 \cdot \frac{4\log(8M^6 N_{\mathsf{test}}^2 H^8/\delta)}{N_{\mathsf{test}}} + 8MH^3 \delta',$$

$$= \frac{9MH^2 \log(8M^2 H|\mathcal{V}|/\delta)}{N_{\mathsf{reg}}} + 8MH^3 \cdot \frac{4\log(8M^6 N_{\mathsf{test}}^2 H^8/\delta)}{N_{\mathsf{test}}} + 8MH^3 \frac{\delta}{4M^7 N_{\mathsf{test}}^2 H^8 |\mathcal{V}|},$$

$$\leq \varepsilon_{\mathsf{reg}}^2,$$
(87)

where we have used that $|\mathcal{C}_h| \leq M$. By the definition of $\widehat{\mathcal{V}}_h$ in (16), (87) implies that $V_h^{\tilde{\pi}} \in \widehat{\mathcal{V}}_h$, which completes the induction. $\qquad\square$

### K.6 Proof of Lemma K.7 (Main Guarantee of RVFS^exo)

**Proof of Lemma K.7.** We condition on the event $\widetilde{\mathcal{E}} := \mathcal{E} \cap \mathcal{E}''' \cap \mathcal{E}'_1 \cap \cdots \cap \mathcal{E}'_H$, where $\mathcal{E}$, $\mathcal{E}'''$, and $(\mathcal{E}'_h)$ are the events in Lemma K.4, Lemma K.6, and Lemma K.8, respectively. Note that by the union bound, we have $\mathbb{P}[\widetilde{\mathcal{E}}] \geq 1 - 5\delta$. By Lemma K.6, we have that

$$\forall h \in [H], \quad \mathbb{P}^{\widehat{\pi}}\left[\sup_{f \in \widehat{\mathcal{V}}_h} \max_{a \in \mathcal{A}} \left|(\mathcal{P}_{h-1}[\widehat{V}_h] - \mathcal{P}_{h-1}[f_h])(\boldsymbol{x}_{h-1}, a)\right| > 3\varepsilon^2\right] \leq \frac{4\log(8M^6 N_{\mathsf{test}}^2 H^8/\delta)}{N_{\mathsf{test}}},$$
(88)

where $M = \lceil 8\varepsilon^{-2} C_{\mathsf{exo}} SAH \rceil$ and $N_{\mathsf{test}} = 2^8 M^2 H \varepsilon^{-2} \log(8M^6 H^8 \varepsilon^{-2} \delta^{-1})$. On the other hand, by Lemma K.6, we have

$$\forall h \in [H], \quad V_h^{\tilde{\pi}} \in \widehat{\mathcal{V}}_h.$$

Thus, substituting $V_h^{\tilde{\pi}}$ for $f_h$ in (88) we get that for all $h \in [H+1]$.

$$\mathbb{P}^{\widehat{\pi}}\left[\max_{a \in \mathcal{A}} \left|(\mathcal{P}_{h-1}[\widehat{V}_h] - \mathcal{P}_{h-1}[V_h^{\tilde{\pi}}])(\boldsymbol{x}_{h-1}, a)\right| > 3\varepsilon^2\right] \leq \frac{4\log(8M^6 N_{\mathsf{test}}^2 H^8/\delta)}{N_{\mathsf{test}}}.$$

This together with Lemma L.2, instantiated with $\mu[\cdot] = \mathbb{P}^{\widehat{\pi}}[\cdot]$; $\tau = h-1$; $V_{\tau+1} = V_h^{\tilde{\pi}}$; and $\delta = \delta'$ (with $\delta'$ as in Algorithm 5), translates to:

$$\forall h \in [H], \quad \mathbb{E}^{\widehat{\pi}}[D_{\mathsf{tv}}(\tilde{\pi}_h(\boldsymbol{x}_h), \widehat{\pi}_h(\boldsymbol{x}_h))] \leq \frac{4\log(8M^6 N_{\mathsf{test}}^2 H^8/\delta)}{N_{\mathsf{test}}} + \delta',$$

$$= \frac{4\log(8M^6 N_{\mathsf{test}}^2 H^8/\delta)}{N_{\mathsf{test}}} + \frac{\delta}{4M^7 N_{\mathsf{test}}^2 H^8 |\mathcal{V}|},$$

$$\leq \frac{\varepsilon^2}{4H^3 SAC_{\mathsf{exo}}},$$
(89)

where the last step follows from the fact that $N_{\mathsf{test}} = 2^8 M^2 H \varepsilon^{-2} \log(8M^6 H^8 \varepsilon^{-2} \delta^{-1})$ (with $M$ as in Line 3). Now, since $\zeta_{1:H} \in \mathcal{E}_{\mathsf{rand}}$ (by assumption), Lemma K.2 implies that $\tilde{\pi} \equiv \bar{\pi}$, where the latter is the deterministic policy defined in (69). Thus, by (89), we have

$$\forall h \in [H], \quad \mathbb{P}^{\widehat{\pi}}[\bar{\pi}_h(\boldsymbol{x}_h) \neq \widehat{\boldsymbol{\pi}}_h(\boldsymbol{x}_h)] = \mathbb{E}^{\widehat{\pi}}[D_{\mathsf{tv}}(\tilde{\pi}_h(\boldsymbol{x}_h), \widehat{\pi}_h(\boldsymbol{x}_h))] \leq \frac{\varepsilon^2}{4H^3 SAC_{\mathsf{exo}}},$$

where the first equality follows by the fact that $\mathbb{P}[\bar{\pi}_h(x) \neq \widehat{\boldsymbol{\pi}}_h(x)] = D_{\mathsf{tv}}(\tilde{\pi}_h(x), \widehat{\pi}_h(x))$, for all $x \in \mathcal{X}$, since $\bar{\pi}_h$ is deterministic.

**Bounding the sample complexity.** We now bound the number of episodes used by Algorithm 8 under $\widetilde{\mathcal{E}}$. First, we fix $h \in [H]$, and focus on the number of episodes used within a to call $\mathsf{RVFS}_h^{\mathsf{exo}}$; excluding any episodes used by any recursive calls to $\mathsf{RVFS}_\tau^{\mathsf{exo}}$ for $\tau > h$. We start by counting the number of episodes used to test the fit of the estimated value functions $\widehat{V}_{h+1:H}$. Starting from Line 8, there are for-loops over $(x_{h-1}, a_{h-1}) \in \mathcal{C}_h$, $\ell = H, \ldots, h+1$, and $n \in [N_{\mathsf{test}}]$ to collected partial episodes using the learned policy $\widehat{\pi}$ in Algorithm 8, where $N_{\mathsf{test}} = 2^8 M^2 H \varepsilon^{-2} \log(8 M^6 H^8 \varepsilon^{-2} \delta^{-1})$ and $M = \lceil 8\varepsilon^{-2} C_{\mathsf{exo}} S A H \rceil$. Note that $\widehat{\pi}$ uses the local simulator and requires $N_{\mathsf{sim}} = 2\log(4 M^7 N_{\mathsf{test}}^2 H^2 |\mathcal{V}|/\delta)/\varepsilon^2$ samples to output an action at each layer (since Algorithm 8 calls Algorithm 7 with confidence level $\delta' = \delta/(8 M^7 N_{\mathsf{test}}^2 H^8 |\mathcal{V}|)$). Also, note that whenever a test fails in Line 14, the for-loop in Line 8 resumes. We also know (by Lemma K.4) that the number of times the test fails in Line 14 is at most $M$. Thus, the number of times the for-loop in Line 8 resumes is bounded by $HM$; here, $H$ accounts for test failures for all layers $\tau \in [h+1 .. H]$. Thus, the number of episodes required to between lines Line 8 and Line 11 is bounded by

$$\text{\# episodes for roll-outs} \leq \underbrace{MH}_{\text{\# of times Line 8 resumes}} \cdot \underbrace{MH^2 N_{\mathsf{test}} N_{\mathsf{sim}}}_{\text{Number of episodes in case of no test failures}} . \tag{90}$$

Note that the test in Line 14 also uses local simulator access because it calls the operator $\widehat{\mathcal{P}}$ for every $a \in \mathcal{A}$. Thus, the number of episodes used for the test in Line 14 is bounded by

$$\text{\# episodes for the tests} \leq \underbrace{MH}_{\text{\# of times Line 8 resumes}} \cdot \underbrace{MHAN_{\mathsf{test}} N_{\mathsf{sim}}}_{\text{Number of episodes used in Line 14}} . \tag{91}$$

We now count the number of episodes used to re-fit the value function; Line 16 onwards. Note that starting from Line 16, there are for-loops over $(x_{h-1}, a_{h-1}) \in \mathcal{C}_h$ and $i \in [N_{\mathsf{reg}}]$ to generate $A \cdot N_{\mathsf{est}}(|\mathcal{C}_h|) \leq A \cdot N_{\mathsf{est}}(M)$ partial episodes using $\widehat{\pi}$, where $N_{\mathsf{est}}(k) = 2 N_{\mathsf{reg}}^2 \log(8 A N_{\mathsf{reg}} H k^3/\delta)$ is as in Algorithm 8. And, since $\widehat{\pi}$ uses local simulator access and requires $N_{\mathsf{est}}$ samples (see Algorithm 7) to output an action at each layer, the number of episodes used to refit the value function is bounded by

$$\text{\# episodes for } V\text{-refitting} \leq M N_{\mathsf{reg}} A N_{\mathsf{est}}(M) H N_{\mathsf{sim}}. \tag{92}$$

Therefore, by (90), (91), and (92), the number of episodes used within a single call to $\mathsf{RVFS}_h^{\mathsf{exo}}$ (not accounting for episodes used by recursive calls to $\mathsf{RVFS}_\tau^{\mathsf{exo}}$, for $\tau > h$) is bounded by

$$\text{\# episodes used locally within } \mathsf{RVFS}_h^{\mathsf{exo}} \leq M^2 H(H+A) N_{\mathsf{test}} N_{\mathsf{sim}} + M N_{\mathsf{reg}} A N_{\mathsf{est}}(M) H N_{\mathsf{sim}}. \tag{93}$$

Finally, by Lemma K.4, $\mathsf{RVFS}_h^{\mathsf{exo}}$ may be called at most $M$ times throughout the execution of $\mathsf{RVFS}_0^{\mathsf{exo}}$. Using this together with (93) and accounting for the number of episodes from all layers $h \in [H]$, we get that the total number of episodes is bounded by

$$M^3 H^2(H+A) N_{\mathsf{test}} N_{\mathsf{sim}} + M^2 H^2 N_{\mathsf{reg}} A N_{\mathsf{est}}(M) N_{\mathsf{sim}}.$$

Substituting the expressions of $M$, $N_{\mathsf{test}}$, $N_{\mathsf{est}}$, $N_{\mathsf{sim}}$, and $N_{\mathsf{reg}}$ from Algorithm 8 and Algorithm 7, we obtain the desired number of episodes, which concludes the proof. $\qquad\square$

# L   Additional Technical Lemmas

**Lemma L.1.** *Let $\tau \in [H]$ and $\varepsilon, \delta, \nu \in (0,1)$ be given. Consider two value functions $V_{\tau+1}, \widehat{V}_{\tau+1} \in [0, H]$ and a measure $\mu \in \Delta(\mathcal{X})$ such that*

$$\mathbb{P}_{\boldsymbol{x}_\tau \sim \mu}\left[ \mathbb{I}\left\{ \max_{a \in \mathcal{A}} \left| (\mathcal{P}_\tau[\widehat{V}_{\tau+1}] - \mathcal{P}_\tau[V_{\tau+1}])(\boldsymbol{x}_\tau, a) \right| > 3\varepsilon \right\} \right] \leq \nu. \tag{94}$$

*Further, for $x \in \mathcal{X}$, let $\widehat{\boldsymbol{\pi}}_\tau(x) \in \arg\max_{a \in \mathcal{A}} \widehat{\boldsymbol{Q}}_\tau(x, a) \coloneqq \widehat{\mathcal{P}}_{\tau, \varepsilon, \delta}[\widehat{V}_{\tau+1}](x, a)$ and inductively define a randomized policy $\tilde{\pi}$ via*

$$\tilde{\boldsymbol{\pi}}_\tau(x) \in \arg\max_{a \in \mathcal{A}} \begin{cases} \widehat{\boldsymbol{Q}}_\tau(x, a), & \text{if } \|\widehat{\boldsymbol{Q}}_\tau(x, \cdot) - \mathcal{P}_\tau[V_{\tau+1}](x, \cdot)\|_\infty \leq 4\varepsilon, \\ \mathcal{P}_\tau[V_{\tau+1}](x, a), & \text{otherwise.} \end{cases}$$

*Then, we have*

$$\mathbb{E}_{\boldsymbol{x}_\tau \sim \mu}[D_{\mathsf{tv}}(\widehat{\pi}_\tau(\boldsymbol{x}_\tau), \tilde{\pi}_\tau(\boldsymbol{x}_\tau))] \leq \nu + \delta.$$

**Proof of Lemma L.1.** In this proof, we let $\mathbb{P}_\mu$ denote the probability law of $\boldsymbol{x}_\tau$ and $\mathbb{P}_{\mathcal{P}}$ denote the probability law of $\widehat{\boldsymbol{\mathcal{P}}}_{\tau,\varepsilon,\delta}$. Denote by $\mathcal{E}$ the $\mathbb{P}_\mu$-measurable event that $\max_{a\in\mathcal{A}}\left|(\mathcal{P}_\tau[\widehat{V}_{\tau+1}] - \mathcal{P}_\tau[V_{\tau+1}])(\boldsymbol{x}_\tau, a)\right| \le 3\varepsilon$. Fix $x \in \mathcal{E}$, and let $\mathcal{E}_x$ be the $\mathbb{P}_{\mathcal{P}}$-measurable event that $\max_{a\in\mathcal{A}}|\widehat{\boldsymbol{\mathcal{P}}}_{\tau,\varepsilon,\delta}[\widehat{V}_{\tau+1}](x,a) - \mathcal{P}_\tau[V_{\tau+1}](x,a)| \le 4\varepsilon$. From the definition of $\tilde{\pi}_\tau$, we have that

$$D_{\mathrm{tv}}(\widehat{\pi}_\tau(x), \tilde{\pi}_\tau(x))$$

$$= \frac{1}{2}\sum_{a\in\mathcal{A}}\left|\mathbb{P}_{\mathcal{P}}\left[\widehat{\boldsymbol{\pi}}_\tau(x) = a\right] - \mathbb{P}_{\mathcal{P}}\left[\tilde{\boldsymbol{\pi}}_\tau(x) = a\right]\right|,$$

$$= \frac{1}{2}\sum_{a\in\mathcal{A}}\left|\mathbb{P}_{\mathcal{P}}[\mathcal{E}_x]\mathbb{P}_{\mathcal{P}}\left[\widehat{\boldsymbol{\pi}}_\tau(x) = a \mid \mathcal{E}_x\right] + \mathbb{P}_{\mathcal{P}}[\mathcal{E}_x^c]\mathbb{P}_{\mathcal{P}}\left[\widehat{\boldsymbol{\pi}}_\tau(x) = a \mid \mathcal{E}_x^c\right] - \mathbb{P}_{\mathcal{P}}[\mathcal{E}_x]\mathbb{P}_{\mathcal{P}}\left[\tilde{\boldsymbol{\pi}}_\tau(x) = a \mid \mathcal{E}_x\right] - \mathbb{P}_{\mathcal{P}}[\mathcal{E}_x^c]\mathbb{P}_{\mathcal{P}}\left[\tilde{\boldsymbol{\pi}}_\tau(x) = a \mid \mathcal{E}\right.$$

$$\le \frac{1}{2}\sum_{a\in\mathcal{A}}\mathbb{P}_{\mathcal{P}}[\mathcal{E}_x]\cdot\left|\mathbb{P}_{\mathcal{P}}\left[\widehat{\boldsymbol{\pi}}_\tau(x) = a \mid \mathcal{E}_x\right] - \mathbb{P}_{\mathcal{P}}\left[\tilde{\boldsymbol{\pi}}_\tau(x) = a \mid \mathcal{E}_x\right]\right|$$

$$+ \sum_{a\in\mathcal{A}}\mathbb{P}_{\mathcal{P}}[\mathcal{E}_x^c]\cdot\left|\mathbb{P}_{\mathcal{P}}\left[\widehat{\boldsymbol{\pi}}_\tau(x) = a \mid \mathcal{E}_x^c\right] - \mathbb{P}_{\mathcal{P}}\left[\tilde{\boldsymbol{\pi}}_\tau(x) = a \mid \mathcal{E}_x^c\right]\right|, \quad \text{(Jensen's inequality)}$$

and since $\mathbb{P}_{\mathcal{P}}\left[\widehat{\boldsymbol{\pi}}_\tau(x) = a \mid \mathcal{E}_x\right] = \mathbb{P}_{\mathcal{P}}\left[\tilde{\boldsymbol{\pi}}_\tau(x) = a \mid \mathcal{E}_x\right] \forall a \in \mathcal{A}$, we have that

$$= \frac{1}{2}\sum_{a\in\mathcal{A}}\mathbb{P}_{\mathcal{P}}[\mathcal{E}_x^c]\cdot\left|\mathbb{P}_{\mathcal{P}}\left[\widehat{\boldsymbol{\pi}}_\tau(x) = a \mid \mathcal{E}_x^c\right] - \mathbb{P}_{\mathcal{P}}\left[\tilde{\boldsymbol{\pi}}_\tau(x) = a \mid \mathcal{E}_x^c\right]\right|,,$$

$$\le \mathbb{P}_{\mathcal{P}}\left[\mathcal{E}_x^c\right],$$

$$= \mathbb{P}_{\mathcal{P}}\left[\max_{a\in\mathcal{A}}|\widehat{\boldsymbol{\mathcal{P}}}_{\tau,\varepsilon,\delta}[\widehat{V}_{\tau+1}](x,a) - \mathcal{P}_\tau[V_{\tau+1}](x,a)| > 4\varepsilon\right],$$

$$\le \mathbb{P}_{\mathcal{P}}\left[\max_{a\in\mathcal{A}}|\widehat{\boldsymbol{\mathcal{P}}}_{\tau,\varepsilon,\delta}[\widehat{V}_{\tau+1}](x,a) - \mathcal{P}_\tau[\widehat{V}_{\tau+1}](x,a)| > \varepsilon\right], \quad \text{(see below)} \tag{95}$$

$$\le \delta, \tag{96}$$

where (95) follows from $x \in \mathcal{E}$ and the last inequality follows from Lemma H.2. Therefore, we have

$$\mathbb{E}_\mu[D_{\mathrm{tv}}(\widehat{\pi}_\tau(\boldsymbol{x}_\tau), \tilde{\pi}_\tau(\boldsymbol{x}_\tau))] \le \mathbb{P}_\mu[\mathcal{E}]\cdot\mathbb{E}_\mu[D_{\mathrm{tv}}(\widehat{\pi}_\tau(\boldsymbol{x}_\tau), \tilde{\pi}_\tau(\boldsymbol{x}_\tau)) \mid \mathcal{E}] + \mathbb{P}_\mu[\mathcal{E}^c],$$
$$\le \delta + \nu,$$

where the first inequality follows by the fact that the total variation distance is bounded by 1, and the last inequality follows by (94) and (96). $\qquad\square$

**Lemma L.2.** *Let $\tau \in [H]$ and $\varepsilon', \delta, \nu \in (0,1)$, and $\zeta_{1:H} \in [0, 1/2]$ be given. Further, consider two value functions $V_{\tau+1}, \widehat{V}_{\tau+1} \in [0, H]$ and measure $\mu \in \Delta(\mathcal{X})$ such that*

$$\mathbb{P}_{\boldsymbol{x}_\tau\sim\mu}\left[\mathbb{I}\left\{\max_{a\in\mathcal{A}}\left|(\mathcal{P}_\tau[\widehat{V}_{\tau+1}] - \mathcal{P}_\tau[V_{\tau+1}])(\boldsymbol{x}_\tau, a)\right| > 3\varepsilon'\right\}\right] \le \nu. \tag{97}$$

*Further, for $x \in \mathcal{X}_\tau$, let $\widehat{\boldsymbol{\pi}}_\tau(x) \in \arg\max_{a\in\mathcal{A}}\lceil\widehat{\boldsymbol{Q}}_\tau(x,a)/\varepsilon' + \zeta_\tau\rceil$, where $\widehat{\boldsymbol{Q}}_\tau(x,a) := \widehat{\boldsymbol{\mathcal{P}}}_{\tau,\varepsilon',\delta}[\widehat{V}_{\tau+1}](x,a)$, and inductively define*

$$\tilde{\boldsymbol{\pi}}_\tau(x) \in \arg\max_{a\in\mathcal{A}}\begin{cases} \lceil\widehat{\boldsymbol{Q}}_\tau(x,a)/\varepsilon' + \zeta_\tau\rceil, & \text{if } \|\widehat{\boldsymbol{Q}}_\tau(x,\cdot) - \mathcal{P}_\tau[V_{\tau+1}](x,\cdot)\|_\infty \le 4\varepsilon', \\ \lceil\mathcal{P}_\tau[V_{\tau+1}](x,a)/\varepsilon' + \zeta_\tau\rceil, & \text{otherwise.} \end{cases}$$

*Then, we have*

$$\mathbb{E}_{\boldsymbol{x}_\tau\sim\mu}[D_{\mathrm{tv}}(\widehat{\pi}_\tau(\boldsymbol{x}_\tau), \tilde{\pi}_\tau(\boldsymbol{x}_\tau))] \le \nu + \delta.$$

**Proof of Lemma L.2.** In this proof, we let $\mathbb{P}_\mu$ denote the probability law of $\boldsymbol{x}_\tau$ and $\mathbb{P}_{\mathcal{P}}$ denote the probability law of $\widehat{\boldsymbol{\mathcal{P}}}_{\tau,\varepsilon,\delta}$. Denote by $\mathcal{E}$ be the $\mathbb{P}_\mu$-measurable event that $\max_{a\in\mathcal{A}}\left|(\mathcal{P}_\tau[\widehat{V}_{\tau+1}] - \mathcal{P}_\tau[V_{\tau+1}])(\boldsymbol{x}_\tau, a)\right| \le 3\varepsilon'$. Fix $x \in \mathcal{E}$, and let $\mathcal{E}_x$ be the $\mathbb{P}_{\mathcal{P}}$-measurable event

that $\max_{a\in\mathcal{A}}|\widehat{\mathcal{P}}_{\tau,\varepsilon',\delta}[\widehat{V}_{\tau+1}](x,a) - \mathcal{P}_\tau[V_{\tau+1}](x,a)| \le 4\varepsilon'$. From the definition of $\tilde{\pi}_\tau$, we have that

$$D_{\mathrm{tv}}(\widehat{\pi}_\tau(x), \tilde{\pi}_\tau(x))$$

$$= \frac{1}{2}\sum_{a\in\mathcal{A}}\left|\mathbb{P}_{\mathcal{P}}\left[\widehat{\boldsymbol{\pi}}_\tau(x) = a\right] - \mathbb{P}_{\mathcal{P}}\left[\tilde{\boldsymbol{\pi}}_\tau(x) = a\right]\right|,$$

$$= \frac{1}{2}\sum_{a\in\mathcal{A}}\left|\mathbb{P}_{\mathcal{P}}[\mathcal{E}_x]\mathbb{P}_{\mathcal{P}}\left[\widehat{\boldsymbol{\pi}}_\tau(x) = a \mid \mathcal{E}_x\right] + \mathbb{P}_{\mathcal{P}}[\mathcal{E}_x^c]\mathbb{P}_{\mathcal{P}}\left[\widehat{\boldsymbol{\pi}}_\tau(x) = a \mid \mathcal{E}_x^c\right] - \mathbb{P}_{\mathcal{P}}[\mathcal{E}_x]\mathbb{P}_{\mathcal{P}}\left[\tilde{\boldsymbol{\pi}}_\tau(x) = a \mid \mathcal{E}_x\right] - \mathbb{P}_{\mathcal{P}}[\mathcal{E}_x^c]\mathbb{P}_{\mathcal{P}}\left[\tilde{\boldsymbol{\pi}}_\tau(x) = a \mid \mathcal{E}\right]$$

$$\le \frac{1}{2}\sum_{a\in\mathcal{A}}\mathbb{P}_{\mathcal{P}}[\mathcal{E}_x]\cdot\left|\mathbb{P}_{\mathcal{P}}\left[\widehat{\boldsymbol{\pi}}_\tau(x) = a \mid \mathcal{E}_x\right] - \mathbb{P}_{\mathcal{P}}\left[\tilde{\boldsymbol{\pi}}_\tau(x) = a \mid \mathcal{E}_x\right]\right|$$

$$+ \sum_{a\in\mathcal{A}}\mathbb{P}_{\mathcal{P}}[\mathcal{E}_x^c]\cdot\left|\mathbb{P}_{\mathcal{P}}\left[\widehat{\boldsymbol{\pi}}_\tau(x) = a \mid \mathcal{E}_x^c\right] - \mathbb{P}_{\mathcal{P}}\left[\tilde{\boldsymbol{\pi}}_\tau(x) = a \mid \mathcal{E}_x^c\right]\right|, \quad \text{(Jensen's inequality)}$$

and since $\mathbb{P}_{\mathcal{P}}\left[\widehat{\boldsymbol{\pi}}_\tau(x) = a \mid \mathcal{E}_x\right] = \mathbb{P}_{\mathcal{P}}\left[\tilde{\boldsymbol{\pi}}_\tau(x) = a \mid \mathcal{E}_x\right] \forall a \in \mathcal{A}$,

$$= \frac{1}{2}\sum_{a\in\mathcal{A}}\mathbb{P}_{\mathcal{P}}[\mathcal{E}_x^c]\cdot\left|\mathbb{P}_{\mathcal{P}}\left[\widehat{\boldsymbol{\pi}}_\tau(x) = a \mid \mathcal{E}_x^c\right] - \mathbb{P}_{\mathcal{P}}\left[\tilde{\boldsymbol{\pi}}_\tau(x) = a \mid \mathcal{E}_x^c\right]\right|,$$

$$\le \mathbb{P}_{\mathcal{P}}\left[\mathcal{E}_x^c\right],$$

$$= \mathbb{P}_{\mathcal{P}}\left[\max_{a\in\mathcal{A}}|\widehat{\mathcal{P}}_{\tau,\varepsilon',\delta}[\widehat{V}_{\tau+1}](x,a) - \mathcal{P}_\tau[V_{\tau+1}](x,a)| > 4\varepsilon'\right],$$

$$\le \mathbb{P}_{\mathcal{P}}\left[\max_{a\in\mathcal{A}}|\widehat{\mathcal{P}}_{\tau,\varepsilon',\delta}[\widehat{V}_{\tau+1}](x,a) - \mathcal{P}_\tau[\widehat{V}_{\tau+1}](x,a)| > \varepsilon'\right], \quad \text{(see below)} \tag{98}$$

$$\le \delta, \tag{99}$$

where (98) follows from $x \in \mathcal{E}$ and the last inequality follows from Lemma H.2. Therefore, we have

$$\mathbb{E}_{\boldsymbol{x}_\tau\sim\mu}[D_{\mathrm{tv}}(\widehat{\pi}_\tau(\boldsymbol{x}_\tau), \tilde{\pi}_\tau(\boldsymbol{x}_\tau))] \le \mathbb{P}_\mu[\mathcal{E}]\cdot\mathbb{E}_{\boldsymbol{x}_\tau\sim\mu}[D_{\mathrm{tv}}(\widehat{\pi}_\tau(\boldsymbol{x}_\tau), \tilde{\pi}_\tau(\boldsymbol{x}_\tau)) \mid \mathcal{E}] + \mathbb{P}_\mu[\mathcal{E}^c],$$

$$\le \delta + \nu,$$

where the first inequality follows by the fact that the total variation is bounded by 1, and the last inequality follows by (97) and (99). $\qquad\square$

**Lemma L.3.** *Let $x \in \mathbb{R}$ and $\nu \in (0, 1/2)$ be given. Further, let $\zeta \in (0, 1/2)$. Then,*

$$x + \zeta + \nu > \lceil x + \zeta \rceil \quad \text{or} \quad x + \zeta - \nu \le \lceil x + \zeta \rceil - 1,$$

*only if*

$$\lceil x \rceil - \nu \le x + \zeta \le \lceil x \rceil + \nu \quad \text{or} \quad \zeta \le \nu.$$

**Proof of Lemma L.3.** To prove the claim, it suffices to show the following items:

1. $x + \zeta + \nu > \lceil x + \zeta \rceil$ only if $\lceil x \rceil \ge x + \zeta > \lceil x \rceil - \nu$; and

2. $x + \zeta - \nu \le \lceil x + \zeta \rceil - 1$ only if $\lceil x \rceil < x + \zeta \le \lceil x \rceil + \nu$ or $\zeta \le \nu$.

We start by showing the first item. We proceed by showing the contrapositive; that is, we will show that if $x + \zeta \le \lceil x \rceil - \nu$ or $x + \zeta > \lceil x \rceil$, then $x + \zeta + \nu \le \lceil x + \zeta \rceil$. Suppose that $x + \zeta \le \lceil x \rceil - \nu$. This, together with the fact that $\zeta \ge 0$, implies that

$$\lceil x + \zeta \rceil = \lceil x \rceil \ge x + \zeta + \nu.$$

Now, suppose that $x + \zeta > \lceil x \rceil$. Then, we have

$$\lceil x + \zeta \rceil \ge \lceil x \rceil + 1 \ge \lceil x \rceil + \zeta + \nu \ge x + \zeta + \nu,$$

where the penultimate inequality follows by $\zeta, \nu \in (0, 1/2)$.

We now prove the second claim. Again, we proceed by showing the contrapositive; that is, we will show that if $\{\lceil x \rceil + \nu < x + \zeta$ or $\lceil x \rceil \ge x + \zeta\}$ and $\zeta > \nu$, then $x + \zeta - \nu > \lceil x + \zeta \rceil - 1$.

Suppose that $\lceil x \rceil + \nu < x + \zeta$ and $\zeta > \nu$. The first inequality together with $\nu \ge 0$ implies that $\lceil x + \zeta \rceil > \lceil x \rceil$. On the other hand, since $\zeta \le 1/2$, we have $\lceil x + \zeta \rceil \le \lceil x \rceil + 1$, and so

$$\lceil x + \zeta \rceil - 1 = \lceil x \rceil < x + \zeta - \nu,$$

where the last inequality follows by the current assumption that $\lceil x \rceil + \nu < x + \zeta$.

Now, suppose that $\lceil x \rceil \geq x + \zeta$ and that $\zeta > \nu$. Then, we have

$$\lceil x + \zeta \rceil \leq \lceil x \rceil \leq x + 1 < x + \zeta - \nu + 1, \tag{100}$$

where the last inequality follows by $\zeta > \nu$. Rearranging (100) completes the proof. $\qquad\square$

## M  BehaviorCloning Algorithm and Analysis

In this section, we give a self-contained presentation and analysis for the standard *behavior cloning* algorithm for imitation learning (e.g., Ross and Bagnell [46]), displayed in Algorithm 10. Given access to trajectories from an expert policy $\widehat{\pi}_{1:H}$ (which may be non-executable in the sense of Definition 2.1) the algorithm learns an executable policy $\pi^{\mathsf{bc}}$ with similar performance. We use this scheme within RVFS.bc and RVFS$^{\mathsf{exo}}$.bc.

---

**Algorithm 10** BehaviorCloning: Imitation Learning Algorithm.

---

1: **input:** Policy class $\Pi \subseteq \Pi_{\mathsf{S}}$, expert policy $\widehat{\pi}_{1:H}$, suboptimality $\varepsilon \in (0,1)$, and confidence $\delta \in (0,1)$.
2: Set $N_{\mathsf{bc}} = 16H^2 \log(|\Pi|/\delta)/\varepsilon$.
3: Set $\mathcal{D} \leftarrow \varnothing$.
4: **for** $i = 1, \ldots, N_{\mathsf{bc}}$ **do**
5: $\quad$ Generate trajectory $\boldsymbol{\tau} = ((\boldsymbol{x}_1, \boldsymbol{a}_1), \ldots, (\boldsymbol{x}_H, \boldsymbol{a}_H)) \sim \mathbb{P}^{\widehat{\pi}}$.
6: $\quad$ Update $\mathcal{D} \leftarrow \mathcal{D} \cup \{\boldsymbol{\tau}\}$.
7: Compute $\pi^{\mathsf{bc}} \in \arg\min_{\pi \in \Pi} \sum_{((x_1,a_1),\ldots,(x_H,a_H)) \in \mathcal{D}} \sum_{h \in [H]} \mathbb{I}\{\boldsymbol{\pi}_h(x_h) \neq a_h\}$.
8: Return $\pi^{\mathsf{bc}}$.

---

**Proposition M.1.** *Let $\varepsilon, \delta \in (0,1)$ be given and let $\Pi \subseteq \Pi_{\mathsf{S}}$ and $\widehat{\pi}_{1:H}$ be an expert policy such that*

$$\inf_{\pi \in \Pi} \sum_{h=1}^{H} \mathbb{P}^{\widehat{\pi}}[\widehat{\boldsymbol{\pi}}_h(\boldsymbol{x}_h) \neq \boldsymbol{\pi}_h(\boldsymbol{x}_h)] \leq \varepsilon_{\mathsf{mis}}. \tag{101}$$

*Then, the policy $\pi^{\mathsf{bc}}_{1:H} = \text{BehaviorCloning}(\Pi, \varepsilon, \widehat{\pi}_{1:H}, \delta)$ returned by Algorithm 10 satisfies, with probability at least $1 - \delta$,*

$$J(\widehat{\pi}) - J(\pi^{\mathsf{bc}}) \leq 4H\varepsilon_{\mathsf{mis}} + \varepsilon/2.$$

**Proof of Proposition M.1.** First, by the performance difference lemma, we have

$$\mathbb{E}[V_1^{\widehat{\pi}}(\boldsymbol{x}_1)] - \mathbb{E}[V_1^{\pi^{\mathsf{bc}}}(\boldsymbol{x}_1)] = \sum_{h=1}^{H} \mathbb{E}^{\widehat{\pi}}[Q_h^{\pi^{\mathsf{bc}}}(\boldsymbol{x}_h, \widehat{\boldsymbol{\pi}}_h(\boldsymbol{x}_h)) - Q_h^{\pi^{\mathsf{bc}}}(\boldsymbol{x}_h, \boldsymbol{\pi}_h^{\mathsf{bc}}(\boldsymbol{x}_h))],$$

$$\leq H \sum_{h=1}^{H} \mathbb{P}^{\widehat{\pi}}[\widehat{\boldsymbol{\pi}}_h(\boldsymbol{x}_h) \neq \boldsymbol{\pi}_h^{\mathsf{bc}}(\boldsymbol{x}_h)]. \tag{102}$$

We now bound the probability terms on the right-hand side. Fix $h \in [H]$ and let $\mathcal{D}$ be the dataset in Algorithm 10, which consists of $N_{\mathsf{bc}}$ i.i.d. trajectories $((\boldsymbol{x}_1, \boldsymbol{a}_1), \ldots, (\boldsymbol{x}_H, \boldsymbol{a}_H))$ generated by rolling with $\widehat{\pi}_{1:H}$. By Lemma C.4 (with i.i.d. data, $B = H$, and $\mathcal{Q} = \Pi$), we have that, with probability at least $1 - \delta$,

$$\forall \pi \in \Pi, \quad \sum_{((x_1,a_H),\ldots,(x_H,a_H)) \in \mathcal{D}} \sum_{h \in [H]} \mathbb{I}\{\boldsymbol{\pi}_h(x_h) \neq \widehat{\boldsymbol{\pi}}(x_h)\} \leq 2 \sum_{h \in [H]} \mathbb{P}^{\widehat{\pi}}[\boldsymbol{\pi}_h(\boldsymbol{x}_h) \neq \widehat{\boldsymbol{\pi}}_h(\boldsymbol{x}_h)]$$
$$+ \frac{2H \log(2|\Pi|/\delta)}{N_{\mathsf{bc}}}, \tag{103}$$

and

$$\forall \pi \in \Pi, \quad \sum_{h \in [H]} \mathbb{P}^{\widehat{\pi}}[\boldsymbol{\pi}_h(\boldsymbol{x}_h) \neq \widehat{\boldsymbol{\pi}}_h(\boldsymbol{x}_h)] \leq 2 \sum_{((x_1,a_H),\ldots,(x_H,a_H)) \in \mathcal{D}} \sum_{h \in [H]} \mathbb{I}\{\boldsymbol{\pi}_h(x_h) \neq \widehat{\boldsymbol{\pi}}(x_h)\}$$
$$+ \frac{4H \log(2|\Pi|/\delta)}{N_{\mathsf{bc}}}. \tag{104}$$

Taking the infimum over $\pi$ on both sides of (103) and using the definition of $\pi_h^{\mathsf{bc}}$ in Algorithm 10 gives:

$$\sum_{((x_1,a_H),\ldots,(x_H,a_H))\in\mathcal{D}}\sum_{h\in[H]}\mathbb{I}\{\boldsymbol{\pi}_h^{\mathsf{bc}}(x_h)\neq\widehat{\boldsymbol{\pi}}(x_h)\}\leq 2\inf_{\pi\in\Pi}\sum_{h\in[H]}\mathbb{P}^{\widehat{\pi}}[\boldsymbol{\pi}_h(\boldsymbol{x}_h)\neq\widehat{\boldsymbol{\pi}}_h(\boldsymbol{x}_h)]$$
$$+\frac{2H\log(2|\Pi|/\delta)}{N_{\mathsf{bc}}},$$
$$\leq 2\varepsilon_{\mathsf{mis}}+\frac{2H\log(2|\Pi|/\delta)}{N_{\mathsf{bc}}},$$

where the last inequality follows from (101). Using this together with (104), instantiated with $\pi\equiv\pi^{\mathsf{bc}}$, we get that with probability at least $1-\delta$:

$$\sum_{h\in[H]}\mathbb{P}^{\widehat{\pi}}[\boldsymbol{\pi}_h^{\mathsf{bc}}(\boldsymbol{x}_h)\neq\widehat{\boldsymbol{\pi}}_h(\boldsymbol{x}_h)]\leq 4\varepsilon_{\mathsf{mis}}+\frac{8H\log(2|\Pi|/\delta)}{N_{\mathsf{bc}}}.$$

Plugging this into (102), we get that with probability at least $1-\delta$:

$$\mathbb{E}[V_1^{\widehat{\pi}}(\boldsymbol{x}_1)]-\mathbb{E}[V_1^{\pi^{\mathsf{bc}}}(\boldsymbol{x}_1)]\leq 4H\varepsilon_{\mathsf{mis}}+\frac{8H^2\log(2|\Pi|/\delta)}{N_{\mathsf{bc}}}\leq 4H\varepsilon_{\mathsf{mis}}+\varepsilon/2,$$

where the last inequality follows by the fact that $N_{\mathsf{bc}}=16H^2\log(2|\Pi|/\delta)/\varepsilon$. This completes the proof. $\qquad\square$

