# OpenReview forum: "The Power of Resets in Online Reinforcement Learning"
_NeurIPS.cc/2024/Conference — NeurIPS 2024 spotlight_

### Official Review · Reviewer_bJ3L · 2024-07-10

**Soundness:** 3
**Presentation:** 3
**Contribution:** 3
**Rating:** 7
**Confidence:** 3

**Summary:**

This paper study the online RL having access to a local simulator with general function approximation. Their results unclock new statistical guarantees. First, $Q^*$ realizability together with coverability assumption are enough for sample-efficient online RL under this setting. Second, their results further implies that the Exogenous Block MDP problem is tractable under this setting. Finally, they complement their theoretical finding with a computationally efficient algorithm.

**Strengths:**

The paper is in general clearly written. The idea of leveraging local simulator to facilitate online RL with general functionn approximation is novel and leads to several important observations that previous works are out of reach.

**Weaknesses:**

Maybe due to space limit, some terms lack of definition or discussion, see questions for details.
The algorithm in section 4 is highly technical and somewhat difficult to follow. Based on the current context, it appears that the role played by the local simulator is underexplained. One might question if most parts of the algorithm are adaptations of previous techniques to the general function approximation setting. It would be great if the novelty and insights can be highlighted.

**Questions:**

1. What is the double sampling problem mentioned on line 170?

2. What is the distribution shift mentioned on line 313?

3. What is the $v_h$ on line 16 of Alg 2?

Minor issues:

1. Definition of the confidence set in Alg 1, the summation $\sum_{h\leq t}$, is this a typo?
2. Double such that on line 221, and the comma and period in the end.

**Limitations:**

Yes.

---

> ### Author Rebuttal · Authors · 2024-08-06
>
> **Novelty relative to prior work:**
>
> The algorithm presented in Section 4 is far from a simple adaptation of previous techniques. We now highlight some of the key algorithmic innovations:
> - RVFS builds on the DMQ algorithm by [1]. Unlike the latter, we use a core set of state-action pairs instead of policies. This is crucial for our algorithm to work without the strong hypercontractivity assumption made by [1].
> - One of the key algorithmic innovations is the use of Bellman backups (as in line 8 of Algorithm 2) to evaluate whether a new state-action pair should be added to the core set. This technique is unique to our paper and enables handling RL settings beyond linear function approximation. Incorporating Bellman backups in the test at line 8 is essential for proving that the algorithm converges under pushforward coverability.
> - Additionally, the modification of RVFS to the exogenous setting, which includes a randomized rounding, is also novel.
>
> Beyond these algorithmic innovations, the analysis of RVFS introduces new proof techniques, especially in the setting without a gap, that could be beneficial for broader applications beyond the settings studied in this paper.
>
> **What is double sampling?**
>
> What is the double sampling problem mentioned on line 170?
> The double sampling problem refers to the challenge of estimating $T_h g_{h+1}(x_h,a_h)=\mathbb{E}[g_{h+1}(x_{h+1}) \mid x_h,a_h]$ in (1), which requires more than one next-state sample starting from the state $x_h$. Generating multiple samples from any given state is only possible with resets; in the online setting without resets, a state $x_h$ may only be observed once, allowing only a single next-state sample. See e.g. https://x.com/nanjiang_cs/status/1672702613744410624 for more details on the double sampling problem.
>
> **What is distribution shift?**
>
> The distribution shift mentioned 313 is a well-known phenomenon the situation where for a new state action pair $(x_{\ell-1},a_{\ell-1})$, the distribution of the next state $x_{\ell}$ puts mass on new regions of the state space where the estimated value function is inaccurate.
> See e.g., sec 7.3 in https://arxiv.org/pdf/2312.16730 for more detail on the distribution shift phenomenon in RL.
>
> **What is the $v_h$ on line 16 of Alg 2?**
>
> $v_h$ represents an esitmate of $\mathbb{E}^{\hat\pi}[\sum_{\ell= h}^H r_\ell \mid x_h]$---see how the set $\mathcal{D}_h$ is constructed on line 15.

---

> > ### Comment · Reviewer_bJ3L · 2024-08-11
> >
> > Thanks for the rebuttal. It addressed my questions.

---

### Official Review · Reviewer_1avF · 2024-07-12

**Soundness:** 3
**Presentation:** 2
**Contribution:** 3
**Rating:** 7
**Confidence:** 3

**Summary:**

The authors show that local simulator access removes Bellman completeness for MDPs with bounded coverability.  This generalizes the results of existing works that are limited to low-rank or linear structures.

On the statistical front, the authors analyze the sample complexity of SimGOLF using coverability, but the algorithm itself is computationally inefficient.
To resolve this they propose RVFS, which is computationally efficient with respect to (1) a regression oracle, and (2) a convex optimization oracle over the value function class $\mathcal{V}$. In order to obtain sample complexity guarantees here, however, stronger assumptions such as pushforward coverability or $Q^\star$ gap are required.

(For some context behind the proceeding comments, I am familiar with the analysis of GOLF and its variants [4,31,61], but not with the local simulator literature [38,63,57] beyond a brief glance. )

**Strengths:**

The motivation for the paper is clear, and generalizing RL in local simulators to structures beyond linear is a valuable contribution.

SimGOLF/ Thm 3.1 are clean results, and I appreciate that the authors tackled the problem of computational efficiency in RVFS.

**Weaknesses:**

My impression is that the analysis of SimGOLF / Theorem 3.1 has limited technical novelty given [61]. That's not to say it's not a valuable contribution.

**Oracle efficiency**

My biggest concern is that RVFS does not seem as "oracle-efficient" as claimed, and the discussion on this is somewhat lacking. For example, the authors could have stated more exactly what their definition of computational efficiency is (or is it just that it "reduces to convex optimization"?). They might have also provided an analysis on the # of calls to computational oracles before convergence.

In particular, it seems RVFS needs to solve the convex optimization in Line 8 for every $(x,a)$ in the core set, for $N_{test}$ times (which is like $\varepsilon^{-1}$...?), and then for every possible action. Further, RVFS makes recursive calls to itself. Can this avoid $\mathrm{poly}(\mathcal{X}, \mathcal{A}, \varepsilon^{-1})$?

More minor, but it seems $\mathcal{V}$ might need to be a convex set for efficient calculation of Line 8 (L355-358), which does not immediately gel with the "holds for general function approximation" or "neural networks" claims? In the grand scheme of things I'm sort of not that bothered by this.

**Assumptions for RVFS**

I am not opposed to make stronger assumptions for RVFS necessarily, but I would have liked to know more about why. For example, is it possible to give some intuition for why coverability alone is insufficient to analyze RVFS? And how do the gap or pushforward coverability assumptions help with this?

**Comparison to existing work**

The authors mention that local simulator algorithms utilizing core sets exist for linear value function approximation exist, and I imagine that RVFS might also be applied to these settings. However, I find the consideration of previous literature to be quite limited, and I would have appreciated some discussion on how the sample/computational complexity of the algorithms compare (if they are comparable).

**Questions:**

The questions below also include the specific ones from "weaknesses" that I would like to be answered. I would be happy to raise my score if my concerns are adequately addressed.

1. Is it possible to analyze the complexity of oracle calls required to achieve the learning guarantee in Theorem 4.1? Will it be dependent on $\mathcal{X}$?
2. I imagine RVFS can be applied to the linear $V^\star$ or $Q^\star$ settings from [55,57,38]. Are the guarantees for RVFS comparable? I'm not necessarily looking for "better" performance because RVFS is more general, but just some sense of how big the gap might be (if there is one). Or just some discussion on the topic as I myself am not familiar.
3. Could you comment on the barriers to obtaining guarantees for RVFS under coverability, and/or why gap / pushforward helps?
4. The definition of the confidence set in Algorithm 1 seems a little weird (maybe typo?), particularly the $\sum_{h \le t}$. Currently it seems like you throw samples from previous iterations away, is that intentional?

**Limitations:**

I believe the limitations of RVFS could benefit from greater discussion per my previous comments.

---

> ### Author Rebuttal · Authors · 2024-08-06
>
> **Technical novelty of SimGOLF in light of [1]**
>
> - We agree that the main technique behind our first result, SimGOLF, is quite simple, but we view this as a positive.
> - In particular, our result shows that the challenging coverability + realizability setting, which was explicitly left as an open problem in prior work and allows for nonlinear function approximation, is tractable under local simulator access.
> - Therefore, even though the proof for the fact that this result is tractable turns out to be remarkably simple, we think that the take-away message of the result is significant enough to merit publication.
> - Finally, we view the simplicity as a strength, since it will likely enable other researchers to build on this technique and explore whether it extends to other interesting settings. Notably, this is the starting point for our second main result, RVFS, which attacks the even more challenging problem of designing statistically and computationally efficient algorithms. Indeed, given the complexity of RVFS, it is very difficult to imagine directly proving such a result or designing such an algorithm without already being aware that the setting under consideration is tractable, which is precisely what SimGOLF provides.
>
>
> **Oracle type:**
>
> The Oracle we require is precisely the one described on Line 356, which solves a convex optimization problem in function space. We note that similar Oracles have been used in prior works; see e.g. [3] (RegCB).
>
> **Number of Oracle calls:**
>
> Although not explicitly stated in Theorem 4.1, our analysis shows that the number of Oracle calls is polynomial in $C_{\text{cov}}, H, A, \log |\mathcal{V}|$, and $1/\epsilon$, with no dependence on $|\mathcal{X}|$. This holds despite the recursion in RVFS. We would not consider the algorithm Oracle efficient if the number of Oracle calls depended on $|\mathcal{X}|$. We will include the number of Oracle calls in Theorem 4.1.
>
> **Convexity of the function class:**
>
> The optimization problem in Line 8 would have the same value if V is replaced by its convex hull. This is because the optimization problem, which is of the form $\sup_{f \in \mathcal{V}_h} |\mathrm{Objective}(f)|$, where $\mathrm{Objective}(f)$ is linear if $f$, can be solved by considering the two subproblems (without the absolute value)
>
> $\sup_{f \in \mathcal{V}_h} \mathrm{Objective}(f)$ and
>
> $\sup_{f \in \mathcal{V}_h} -\mathrm{Objective}(f)$.
>
> Since the $\mathrm{Objective}(f)$ is linear in $f$, the suprema in these subproblems will always be attained at vertecies of $\mathcal{V}_h$, so the convex hull would not change the values of these objectives. We will provide more details on this.
>
>
> **Coverability alone for RVFS:**
>
> The SimGolf algorithm uses the concept of global optimism, where the estimated value functions at each iteration are greater than the optimal value function in a pointwise sense; this greatly simplifies the analysis of the coverability setting using SimGolf. In contrast, RVFS does not employ global optimism.
>
> There are RL settings that are only known to be statistically tractable using a global optimism approach; for example, the linear Bellman complete and the linear Q*/V* settings (see e.g. [3]). Our paper shows that a local simulator makes the coverability setting statistically tractable using global optimism (through SimGolf). It is unclear if the stronger pushforward-coverability assumption is necessary for algorithms that do not employ global optimism, like RVFS. We could not find a way to make the analysis of RVFS work with full coverability; the reasons for this are deeply rooted in the analysis of RVFS.
>
>
> **Handling the linear $Q^\star/V^\star$ settings:**
>
> RVFS can be slightly modified to handle the linear $V^\star$ and $Q^\star$ settings.
> Similar to existing results for these settings, the corresponding sample complexity would be polynomial in $d$, $1/\epsilon$, and $H$, without any dependence on $|A|$ or $|X|$. However, we have not worked out the exact exponents of $d$, $1/\epsilon$, and $H$ in the sample complexity. We would be happy to include such an extension in the final version.
> With the additional page available in the camera-ready version, we will provide a more detailed comparison to existing results.
>
> **Summation in the confidence set of Algorithm 1:**
>
> The sum in the definition of the confidence set of simGolf contains a typo indeed. We will correct this in the camera-ready version.
>
>
> [1] Xie, Tengyang, Dylan J. Foster, Yu Bai, Nan Jiang, and Sham M. Kakade. "The role of coverage in online reinforcement learning." arXiv preprint arXiv:2210.04157 (2022).
>
> [2] Foster, Dylan J., Alexander Rakhlin, David Simchi-Levi, and Yunzong Xu. "Instance-dependent complexity of contextual bandits and reinforcement learning: A disagreement-based perspective." arXiv preprint arXiv:2010.03104 (2020).
>
> [3] Kane, Daniel, et al. "Computational-statistical gap in reinforcement learning." Conference on Learning Theory. PMLR, 2022.

---

> > ### Comment · Reviewer_1avF · 2024-08-10
> > **Follow-up re: number of oracle calls**
> >
> > Thank you for your detailed reply, which has addressed most of my comments. However, my primary concern was the number of Oracle calls required (which is important for computational efficiency), and I'd like to ask one follow-up question about this.
> >
> > The rebuttal states that RVFS requires only polynomial in $C_{\mathrm{cov}}$, not $|\mathcal{X}|$, calls. Does the analysis currently in the paper show this, and is there a section or result that I can reference to see this (or at least get a sense of the argument)?
> >
> > I wasn't able to find this given a brief scan of the appendix, and I apologize if I've missed it. Thank you.

---

> > > ### Author Response · Authors · 2024-08-10
> > > **Clarification around the number of  Oracle. calls**
> > >
> > > Thank you for your interest! The current analysis does indeed bound the number of Oracle calls, as we will clarify now.
> > >
> > > First, let's look at the full version of RVFS in Algorithm 5.
> > > - The Oracle in question is invoked in the test of Line 14.
> > > - To bound the number of Oracle calls is the same as bounding the number of times Line 14 is executed, or the number of times the $\widehat{P}$ operator in Line 14 is called throughout the execution of $\mathrm{RVFS}\_0$; this includes all subsequent recursive calls to $(\mathrm{RVFS}\_{h})\_{h\in [H]}$.
> > > - The proof of Lemma I.2 directly bounds the number of times $T_\ell$ the operator $\widehat{P}$ in Line 14 is called throughout the execution of $\mathrm{RVFS}\_0$ (again this takes into account all subsequent recursive calls to $(\mathrm{RVFS}\_{h})\_{h\in [H]}$).
> > > - In particular, Eq. 34 bounds $T_\ell$ by $M^3 N_\mathrm{test} H^3$, where $M$ and $N_{\mathrm{test}}$ are as in Algorithm 5; these are polynomial in problem parameters and do *not* depend on $\mathcal{X}$.
> > >
> > > We hope this answers your question and will be happy to highlight that the number of oracle calls is bounded in the final revision of the paper. Please let us know if you have any other questions.

---

> > > > ### Comment · Reviewer_1avF · 2024-08-14
> > > >
> > > > Thanks for your response. I think that this paper has a number of interesting contributions and techniques, and more than meets the bar for acceptance. I've increased my score accordingly.
> > > >
> > > > I do feel that the paper could have benefited significantly from expanded discussion of the algorithms, results, and proof techniques, though I imagine that this was hard to do given the page limit.

---

### Official Review · Reviewer_XHMe · 2024-07-23

**Soundness:** 3
**Presentation:** 2
**Contribution:** 3
**Rating:** 6
**Confidence:** 2

**Summary:**

The paper introduces the SimGolf algorithm, which leverages local simulator access to reset to previously visited states and sample multiple trajectories. This approach enhances sample efficiency and accuracy in value function approximation, particularly in high-dimensional MDPs. The SimGolf algorithm uses local simulator access to achieve new statistical guarantees, allowing for efficient learning in environments with low coverability. Additionally, the paper presents RVFS (Recursive Value Function Search), a computationally efficient algorithm that achieves sample complexity guarantees under a strengthened statistical assumption known as pushforward coverability.

**Strengths:**

- Introduces a novel approach that uses reset capability in reinforcement learning to significantly improve sample efficiency.
- Provides strong theoretical analysis and new statistical guarantees for reinforcement learning with local simulator access.
- Proposes two innovative algorithms (SimGolf and RVFS) with clear theoretical benefits.
- The paper is generally well-written and explains the new algorithms and their theoretical foundations clearly.
- Addresses a significant problem in reinforcement learning by enhancing sample efficiency and providing robust theoretical guarantees.

**Weaknesses:**

- The paper breaks with the expected shape of a NeurIPS paper (numbered list in abstract, missing discussion or future work). While deviations are acceptable if justified, the current format lacks some important aspects.
- The abstract includes references and attempts to serve as a conclusion, which is unconventional and detracts from its clarity. The abstract should be a short, plain summary of the paper It is also not the place to reference other works, as the abstract should be self contained.
- The paper lacks a dedicated conclusion or discussion section, which would be crucial for contextualizing results and suggesting future work.
- There is no experimental data provided (which is fine for a theory paper), but there should then at least be a discussion on expected practical benefits and an outline of plans for empirical validation as part of a future works section. In the area of RL we observe a significant gap between the theoretical understanding of performance guarantees and the actual observed performance of certain setups in practice. I would therefore regard the benefit of a paper, that restricts itself to only theoretical work itself as limited; actual experiments are needed to validate the practical applicability of the new findings. Please outline such in a future works section.

**Questions:**

- How does SimGolf perform in practical environments compared to regular RL and MCTS?
- What specific types of environments are expected to benefit most from SimGolf?
- Are there any preliminary empirical results or plans for validation in real-world applications?
- The training process produces a final policy and set of Q functions. It should therefore be possible to use the presented algorithm for 'pre-training' in a simulator with reset capabilities, but then continue training with a SAC-like approach on real world data?

- The paper lacks an explicit differentiation from existing methods. Is the following summary correct for contextualizing the work? (This is mostly just for my own understanding of the presented work)
   - Regular RL requires only the ability to take actions and observe resulting new states, enabling linear rollouts. It performs single, uninterrupted rollouts without the ability to reset.
   - MCTS (Monte Carlo Tree Search) requires a complete and known transition function, allowing iteration over all possible child nodes from each state. It builds and expands a search tree incrementally by iterating over all possible actions and simulating outcomes. This method utilizes comprehensive exploration through rollouts and backpropagation within the search tree, resulting in high accuracy of value estimation. However, it is computationally intensive and requires full knowledge of the transition dynamics.
   - SimGolf strikes a balance between the simplicity of regular RL and the higher performance of MCTS. It requires the ability to reset to previously visited states and resample the next state under the same or different actions. This method does not necessitate complete iteration over all possible actions and transitions. Instead, it enhances sample efficiency by allowing targeted resampling after initial rollouts (under the same or different actions). The agent can reset to critical states identified during the initial exploration and simulate multiple future trajectories from those states, improving the accuracy of value function estimation. SimGolf uses the reset capability to gather multiple samples from key state-action pairs, updating a confidence set of value functions based on empirical Bellman error estimates, combining the benefits of comprehensive exploration with more modest demands on the environment.

I was assigned to this paper after the regular review period so I did not have time to go through the presented math inn detail. Rely on the other reviews to check those.

**Limitations:**

- The lack of experimental validation means the practical benefits and real-world performance of the algorithm remain uncertain.
- A discussion on future work and plans for empirical validation would be beneficial, detailing how the authors intend to demonstrate practical improvements.

---

> ### Author Rebuttal · Authors · 2024-08-06
>
> **Paper format:**
>
> Thank you for your feedback. We understand the importance of adhering to a standard format; however, we chose to prioritize a detailed explanation of our novel and complex algorithm to ensure that its intricacies were fully conveyed. We believe this approach still aligns with the presentation style of other papers accepted at NeurIPS. With the additional page available for the camera-ready version, we will include a more detailed discussion section and suggestions for future work to further enhance the clarity and completeness of our paper.
>
> **Experimental results:**
>
> While we acknowledge the importance of empirical results, these are beyond the scope of the current paper. Our focus is to understand the sample and computational complexity of RL when a local simulator is available. In terms of understanding the theoretical limitations of RL, we believe the results presented here are significant on their own. However, our paper does pave the way for new empirical questions, which we are excited to explore in future works, as we will highlight in the future works section.
>
> **How does SimGolf perform in practical environments compared to regular RL and MCTS?**
>
> SimGolf was never intended to be a practical algorithm. Its primary purpose is to demonstrate that the coverability setting, which we argue is a general RL setting, is statistically tractable under resets—a fact that was not previously known. For practical applications, RVFS, which shares similarities with MCTS and Go-Explore, is the more viable algorithm. We discuss the similarities between RVFS and MCTS in detail on page 9, under the section ‘Connection to empirical algorithms.’
>
> **What specific types of environments are expected to benefit most from SimGolf?**
>
> SimGolf operates under the coverability assumption, an intrinsic structural property of the underlying MDP. Therefore, environments expected to benefit most from SimGolf include Low-Rank MDPs and (Exogenous) Block MDPs, both of which satisfy coverability. These environments exhibit high-dimensional state spaces and require nonlinear function approximation. Exogenous Block MDPs, described in Section 3.3, capture real-world RL settings where observations include a high-dimensional, time-correlated signal irrelevant to the RL task. Learning to ‘ignore’ this signal in a sample-efficient manner is extremely challenging, an issue explicitly left as an open problem in previous work. However, we demonstrate in this paper that SimGolf and RVFS make this possible when resets are available. Additionally, we expect RVFS to perform well even in settings that do not strictly satisfy coverability, including those where MCTS has been used. RVFS can be viewed as a more principled version of MCTS.
>
> **Using our algorithms for 'pre-training':**
>
> As the cost of simulating real-world environments decreases, efficient algorithms like ours, which utilize a local simulator, will become extremely valuable for pretraining. One of the key points of our paper is to demonstrate that a local simulator enables the development of both statistically and computationally efficient algorithms for challenging RL settings, such as the Exogenous Block MDP setting.
>
>
> **Differentiation from existing methods:**
>
> We would like to reiterate that the primary objective of our paper and the algorithms we present is not to propose a competitor to MCTS. The goal is to understand the sample and computational complexity of RL under coverability when a local simulator is available.
> That said, your understanding of online RL and the workings of MCTS appears correct. We would like to add that MCTS has several major limitations compared to RVFS, including:
> - MCTS requires finite states to iterate over all possible child nodes of each state, making it inapplicable in environments with continuous states, unlike SimGolf and RVFS.
> - MCTS does not come with any provable sample-complexity guarantees and can fail even in simpler tabular RL settings. One reason for this is that MCTS (or some versions of it) uses rollouts with a fixed default policy (usually a policy that takes random actions), and the estimated value will correspond to this rollout policy. In contrast, our work reveals (through RVFS) that one needs to use the policy induced by the current estimated value function for rollouts. This approach is key for proving the sample complexity guarantees of RVFS. This simple idea can potentially be used to modify MCTS for better performance.
>
> We will include a more detailed comparison between RVFS and MCTS in the camera-ready version.

---

> > ### Comment · Reviewer_XHMe · 2024-08-10
> >
> > Thank you for your detailed rebuttal and clarifications. I appreciate the effort to address the concerns raised, particularly regarding the paper's format and the absence of a discussion section. Your explanation of the intended purpose of SimGolf, its relationship to RVFS, and how these algorithms fit within the broader RL framework is much clearer now.
> >
> > While I did not delve deeply into the mathematical details, the overall feedback from other reviewers and your responses indicate that the theoretical contributions are meaningful and have the potential to advance the field.
> >
> > Given your clarifications and the planned improvements, I have decided to adjust my rating from borderline reject (4) to weak accept (6).

---

### Official Review · Reviewer_DqGS · 2024-07-30

**Soundness:** 2
**Presentation:** 2
**Contribution:** 2
**Rating:** 4
**Confidence:** 1

**Summary:**

The paper presents some theoretical results for new reinforcement learning algorithms with a sophisticated approach to a simulator environment.

**Strengths:**

Paper presents an extensive theoretical study.

**Weaknesses:**

Practical applications of the algorithm remain questionable.

The modifications themselves might seem trivial.

**Questions:**

A clarification of the application of the algorithms and their implementation might improve the paper.

**Limitations:**

I don't think there's any negative societal impact.

---

### Decision · Program_Chairs · 2024-09-25

**Decision:**

Accept (spotlight)

**Comment:**

This paper presents a powerful use of local simulator access in online reinforcement learning with general function approximation. The authors introduce two new algorithms, SimGolf (inspired by a previous work) and RVFS. Both algorithms achieve sample efficient gaurantees in this setting. While the first uses global optimism and is not computational, the latter solves this problem by strengthening some of the underlying coverability and representation assumptions.

SimGolf leverages global optimism, augmented with local simulator access, to achieve sample-efficient learning while requiring only Q*-realizability and coverability. This significantly relaxes the representational constraints required by previous work. A key innovation lies in the direct estimation of squared Bellman error using local simulator access. This method is simple, and potentially very useful for practical applications.

The authors acknowledge the inherent computational complexities associated with global optimism, and introduce RVFS, a computationally efficient algorithm based on a refined recursive exploration scheme guided by core-sets.  RVFS offers provable guarantees under pushforward coverability, a new statistical assumption the authors introduce. The authors acknowledge that the high polynomial dependence on problem parameters in the sample complexity bounds necessitates further research of tighter bounds and their practical implications.

Overall, this work makes a significant contribution to the theoretical understanding of RL with general function approximation and I strongly recommend to accept it.